# Cell volume changes contribute to epithelial morphogenesis in zebrafish Kupffer's vesicle

Agnik Dasgupta[1], Matthias Merkel[2], Madeline J Clark[1], Andrew E Jacob[1], Jonathan Edward Dawson[2], M Lisa Manning[2]*, Jeffrey D Amack[1]*

[1]Department of Cell and Developmental Biology, State University of New York, Upstate Medical University, Syracuse, United States; [2]Department of Physics, Syracuse University, Syracuse, United States

**Abstract** How epithelial cell behaviors are coordinately regulated to sculpt tissue architecture is a fundamental question in biology. Kupffer's vesicle (KV), a transient organ with a fluid-filled lumen, provides a simple system to investigate the interplay between intrinsic cellular mechanisms and external forces during epithelial morphogenesis. Using 3-dimensional (3D) analyses of single cells we identify asymmetric cell volume changes along the anteroposterior axis of KV that coincide with asymmetric cell shape changes. Blocking ion flux prevents these cell volume changes and cell shape changes. Vertex simulations suggest cell shape changes do not depend on lumen expansion. Consistent with this prediction, asymmetric changes in KV cell volume and shape occur normally when KV lumen growth fails due to leaky cell adhesions. These results indicate ion flux mediates cell volume changes that contribute to asymmetric cell shape changes in KV, and that these changes in epithelial morphology are separable from lumen-generated forces.
DOI: https://doi.org/10.7554/eLife.30963.001

*For correspondence:
mmanning@syr.edu (MLM);
amackj@upstate.edu (JDA)

Competing interests: The authors declare that no competing interests exist.

## Introduction

In an embryo, epithelial cells undergo tightly regulated shape changes to drive tissue remodeling and organ development. Changes in epithelial cell morphology can be mediated by intrinsic mechanisms, such as rearrangements of the actomyosin cytoskeleton that often occur in response to biochemical signaling cascades (*Fuss et al., 2004*; *Escudero et al., 2007*). Extrinsic biophysical forces can also influence epithelial morphogenesis (*Navis and Nelson, 2016*; *Navis and Bagnat, 2015*). For example, the mechanical properties of neighboring cells can help shape how an epithelium develops (*Luu et al., 2011*; *Sedzinski et al., 2016*). Another source of extrinsic force found in many organs is a fluid-filled lumen. Forces generated by increased fluid pressure during lumen expansion can have an impact on individual cell shapes and the overarching epithelial architecture (*Bagnat et al., 2010*). Conversely, movements of fluids from epithelial cells to the lumen have been proposed to regulate both lumen growth and thinning of the epithelium (*Hoijman et al., 2015*). Thus, exactly how intrinsic molecular mechanisms and extrinsic mechanical forces interact to regulate epithelial cell shape changes during organogenesis remains an open and intriguing question.

In this study, we used the zebrafish Kupffer's vesicle (KV) as a model organ to investigate mechanisms that regulate shape changes of single cells during epithelial morphogenesis. KV, which functions as a 'left-right organizer' to determine left-right asymmetry of the zebrafish embryo (*Essner et al., 2005*; *Kramer-Zucker et al., 2005*), is a transient organ comprised of a single layer of ~50 epithelial cells that surround a fluid-filled lumen. Each KV cell extends a motile monocilium into the lumen to generate asymmetric fluid flows that direct left-right patterning signals. Fate mapping studies have identified precursor cells, called dorsal forerunner cells (DFCs), which differentiate

into epithelial KV cells at the end of gastrulation stages of development (*Melby et al., 1996*; *Cooper and D'Amico, 1996*). These cells then form the KV organ in the tailbud at the embryonic midline (marked by notochord) during early somitogenesis stages (*Figure 1A*). Similar to other organs that develop a fluid-filled lumen—such as the gut tube (*Alvers et al., 2014*) or pancreas (*Villasenor et al., 2010*)—KV cells form a rosette-like structure to give rise to a nascent lumen that expands over time (*Amack et al., 2007*; *Oteíza et al., 2008*). Previous 2-dimensional (2D) analyses of KV cells revealed that during expansion of the KV lumen, KV cells at the middle focal plane undergo asymmetric cell shape changes along the anteroposterior (AP) axis that sculpt the architecture of the mature KV organ (*Wang et al., 2012*). A transgenic strain developed in this study, *Tg (sox17:GFP-CAAX)*, provides bright labeling of KV cell membranes with green fluorescent protein (GFP) that shows the AP asymmetric architecture of the whole organ (*Video 1*) and the 2D shapes of epithelial KV cells (*Figure 1A*). KV cells have similar shapes at early stages of development, whereas at later stages the cells in the anterior half of KV (KV-ant cells) develop columnar morphologies that allow tight packing of these cells and posterior KV cells (KV-post cells) become wide and thin (*Wang et al., 2012*) (*Figure 1B*). This morphogenetic process, which we refer to as 'KV remodeling', results in an AP asymmetric distribution of cilia that is necessary to drive fluid flows for left-right patterning (*Wang et al., 2012*). Thus, KV is a simple and accessible organ that is ideal for probing the relationship between intrinsic and extrinsic mechanisms that drive epithelial morphogenesis.

Previous studies of KV have successfully contributed to our understanding of how epithelial cell shapes are regulated during embryogenesis, but the mechanisms that control KV cell shape changes are not fully understood. Experimental results and mathematical simulations from our group indicate that actomyosin contractility and differential interfacial tensions between KV cells mediate asymmetric cell shape changes (*Wang et al., 2012*). Additional studies identified an AP asymmetric deposition of extracellular matrix (ECM) implicated in restricting anterior KV cell shape during KV lumen expansion (*Compagnon et al., 2014*). We reasoned that these mechanisms likely work in concert with yet additional mechanisms to fully instruct epithelial morphogenesis during KV organ formation. Here, we developed methods to analyze single KV cells in 3-dimensions (3D) and created novel mathematical vertex models of KV development to identify mechanisms that contribute to AP asymmetric epithelial cell shape changes in KV. 3D analyses revealed that KV-ant cells increase their volume and KV-post cells decrease their volume during KV morphogenesis. These asymmetric cell volume changes occur at the same time as asymmetric cell shape changes. At the molecular level, KV cell volume and shape changes are mediated by ion channel activity that regulates ion flux and fluid transport. We next tested whether extrinsic biophysical forces had an impact on these cell morphology changes. Mathematical models indicate that mechanical properties of external tissues surrounding the KV can impact cell shape changes in the KV. Models predicted that when external tissues are solid-like, asymmetric cell volume changes in KV cells contribute to cell shape changes even in the absence of lumen expansion. Consistent with mathematical model predictions, experimental perturbations of lumen expansion indicated that changes in KV cell volume and shape can occur independent of forces associated with lumen growth. Together, our results suggest ion channel mediated fluid flux serves as an intrinsic mechanism to regulate epithelial cell morphodynamics that create asymmetry in the KV organ. These findings shed new light on the interplay between lumenogenesis and epithelial morphogenesis and provide an example of cell morphology changes that can be uncoupled from mechanical forces exerted during lumen expansion.

## Results

### Mosaic labeling enables 3D analysis of single KV cells

To investigate 3D behaviors of KV cells we first generated stable *Tg(sox17:GFP-CAAX)* transgenic zebrafish using a *sox17* promoter (*Sakaguchi et al., 2006*) to express membrane localized GFP (GFP-CAAX) in KV cells. This transgene marks all cells in the KV and is useful for delineating 2D cell morphology (*Figure 1A*). However, due to difficulties in determining exact cell-cell boundaries in the KV (*Video 1*), this strain is not ideal for visualizing individual KV cells in 3D. Therefore, we next developed a Cre-*loxP* based mosaic cell labeling method to visualize single KV cells. For this approach, we generated transgenic *Tg(sox17:Cre^ERT2)* zebrafish that expresses Cre recombinase in the KV cell lineage that has inducible activity through the addition of 4-hydroxytamoxifen (4-OHT) (*Feil et al.,*

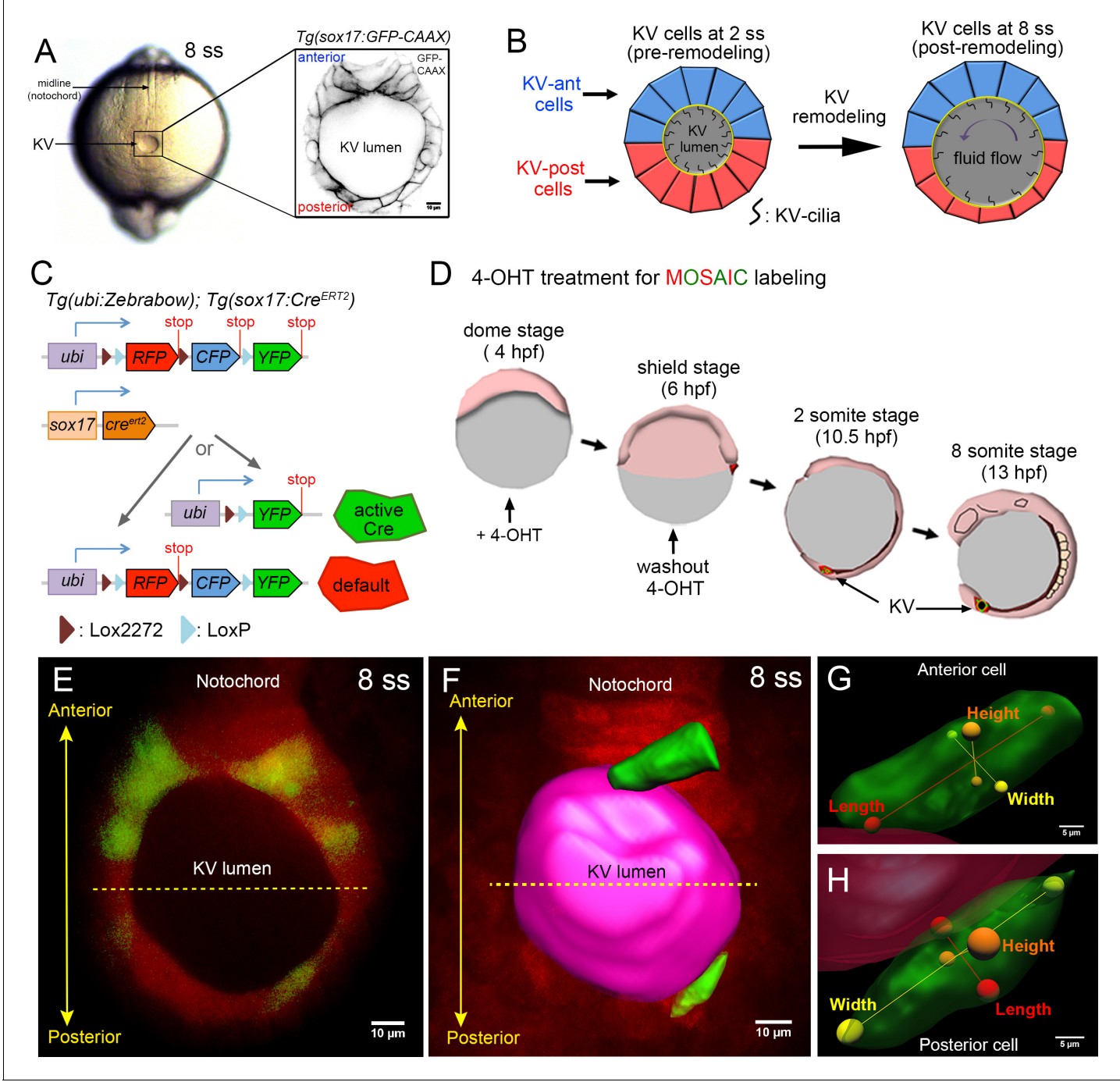

**Figure 1.** Mosaic labeling of KV cells. (**A**) A dorsal view of the tailbud in a live zebrafish embryo at the 8-somite stage (8 ss) of development. Kupffer's vesicle (KV) is positioned at the end of the notochord. The inset shows GFP-labeled KV cells surrounding the fluid-filled KV lumen in a *Tg(sox17:GFP-CAAX)* transgenic embryo at 8 ss. This is the middle plane of the KV. Scale = 10 µm. (**B**) Schematic of cell shape changes during KV remodeling. KV-ant cells (blue) and KV-post cells (red) have similar shapes at 2 ss, but then undergo regional cell shape changes such that KV-ant cells are elongated and KV-post cells are wide and thin by 8 ss. These cell shape changes result in asymmetric positioning of motile cilia that generate fluid flows for left-right patterning. (**C**) Structure of the *ubi:zebrabow* and *sox17:CreERT2* transgenes and the possible recombination outcomes of the 'zebrabow' transgene by Cre recombinase activity in KV cell lineages. (**D**) Time course of mosaic labeling of KV cells. Brief treatment of double transgenic *Tg(sox17:CreERT2); Tg (ubi:Zebrabow)* embryos with 4-OHT from the dome stage to the shield stage generates low levels of Cre activity that changes expression of default RFP to expression of YFP in a subset of KV cells. (**E**) Mosaic labeled YFP+ KV cells at the middle plane of KV at 8 ss. (**F**) 3D reconstructed KV cells (green) and KV lumen (magenta) at 8 ss. Scale = 10 µm. (**G–H**) Morphometric parameters of 3D rendered KV-ant (**G**) and KV-post (**H**) cells: length = axis

*Figure 1 continued on next page*

*Figure 1 continued*

spanning from apical to basal side of the cell, height = axis spanning from dorsal to ventral side of the cell, and width = axis connecting lateral sides of the cell. Scale = 5 µm.

DOI: https://doi.org/10.7554/eLife.30963.002

*1997*). Transgenic *Tg(sox17:Cre^{ERT2})* fish were crossed with a previously described *Tg(ubi:Zebrabow)* ('zebrabow') strain (*Pan et al., 2013*) that can be used to generate differential fluorescent labeling of cells based on stochastic Cre-mediated recombination of the zebrabow transgene (*Figure 1C*). Double transgenic *Tg(sox17:Cre^{ERT2})*; *Tg(ubi:Zebrabow)* embryos were briefly treated with 4-OHT from the dome stage (4 hr post-fertilization or hpf) to the shield stage (6 hpf) (*Figure 1D*) to induce low levels of Cre activity that resulted in a switch from the default RFP (red fluorescent protein) expression to YFP (yellow fluorescent protein) expression in a subset of KV cells (*Figure 1E*). This strategy reliably created mosaic labeled KVs containing a few YFP$^+$ KV cells with boundaries that are easily distinguished from surrounding RFP$^+$ cells (*Video 2*). Images of mosaic labeled KVs in living *Tg(sox17:Cre^{ERT2})*; *Tg(ubi:Zebrabow)* embryos were volume-rendered using Imaris (Bitplane) software to generate 3D reconstructions of the lumen and KV cells (*Figure 1F*). To assess the 3D morphology of individual KV cells residing in either the anterior or posterior region of the KV organ, we used the lumen surface as a reference point to ensure that all 3D datasets were analyzed from the same perspective. To make morphometric measurements, we defined three axes of KV cells: (1) cellular 'height (H)' is the length of the axis connecting dorsal and ventral surfaces of the cell, (2) cellular 'length (L)' is the length of the axis connecting apical (associated with the lumen) and basal surfaces of the cell, and (3) cellular 'width (W)' is the length of the axis connecting two lateral sides of the cell (*Figure 1G,H*). The combination of mosaic labeling, live imaging and 3D cell morphometrics provides a new approach to investigate the cellular and mechanical mechanisms underlying KV epithelial morphogenesis at single cell resolution.

To assess the dynamics of KV cells in 3D, we first performed time-lapse imaging of mosaic labeled KVs in live *Tg(sox17:Cre^{ERT2})*; *Tg(ubi:Zebrabow)* embryos from the 2-somite stage (2 ss) when the lumen first forms to the 8 somite stage (8 ss) when the lumen is fully expanded (*Amack et al., 2007*; *Wang et al., 2012*; *Gokey et al., 2016*). These stages span the process of KV remodeling when KV cells change their shapes: 2 ss is pre-remodeling and 8 ss is post-remodeling (*Wang et al., 2012*). Time-lapse images from these stages indicated KV cells are highly dynamic during KV morphogenesis (*Video 3*). To quantify KV cell dynamics, we took several precautions in subsequent experiments to avoid artifacts. First, to avoid potential photobleaching from time-lapse imaging, live embryos were imaged only once at one stage of development—not continuously or at multiple stages. Second, to avoid differences in fluorescence signal due to differences in imaging depth, all KVs were visualized laterally (YZ orientation) and only mosaically labeled cells from the middle plane of the KV organ that is perpendicular to the dorsoventral axis were selected for analysis (*Figure 2A*). We define the middle plane as the plane with the largest lumen diameter when viewed dorsally (*Figure 1A*). Finally, we determined that the Cre activity was not spatially biased, but rather randomly labeled cells throughout the KV. By analyzing enough embryos, we sampled KV cells from all positions along the middle plane of KV at different stages of development (*Figure 2—figure supplement 1*).

To quantify the morphometric properties of KV-ant and KV-post cells during KV morphogenesis, live embryos with mosaic labeled KVs were imaged at a specific stage of development (2 ss, 4 ss, 6 ss and 8 ss) and then individual cells at the middle plane of the organ were volume-rendered (*Figure 2B*). We measured the height, length and width of KV cells from several wild-

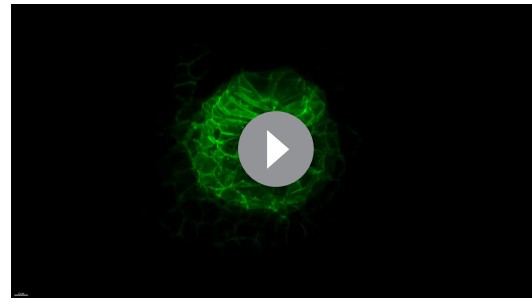

**Video 1.** KV organ architecture in 3D in a live *Tg(sox17: GFP-CAAX)* embryo at 8 ss. The membrane-localized GFP marks all cells in KV. KV is rotating along its anteroposterior (AP) axis. Scale = 20 µm.

DOI: https://doi.org/10.7554/eLife.30963.003

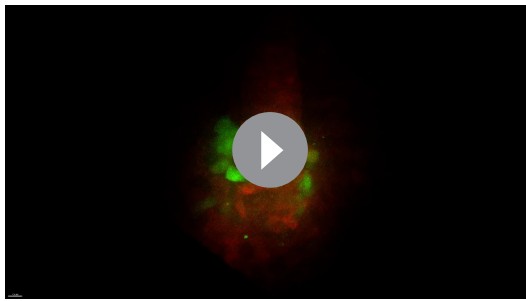

**Video 2.** 3D projection of KV in a live mosaic labeled *Tg(sox17:Cre^ERT2^); Tg(ubi:Zebrabow)* embryo at 8 ss. Stochastic Cre-mediated recombination labels only few cells with YFP expression with clear boundaries that are easily distinguishable from non-recombined RFP⁺ cells. The KV organ is rotating along its anteroposterior (AP) axis. Scale = 20 μm.
DOI: https://doi.org/10.7554/eLife.30963.004

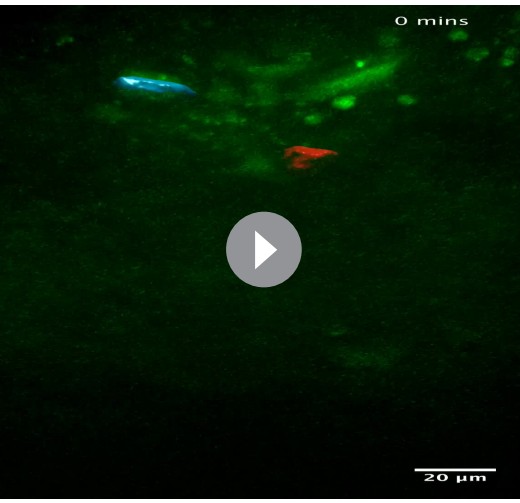

**Video 3.** Time-lapse imaging of KV cells in a live mosaic labeled *Tg(sox17:Cre^ERT2^); Tg(ubi:Zebrabow)* embryo treated with 4-OHT from dome stage to shield stage. Images were collected every 5 min from 2 ss to 8 ss. The movie spans 105 min of development. Single 3D rendered KV-ant (blue) and KV-post (red) cells are followed during KV morphogenesis. Scale = 20 μm.
DOI: https://doi.org/10.7554/eLife.30963.009

type embryos to determine the average parameters for KV-ant and KV-post cells. The results showed cell-to-cell variability, but pooling measurements from multiple cells at a specific developmental stage identified trends and statistically significant differences during KV morphogenesis. First, cell height of both KV-ant cells and KV-post cells increased from 2 ss to 8 ss (*Figure 2C*), reflecting the expansion of the apical surface of cells (see dotted line in *Figure 2B*) to accommodate lumen growth. There were no significant differences in cell height between KV-ant cells and KV-post cells during these stages (*Figure 2C*). Second, cell length decreased in both KV-ant cells and KV-post cells during KV morphogenesis, but a sharp reduction was observed from 4 ss to 6 ss in KV-post cells (*Figure 2D*). This resulted in the cell length of KV-post cells to become significantly different from KV-ant cells after 4 ss. Importantly, it is between 4 ss to 6 ss when cell shape changes associated with KV remodeling were previously observed (*Compagnon et al., 2014*; *Wang et al., 2012*). Third, analysis of cell width indicated KV-ant cells remained relatively constant during KV morphogenesis, whereas KV-post cells showed a distinctive increase in width between 4 ss and 6 ss that resulted in significant differences between KV-ant cells and KV-post cells after 4 ss (*Figure 2E*). In order to make comparisons with previous 2D studies, we calculated length-to-width ratios (LWR) to describe cell morphology. We found that both KV-ant and KV-post cells had similar morphologies at 2 ss and 4 ss and that these morphologies changed after 4 ss such that KV-ant cells had a significantly larger LWR than KV-post cells (*Figure 2F*). This analysis of individual KV-ant and KV-post cell shapes is consistent with asymmetric cell shape changes associated with KV remodeling that were previously identified using 2D measurements of KV cell LWRs (*Wang et al., 2012*) and provides for the first time a 3D quantification of epithelial cell morphodynamics that occur during KV development.

## 3D analysis of single cells reveals asymmetric cell volume changes during KV morphogenesis

We next used mosaic labeling to address whether the volume of individual KV cells changes during development. Since the cells are not regularly shaped, their volume is not simply given by the product of the length, width and height. Measuring the volume of 3D reconstructed KV cells revealed striking dynamics in cell size that mirrored changes in cell shape. At 2 ss and 4 ss, KV-ant and KV-post cells showed similar cell volumes (*Figure 2G*). However, between 4 ss to 6 ss—when KV cells change shape—the volume of KV-ant cells increased and the volume of KV-post cells decreased (*Figure 2G*). These dynamics were visualized in real time by tracking individual KV-ant and KV-post cells continuously for a brief time window (55 min) via time-lapse imaging during the critical period

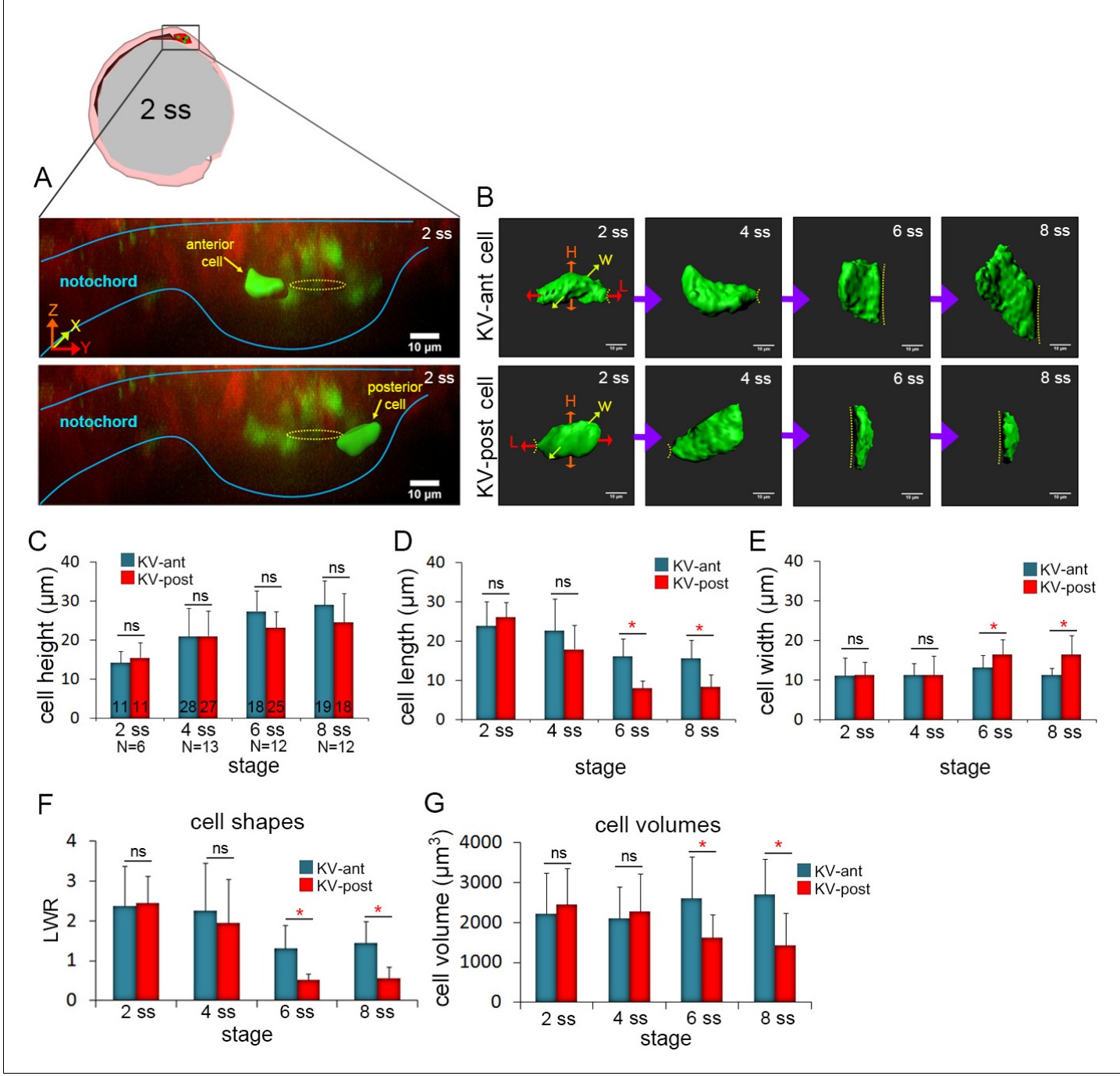

**Figure 2.** 3D morphometric analysis of single cells reveals asymmetric cell volume changes during asymmetric KV cell shape changes. (A) A lateral view of a mosaic labeled KV in a *Tg(sox17:Cre^{ERT2})*; *Tg(ubi:Zebrabow)* at 2 ss. The embryo diagram represents the orientation of the image. The notochord and KV are outlined in blue. Yellow lines mark the KV lumen. Examples of 3D reconstructed KV-ant and KV-post cells along the middle plane of KV are shown. Scale = 10 µm. (B) Representative snapshots of 3D rendered KV-ant and KV-post cells at different stages of KV development between 2 ss and 8 ss. The parameters including height (*H*), length (*L*) and width (*W*) were used to quantify cell morphology. Yellow lines indicate the KV luminal surface. Scale = 10 µm. (C–E) Quantification of height (C), length (D) and width (E) of individual KV-ant and KV-post cells during development. (F) A length-to-width ratio (LWR) was used to describe KV cell shapes. KV-ant and KV-post cells change shape between 4 ss and 6 ss. (G) Volume measurements of individual KV cells at different stages of development. Similar to cell shapes, KV-ant and KV-post cells change volume between 4 ss and 6 ss. All measurements presented in C-G were made on the same group of reconstructed cells. The number of KV-ant and KV-post cells analyzed is indicated in the graph in C. N = number of embryos analyzed at each stage. Graphs show the mean + SD. Results were pooled from three independent experiments. *p<0.01 and ns = not significant (p>5% with Welch's T-Test).

DOI: https://doi.org/10.7554/eLife.30963.005

*Figure 2 continued on next page*

*Figure 2 continued*

The following figure supplements are available for figure 2:

**Figure supplement 1.** Single KV cells were sampled from all positions along the middle plane of KV for morphometric analysis.
DOI: https://doi.org/10.7554/eLife.30963.006
**Figure supplement 2.** Single KV cell dynamics during 4 ss to 6 ss KV stages.
DOI: https://doi.org/10.7554/eLife.30963.007
**Figure supplement 3.** Total changes in KV volume during KV morphogenesis.
DOI: https://doi.org/10.7554/eLife.30963.008

between 4 ss and 6 ss in live mosaic labeled embryos (*Figure 2—figure supplement 2*; *Video 4*). Our results indicate cell volume becomes significantly different between KV-ant and KV-post cells after 4 ss (*Figure 2G*). Thus, differential cell volume changes along the AP axis occur during the same stage as cell shape changes. Overall, between 2 ss and 8 ss, KV-ant cells increased volume from $2106 \pm 1014$ $\mu m^3$ to $2547 \pm 693$ $\mu m^3$ (shown are mean ±one standard deviation) and KV-post cell decreased volume from $2180 \pm 1034$ $\mu m^3$ to $1564 \pm 539$ $\mu m^3$ (*Figure 2G*). The considerable standard deviations here likely reflect the high degree of variability in KV size among wild-type embryos (*Gokey et al., 2016*). It is important to note that while the cell sizes are variable, the direction of size changes during development is constant. KV-ant cells always increase in volume and KV-post cells always decrease in volume. We refer to these regional changes in KV cell size along the AP body axis as asymmetric cell volume changes, which provide new insight into the mechanics of epithelial morphogenesis in KV.

To better understand volume changes during KV development, we measured the volume of the entire KV organ at different stages. To make these measurements in live embryos, *Tg(sox17:GFP-CAAX)* transgenic fish were crossed with *Tg(actb2:myl12.1-MKATE2)* zebrafish that have enriched apical membrane expression of the fluorescent mKate2 protein fused to Myl12.1 (myosin light chain 12 genome duplicate 1) in KV cells. Double transgenic *Tg(actb2:myl12.1mKATE2; Tg(sox17:GFP-CAAX)* embryos were used to visualize both KV lumen and KV cells in living embryos (*Figure 2—figure supplement 3A*). The KV lumen was reconstructed in 3D using *Tg(actb2:myl12.1-MKATE2)* expression and then divided into two equal halves (hence equal volumes) along the AP axis (*Figure 2—figure supplement 3B,B'*). Next, the total KV cellular component of KV-ant and KV-post cells was reconstructed using *Tg(sox17: GFP-CAAX)* expression (*Figure 2—figure supplement 3B,B'*). Volume measurements indicated that both 'total KV-ant cellular volume' and 'total KV-post cellular volume' were similar at 2 ss, but then were significantly different at 8 ss (*Figure 2—figure supplement 3C*). Overall, 'total KV-ant cellular volume' increased from $1.4 \times 10^5$ $\mu m^3$ to $1.7 \times 10^5$ $\mu m^3$ and 'total KV-post cell volume' decreased from $1.2 \times 10^5$ $\mu m^3$ to $0.98 \times 10^5$ $\mu m^3$ between 2 ss and 8 ss stages (*Figure 2—figure supplement 3C*). Thus, consistent with volume changes in single cells at the middle plane of KV, we observed asymmetric volume changes along the anterior and posterior axis of KV when the entire cellular component of the organ was analyzed. These results suggested that asymmetric cell volume changes might contribute to lumen growth and/or cell shape changes during the concurrent processes of lumen expansion and epithelial morphogenesis in KV.

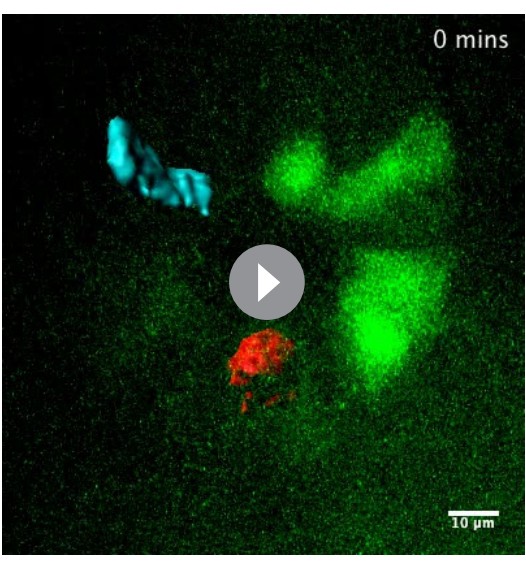

**Video 4.** Time-lapse imaging of KV cells in a live mosaic labeled *Tg(sox17:Cre^{ERT2})*; *Tg(ubi:Zebrabow)* embryo treated with 4-OHT from dome stage to shield stage. Images were collected every 5 min from 4 ss to 6 ss. The movie spans 55 min of development. Single 3D rendered KV-ant (blue) and KV-post (red) cells are followed during KV morphogenesis. Scale = 10 μm.
DOI: https://doi.org/10.7554/eLife.30963.010

## Inhibiting ion flux disrupts asymmetric cell volume changes, lumen expansion and cell shape changes in KV

We next sought to identify a mechanism that mediates KV cell volume changes between 2 ss and 8 ss. At the molecular level, cellular volume can be controlled through coordinated flux of $Na^+$, $K^+$ and $Cl^-$ ions via specific ion channels and pumps that results in osmotically driven water transport (*Wilson et al., 2007*; *Barrett and Keely, 2000*; *Frizzell and Hanrahan, 2012*; *Frizzell, 1995*; *Damkier et al., 2013*; *Spring and Siebens, 1988*; *Hoijman et al., 2015*; *Saias et al., 2015*). To test whether cell volume changes in KV cells are mediated by ion flux, we inhibited either the sodium-potassium pump ($Na^+$/$K^+$-ATPase) or the cystic fibrosis transmembrane conductance regulator (Cftr). Both have previously been shown to play a role in KV lumen expansion (*Navis et al., 2013*; *Compagnon et al., 2014*). We first treated mosaic-labeled (e.g. 4-OHT treated) *Tg(sox17:CreERT2)*; *Tg(ubi:Zebrabow)* embryos with the small molecule ouabain to inhibit the $Na^+$/$K^+$-ATPase pump as previously described (*Compagnon et al., 2014*). The $Na^+$/$K^+$-ATPase has a central role in generating an electrochemical gradient across the plasma membrane that drives transport of water and solutes (*Wilson et al., 2007*; *Damkier et al., 2013*). Ouabain treatments are known to disrupt ion flux, resulting in an increase in intracellular sodium and calcium concentrations. To block $Na^+$/$K^+$-ATPase during the critical stages when cells change volume, we treated mosaic-labeled embryos with ouabain starting at 4 ss (11 hpf) and then imaged KVs at 6 ss (12 hpf). However, these brief treatments did not reduce KV lumen expansion (*Figure 3—figure supplement 1A–C*) as expected from previous work (*Compagnon et al., 2014*). In contrast, embryos treated with ouabain from the bud stage (10 hpf) to 8 ss (13 hpf) did show reduced KV lumen expansion (*Figure 3B,E*) relative to vehicle (DMSO; dimethyl sulfoxide) treated controls (*Figure 3A,E*). This indicated pharmacological treatments from the bud stage are more effective at blocking the lumen expansion process. Similar to wild-type embryos, mosaic labeled KV cells in control embryos treated with DMSO from bud stage to 8 ss underwent normal AP asymmetric changes in cell volume and shape (*Figure 3A*). KV-ant cells increased volume from $2019 \pm 668 \ \mu m^3$ to $2304 \pm 618 \ \mu m^3$ and KV-post cells decreased volume from $1811 \pm 422 \ \mu m^3$ to $1479 \pm 323 \ \mu m^3$ between 2 ss and 8 ss (*Figure 3A*). In contrast, AP asymmetric cell volume changes were not observed in embryos treated with ouabain between bud stage and 8 ss. In ouabain treated embryos both KV-ant and KV-post cells increased volume from $1920 \pm 548 \ \mu m^3$ to $2328 \pm 1050 \ \mu m^3$ and from $1819 \pm 514 \ \mu m^3$ to $2407 \pm 493 \ \mu m^3$ respectively (*Figure 3B*). These results support a model in which ion flux mediates asymmetric cell volume changes during KV morphogenesis.

As a second approach to test the role of ion flux in regulating KV cell volumes, we interfered with Cftr activity. Cftr is an apically localized chloride channel that moves $Cl^-$ ions out of the cell and establishes electrochemical gradients, which drive water into the lumen through osmosis (*Navis and Bagnat, 2015*). Cftr activity can also modulate several other ion-channels and transporters (*Vennekens et al., 1999*), making it a key driver of ion flux and fluid secretion (*Braunstein et al., 2004*; *Valverde et al., 1995*). Mosaic-labeled embryos were treated with the pharmacological compound CFTRinh-172 to inhibit Cftr activity (*Roxo-Rosa et al., 2015*) or a previously characterized antisense *cftr* morpholino oligonucleotide (*cftr* MO) (*Gokey et al., 2016*) to reduce Cftr protein expression. Treating embryos with 30 μM CFTRinh-172 from bud stage to 8 ss reduced KV lumen expansion (*Figure 3C,E*) as expected (*Roxo-Rosa et al., 2015*). To test whether Cftr has a role in modulating KV cell size, we performed a 3D analysis of mosaic labeled single cells. Similar to ouabain treatments, CFTRinh-172 treatments eliminated asymmetric volume changes in KV cells. In contrast to controls (*Figure 3A*), KV-post cells in CFTRinh-172 treated embryos did not lose volume, but rather increased in volume from $1878 \pm 361 \ \mu m^3$ to $2123 \pm 632 \ \mu m^3$. KV-ant cells increased in volume from $1513 \pm 289 \ \mu m^3$ to $1944 \pm 483 \ \mu m^3$ (*Figure 3C*). Reducing Cftr expression by injecting embryos with *cftr* MO had effects that were similar to CFTRinh-172 treatments: KV lumen failed to expand and asymmetric cell volume changes were disrupted (*Figure 3D,E*). In *cftr* MO treated embryos both KV-ant and KV-post cells increased their volume from $1872 \pm 726 \ \mu m^3$ to $2252 \pm 842 \ \mu m^3$ and from $2082 \pm 637 \ \mu m^3$ to $2367 \pm 770 \ \mu m^3$ respectively (*Figure 3D*). This suggests that in both CFTRinh-172 and *cftr* MO treated embryos KV-post cells fail to undergo volume loss due to inhibition of fluid efflux from these cells. Taken together (for statistical power analysis, see *Figure 3—source data 1*), these results indicate ion flux regulated by $Na^+$/$K^+$-ATPase and Cftr activity is a mechanism that drives asymmetric changes in KV cell volumes along the AP axis.

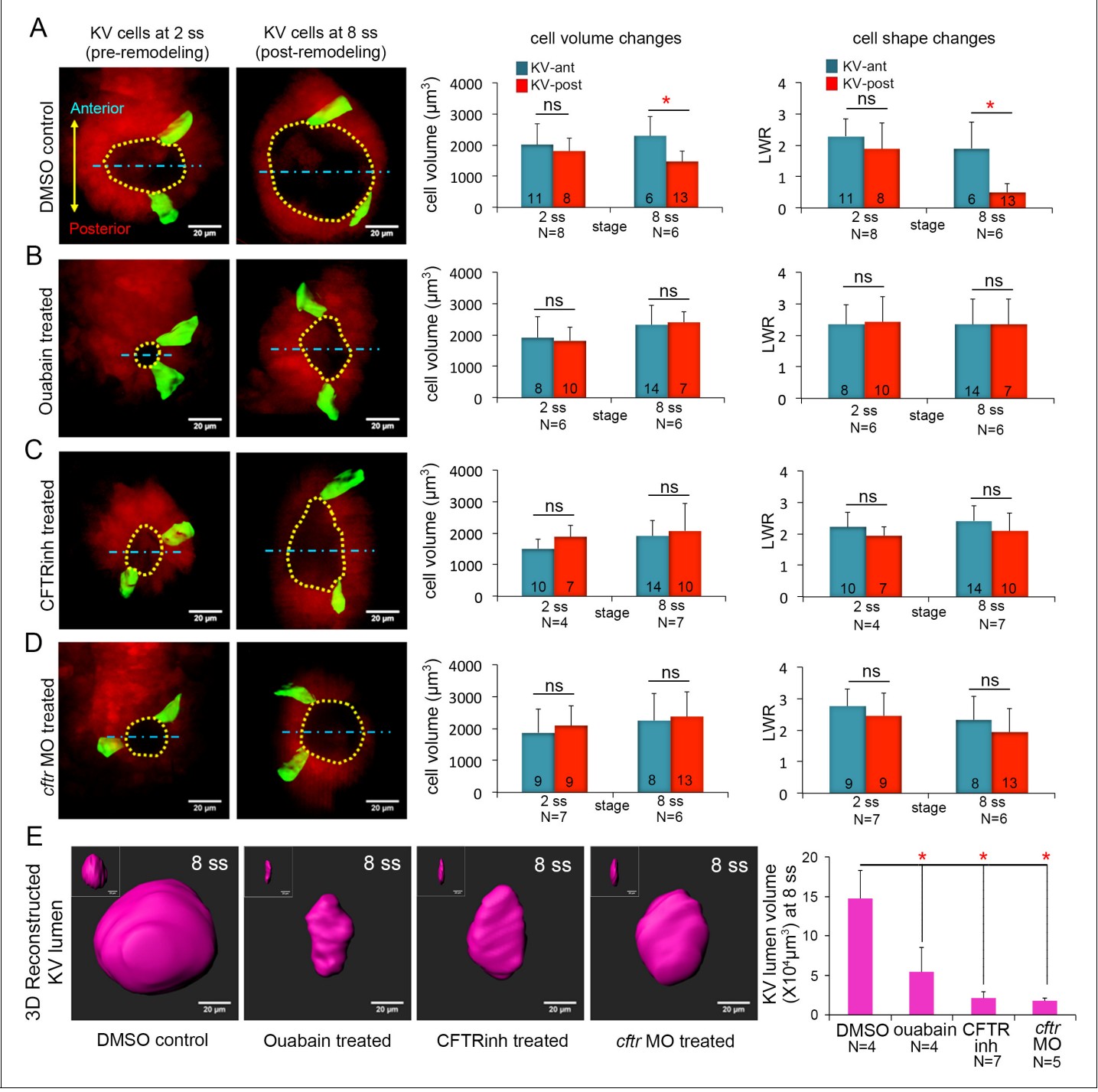

**Figure 3.** Ion channel activity mediates asymmetric KV cell volume changes, KV lumen expansion and KV cell shape changes. (**A**) 3D reconstructed KV-ant and KV-post cells in mosaic labeled *Tg(sox17:Cre^{ERT2})*; *Tg(ubi:Zebrabow)* control embryos (treated with vehicle DMSO) showed asymmetric cell volume changes and asymmetric cell shape changes (length-to-width ratio) between 2 ss and 8 ss. (**B**) Inhibiting the $Na^+/K^+$-ATPase with ouabain treatments reduced KV lumen expansion and disrupted asymmetric cell volume changes. KV cells in ouabain treated embryos did not undergo asymmetric shape changes. (**C–D**) Interfering with Cftr function using the small molecule inhibitor CFTRinh-172 (**C**) or *cftr* MO (**D**) also blocked KV lumenogenesis and disrupted asymmetric cell volume changes and shape changes of KV cells. (**E**) Quantification of 3D reconstructed KV lumen volumes (insets depict lumen in YZ axis) in control and treated live embryos at 8 ss. For all quantitative analyses, the mean + SD is shown. The number of KV-ant and KV-post cells analyzed is indicated in the graphs in A-D. N = number of embryos analyzed. Results were pooled from two independent trials. Scale = 20 μm, *p<0.01 and ns = not significant (p>5% with Welch's T-Test).

DOI: https://doi.org/10.7554/eLife.30963.011

*Figure 3 continued on next page*

*Figure 3 continued*

The following source data and figure supplements are available for figure 3:

**Source data 1.** Statistical power analysis for the non-significant AP cell volume and LWR differences in *Figure 3A–D*.
DOI: https://doi.org/10.7554/eLife.30963.015
**Figure supplement 1.** Ouabain treatments between 4–6 ss do not block KV lumen expansion.
DOI: https://doi.org/10.7554/eLife.30963.012
**Figure supplement 2.** KV cell heights in embryos treated with ion channel inhibitors.
DOI: https://doi.org/10.7554/eLife.30963.013
**Figure supplement 3.** Morphology of external cells surrounding KV in ouabain treated embryos.
DOI: https://doi.org/10.7554/eLife.30963.014

We next evaluated the impact of perturbing ion flux on cell shape changes during KV remodeling. The height (apical surfaces) of KV cells did not increase between 2 ss and 8 ss in embryos treated with inhibitors of $Na^+/K^+$-ATPase or Cftr activity as they did in controls (*Figure 3—figure supplement 2A,B*), which is consistent with a failure of lumen expansion. Analysis of length to width ratios (LWRs) of individual KV-ant and KV-post cells revealed that AP asymmetric cell shape changes observed in control embryos between 2 ss and 8 ss (*Figure 3A*) failed to occur in embryos treated with ouabain or Cftr inhibitors (*Figure 3B–D*). These results indicated that ion flux is necessary for both cell volume changes and cell shape changes during KV remodeling.

Since our ion flux inhibitor treatments were global, we wanted to test whether blocking ion channels altered other tissues in the embryo, including cells surrounding KV that could have an impact on KV cell shapes. Since the effect of loss of Cftr function on KV has already been determined genetically (*Navis et al., 2013*), we focused on the effects of ouabain treatments on external cells. The overall morphology of ouabain treated embryos was similar to controls at 8 ss, except the KV lumen was smaller (*Figure 3—figure supplement 3A,B*), indicating ouabain treatments did not cause severe developmental defects. To analyze cells surrounding KV, we ubiquitously expressed a membrane-localized mCherry (*mCherry-CAAX*) in *Tg(sox17:GFP-CAAX)* embryos that allowed us to simultaneously visualize both KV cells at the middle plane of KV and the surrounding external cells (*Figure 3—figure supplement 3C*). Due to lack of a reference frame (e.g. the lumen surface) to quantitate LWRs of surrounding cells, we used a different parameter called the 'cell shape index, $q$' ($q = [(\text{cell cross-sectional perimeter})/\sqrt{(\text{cell cross-sectional area})}]$) to define surrounding cell morphology (*Bi et al., 2016*). Analysis of external cells with clearly defined boundaries (*mCherry-CAAX* labeling) that were positioned either anterior or posterior of KV indicated that there was no significant difference in cell shapes at 2 ss or 8 ss stages between control or ouabain treated embryos (*Figure 3—figure supplement 3D*). These results indicate that ouabain does not alter the shapes of cells surrounding KV and suggest that defects in KV cell shape changes result from altered ion flux in KV. This is consistent with a previous study (*Compagnon et al., 2014*), in which blocking ion flux suggested forces associated with lumen expansion drive KV remodeling. However, because our 3D analyses showed that altering ion flux disrupts both lumen expansion and KV cell volume changes, it remained unclear whether failed cell shape changes were due to defects in lumen expansion or asymmetric cell volume dynamics or both.

## Mathematical simulations of KV cell shape changes

To begin to tease apart how intrinsic cell size changes and extrinsic lumen expansion forces contribute to asymmetric KV cell shape changes, we developed and simulated a mathematical vertex model for cell shapes in KV. Vertex models, which have been used successfully by our group and others for predicting features of developing tissues (*Fletcher et al., 2014*; *Bi et al., 2015*; *Farhadifar et al., 2007*; *Hufnagel et al., 2007*; *Wang et al., 2012*), represent two-dimensional cross-sections of cells in a tissue as a network of edges and vertices, as shown in *Figure 4A–D*. Adhesion molecules and cytoskeletal machinery generate forces that affect cell shape in different ways. In the vertex model, the balance between these forces is represented by an effective interfacial tension parameter $\Lambda$. Positive interfacial tensions describe cytoskeletal forces that tend to decrease interface lengths, while negative interfacial tensions describe adhesion effects that tend to increase interface lengths. Additionally, cellular volume control is described in the 2D vertex model by a preferred cross sectional

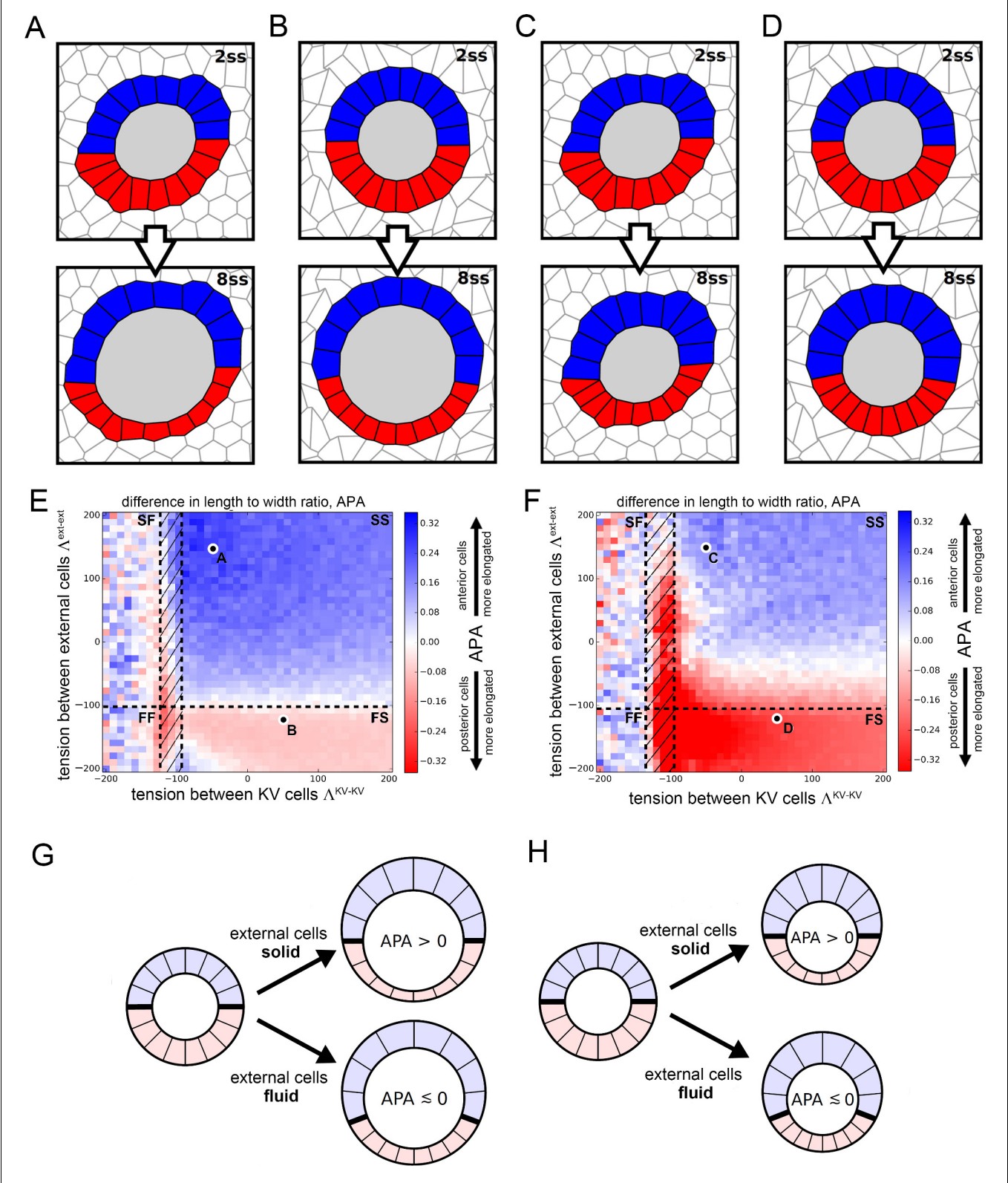

**Figure 4.** Vertex model simulations for cell shapes during KV remodeling. (A–D) Vertex model simulations with N = 10 KV-ant and KV-post cells. Upper and lower panels respectively show force-balanced states at 2 ss and 8 ss. All shown simulations start from the same initial cell positions, but the mechanical parameters differ. The full simulation box is cropped in order to focus on the KV. For the example of panel A, *Figure 4—figure supplement 1B* shows the respective full state. (A) Both KV and external cells are solid-like (interfacial tensions $\Lambda^{KV-KV} = -50$ and $\Lambda^{ext-ext} = 150$), and

*Figure 4 continued on next page*

*Figure 4 continued*

the lumen cross-sectional area expands according to experimental measurements between 2 and 8 ss. (B) KV cells are solid-like ($\Lambda^{\text{KV}-\text{KV}} = 50$), external cells are fluid-like ($\Lambda^{\text{ext}-\text{ext}} = -120$), and the lumen cross-sectional area expands. (C) Both KV and external cells are solid-like ($\Lambda^{\text{KV}-\text{KV}} = -50$ and $\Lambda^{\text{ext}-\text{ext}} = 150$) and the lumen cross-sectional area stays constant between 2 and 8 ss. (D) KV cells are solid-like ($\Lambda^{\text{KV}-\text{KV}} = 50$), external cells are fluid-like ($\Lambda^{\text{ext}-\text{ext}} = -120$), and the lumen cross-sectional area is constant. (E,F) Parameter scan for the anterior-posterior asymmetry, APA (LWR-ant - LWR-post), depending on the respective interfacial tensions of KV cells and external cells, which defines whether these cells are solid-like or fluid-like (FS = external cells fluid-like, KV cells solid-like; SS = both external and KV cells are solid-like; FF = both external and KV cells fluid-like; SF = external cells solid-like, KV cells fluid-like; hatched region = KV ant cells solid-like and KV-post cells fluid-like). For each pair of interfacial tensions, the APA was computed from the average of 100 separate simulation runs. When KV cells are solid-like, the standard error of the mean APA is typically on the order of 0.05. However, for fluid-like KV cells standard error of the mean APA can become much larger, which is reflected by the large mean APA fluctuations in this regime. (E) The lumen cross-sectional area changes normally between 2 and 8 ss. (F) The lumen cross-sectional area is fixed at a constant value between 2 and 8 ss. The parameter values corresponding to panels A-D are marked in E,F. For both E and F, a positive APA is robustly obtained only when KV and external cells are both solid-like. (G,H) Illustrations of how mechanical properties of external cells affect APA values in our simulations. For solid external cells, the interface between KV-ant and KV-post cells is prevented from moving posteriorly upon decreasing KV-post cell cross-sectional areas between 2 ss and 8 . As a consequence, the posterior KV cells flatten and obtain a smaller LWR-post value, which results in a positive APA. Conversely for fluid external cells, a decrease in KV-post cell cross-sectional area is accommodated by a posterior sliding of the interface between KV-ant and KV-post cells. Consequently, the APA does not increase and may even decrease. These mechanisms work both for increasing lumen cross-sectional area (G) and for constant lumen cross-sectional area (H).

DOI: https://doi.org/10.7554/eLife.30963.016

The following source data and figure supplements are available for figure 4:

**Source data 1.** Percentage differences of cell volume, cell cross sectional area, and cell height between anterior and posterior cells [(Post – Ant)/Post] at 2 ss and 8 ss for DMSO control embryos.
DOI: https://doi.org/10.7554/eLife.30963.021
**Source data 2.** Preferred areas $A_0$ prescribed in our vertex model simulations for the different cell types at 2 ss and 8 ss.
DOI: https://doi.org/10.7554/eLife.30963.022
**Figure supplement 1.** Definition of the length-width ratio (LWR) in the simulations, and example for full force-balanced state.
DOI: https://doi.org/10.7554/eLife.30963.017
**Figure supplement 2.** Vertex model simulations for cell shapes during KV remodeling (N = 8).
DOI: https://doi.org/10.7554/eLife.30963.018
**Figure supplement 3.** Vertex model simulations for cell shapes during KV remodeling (N = 12).
DOI: https://doi.org/10.7554/eLife.30963.019
**Figure supplement 4.** Vertex model simulations with asymmetric properties of the external cells.
DOI: https://doi.org/10.7554/eLife.30963.020

area $A_0$ that the cells strive to attain. Deviations of a cell's actual area from its preferred area correspond to cellular pressures. A 'conjugate gradient' computer algorithm was applied to alter the positions of the vertices based on the forces acting on them until a relaxed state is reached where all interfacial tensions are balanced by cellular pressures. Additional details about the model and the computer algorithm can be found in Materials and ethods.

Using the vertex model, we studied whether the observed AP asymmetry in KV cell volume changes could act upstream of the observed AP asymmetry in KV cell shapes as described by aspect ratios. To this end, we focused our modeling efforts on the middle plane of the KV (*Figure 4A–D*). We first measured the cross-sectional areas of KV cells within this plane and found that changes in cell cross-sectional areas correlated with the corresponding cell volume changes (*Figure 4—source data 1*). Thus, to test whether KV cell volume changes can be sufficient to induce changes in KV cell aspect ratios, we prescribed the measured cross-sectional areas for KV lumen and KV cells (*Figure 4—source data 2*) as preferred areas $A_0$ in our model and then studied the induced AP asymmetry of KV cell aspect ratio.

To simulate KV cell shape changes, we initialized a vertex model where a lumen is surrounded by $2N$ adjacent KV cells (split into KV-ant and KV-post cells with $N$ chosen between 8 and 12) and the KV organ is surrounded by 100 'external' cells. To understand how the volume changes between 2 ss and 8 ss affect cell shape, we do not take into account the full time-dependent evolution of the system. Rather, we performed quasi-static simulations consisting of two parts. Initially, preferred area values $A_0$ for lumen, KV-ant cells, and KV-post cells were set equal to the respective 2 ss values reported in the first column of *Figure 4—source data 2*, and the system was relaxed to a force-balanced state using our computer algorithm (upper panels in *Figure 4A–D*). Subsequently, the

preferred area values $A_0$ were changed to their respective values at 8 ss chosen from the second column of *Figure 4—source data 2* and the system was relaxed again (lower panels in *Figure 4A–D*). Such a quasi-static approach is appropriate if in the zebrafish KV, relaxation to mechanical equilibrium is faster than the volume changes of lumen and KV cells. In other developing epithelia, this relaxation timescale has been measured using laser ablation and is on the order of seconds (*Fernandez-Gonzalez et al., 2009*), which is significantly faster than the rate of lumen volume expansion, which is on the order of hours. Note that the images in *Figure 4A–D* are cropped to focus on the KV rather than the surrounding cells (same for *Figure 4—figure supplements 2* and *3A–D*). We show the full system for *Figure 4A* in *Figure 4—figure supplement 1B*.

There are still additional free parameters in the model, corresponding to the interfacial tension values for each cell, but these are not constrained by experimental data. Therefore, we decided to perform a wide parameter sweep to determine how these parameters affect cell shapes. In previous work (*Wang et al., 2012*), we demonstrated that AP asymmetric interfacial tensions were sufficient to drive KV cell shape remodeling even in the absence of asymmetric volume changes. To analyze whether asymmetric area changes alone are sufficient to drive the asymmetric KV cell shape changes, we choose the interfacial tensions between KV cells, $\Lambda^{\mathrm{KV-KV}}$, to be identical between KV-ant and KV-post cells. For simplicity, we assume that all cells external to the KV all have the same interfacial tension $\Lambda^{\mathrm{ext-ext}}$, which is allowed to differ from $\Lambda^{\mathrm{KV-KV}}$. For the purpose of illustration, we show two example simulations for different tension pairs $(\Lambda^{\mathrm{KV-KV}}, \Lambda^{\mathrm{ext-ext}})$ in *Figure 4A,B*. In *Figure 4A* with $\Lambda^{\mathrm{KV-KV}} = -50$ and $\Lambda^{\mathrm{ext-ext}} = 150$, KV-ant cells at 8 ss appear narrow and elongated while KV-post cells appear wide and short, qualitatively reflecting the experimentally observed KV cell shape asymmetry at 8 ss. However, in *Figure 4B* with $\Lambda^{\mathrm{KV-KV}} = 50$ and $\Lambda^{\mathrm{ext-ext}} = -120$, KV-ant and KV-post cells at 8 ss have very similar morphology, suggesting no cell shape change. To quantitatively compare simulations with experimental data, we developed a metric that captures the anterior-posterior asymmetry (APA) of KV cell shapes that is characteristic of KV remodeling. APA is defined as the difference between the length-width ratios (LWRs) of KV-ant and KV-post cells: APA = LWR ant – LWR-post (for the definition of the LWR, see *Figure 4—figure supplement 1A*). Note that because the definition of the APA is based on length-to-width ratios, it is size-independent. In our in vivo measurements, wild-type embryos at 8 ss correspond to an APA value of ~0.9. For the simulation shown in *Figure 4B*, we found an APA value of 0, suggesting no asymmetry, while the simulation in *Figure 4A* has APA value of 0.29, which is clearly asymmetric but not as high as in wild-type experiments.

Because the interfacial tensions $\Lambda^{\mathrm{KV-KV}}$ and $\Lambda^{\mathrm{ext-ext}}$ cannot be determined from our experimental data, an obvious question is whether there is *any* choice of those tension values in our model that would allow area changes alone to drive the observed shape changes. *Figure 4E* is a plot of APA as the interfacial tension in KV cells and external cells are varied. For each $(\Lambda^{\mathrm{KV-KV}}, \Lambda^{\mathrm{ext-ext}})$ parameter pair, the indicated APA value represents an average computed from 100 individual simulation runs. Blue areas indicate a positive APA corresponding to regions where KV-ant cells are more radially elongated than KV-post cells are, while red regions indicate negative APA corresponding to regions with more elongated KV-post cells. A first observation is that the APA is never above 0.31, which is much smaller than wild-type experimental observations. This suggests that changes to cross-sectional area may be an important contribution to shape remodeling, but alone they are not sufficient to generate the observed shape changes.

A second observation is that there are coherent regions in parameter space with similar values of APA, which suggests that our model may be able to identify a simple mechanism for how changes to cross-sectional area drive shape change. It has recently been discovered that as cells increase their interfacial tension $\Lambda$ or increase their preferred area $A_0$, the tissue transitions from fluid-like to solid-like behavior, undergoing a so-called rigidity transition (*Bi et al., 2014*, *2015*; *Park et al., 2015*). Moreover, some of us have recently reported that a similar fluid-solid transition also occurs in bulk three-dimensional tissues (*Merkel and Manning, 2017*). Therefore, we expect that if we can identify a simple mechanism for 2D shape asymmetry that depends on the fluidity of the surrounding tissue, that same mechanism will also be present in 3D.

To do so, we map the results for 2D fluid-solid transitions onto our model, where black dashed lines in *Figure 4E* indicate phase boundaries between solid and fluid. In the upper right quadrant of *Figure 4E,F*, both the KV cells and external cells are solid-like (SS), in the lower right quadrant

external cells are fluid-like while KV cells are solid-like (FS), in the upper left quadrant KV is fluid-like and external tissue is solid-like (SF), and in the lower left quadrant both tissues are fluid-like (FF). There is also a small, hatched region of parameter space where KV-ant cells are solid-like while KV-post cells are fluid-like. To test the significance of our APA values, we computed the standard error of the APA mean for each $\left(\Lambda^{\mathrm{KV-KV}}, \Lambda^{\mathrm{ext-ext}}\right)$ parameter pair. We found that for solid-like KV cells, the error is typically on the order of 0.05, indicating that our results are robust in this regime. Conversely, for fluid-like KV cells, the standard error of the mean can become much larger, which is reflected in the higher APA fluctuations in this regime. They are a direct consequence of softer or floppier KV cells, leading to large fluctuations in their LWRs.

Interestingly, our model predicts that cross-sectional area changes drive the observed shape changes primarily in the solid-solid region, and our understanding of the fluid-solid transition helps us to understand this effect (*Figure 4G*). When the external cells are fluid-like, the KV cells are able to slide past the external cells, so that the interface between the KV-ant and KV-post cells (indicated by a thick black line in *Figure 4G*) moves towards the posterior as the area of the KV-post cells is reduced. In contrast, when the external cells are solid-like, the interface between the KV-ant and KV-post cells is pinned. In this second case, when the KV-post cells lose area they must maintain their lateral width and so the apico-basal extension must decrease. We note that this proposed mechanism works equally well in 3D as in 2D – solid-like external tissue would pin the anterior-posterior interface so that cell volume changes would affect the area of lateral interfaces between KV cells but not the apical area in contact with the lumen. Therefore, in 3D asymmetric cell volume changes would lead to asymmetric cell shape changes and similar APA values to the ones we identified in our 2D model. Although we cannot rule out a more complex model for KV cell shape changes (e.g. with additional parameters characterizing the mechanical heterogeneities in each cell), our simple model suggests that asymmetric cell volume changes contribute to cell shape changes, though additional mechanisms are necessary to explain the very high APA values that are observed in experiments.

The number of epithelial cells in KV can vary in a wild-type population (*Gokey et al., 2016*), therefore we checked the robustness of our result (APA values) with respect to small changes in $N$. In particular, while *Figure 4* shows the simulation results for $N = 10$ KV-ant and KV-post cells, *Figure 4—figure supplements 2–3* show the corresponding results for $N = 8$ and $N = 12$, respectively. In particular, independent of $N$, we observe positive APA only in the regime where both KV and external cells are solid-like. Moreover, there is a general trend of higher APA for a smaller KV cell number. Note that in addition to the mechanism creating the AP cell shape asymmetry illustrated in *Figure 4G*, which works largely independent of $N$, we have also discovered a quite different mechanism, which only works for small $N$ if KV-ant cells are solid and KV-post cells are fluid (for details, see Materials and methods). In this case, KV-post cells are more easily deformed and accommodate the lumen expansion by increasing their apical lumen interface, which leads to flatter KV-post cells and thus a high APA at 8 ss (see *Figure 4—figure supplement 2E*). Note however that this mechanism depends on lumen expansion (compare *Figure 4—figure supplement 2F*).

Another benefit of the model is that we can test specific hypotheses prior to exploring them experimentally. First, to investigate whether lumen expansion is necessary to create an asymmetry in KV cell elongation, we repeated the numerical simulations shown in *Figure 4A,B,E* — which included both asymmetric cell cross-sectional area changes and increase in lumen cross-sectional area between 2 ss and 8 ss — except in this simulation we kept the lumen cross-sectional area fixed (*Figure 4C,D,F*). The APA values, shown in *Figure 4F* are generally smaller (max. APA value of 0.23) than those in *Figure 4E* (max APA value of 0.31). However, most of the regimes where both KV and external cells are solid-like still show positive APA values. These results suggest that in an environment in which cells have solid-like mechanical properties, asymmetric volume changes in KV cells can partially drive asymmetric KV cell shape changes even in the absence of lumen expansion (*Figure 4H*).

Second, using our model, we can explore whether heterogeneous mechanical properties of the external cells can have a significant effect on KV cell shape changes. So far, we have in our model described all external cells using the same parameters. However, given the presence of morphogenetic gradients along the AP axis within the presomitic mesoderm that include FGF and Wnt signals (*Oates et al., 2012*), it is plausible that the mechanical properties of the tailbud cells surrounding the KV may also show an AP-oriented gradient. Moreover, the KV is anteriorly abutting the

notochord with likely different mechanical properties from the tailbud cells (*Zhou et al., 2009*). We thus wondered how our simulation results would depend on such heterogeneities of the external cells. To study this question, we performed simulations similar to that shown in *Figure 4A* where we additionally allowed all external cells on either the posterior side or the anterior side to be fluid-like (*Figure 4—figure supplement 4A and B*, respectively). These fluid-like subsets of the external cells have an interfacial tension of $\Lambda^{\text{ext}-\text{ext}} = -120$ (as in *Figure 4B*). All other parameters were chosen as in *Figure 4A*, which included asymmetric cell cross-sectional area changes and lumen expansion. In particular, the solid-like external cells had $\Lambda^{\text{ext}-\text{ext}} = 150$. We found that AP cell asymmetry was much more pronounced if only the anterior external cells were solid-like (*Figure 4—figure supplement 4A*) than if only the posterior external cells were solid-like (*Figure 4—figure supplement 4B*). For the solid-like anterior external cells, the average APA computed from 100 simulations was 0.39 with a standard error of the mean of 0.03. Thus, solidity of only anterior external cells can be sufficient to introduce an asymmetry in KV cell shapes that can be slightly stronger than if all external cells were solid-like. Conversely, with solid-like external cells only in the posterior region the average APA was 0.07 with a standard error of the mean of 0.04. Thus, if there are solid-like cells present only in the posterior region, the induced AP asymmetry in KV cell shape was much weaker. Hence, asymmetry in the mechanical properties of the cells (and/or material) surrounding the KV can support asymmetric cell shape changes in KV.

## Interfering with junction plakoglobin function inhibits KV lumen expansion

We were intrigued by our modeling results that predicted that given the right environment of surrounding cells, changes in KV cell volumes contribute to changes in KV cell shapes even when the lumen fails to expand. To test this prediction experimentally, we wanted to take an approach that would allow us to monitor cell shape changes in KVs in which ion flux and cell volume changes occur normally but lumen expansion is inhibited. Since coordinated remodeling of adherens junctions between epithelial cells plays important roles during lumen formation (*Alvers et al., 2014*), we chose to interfere with junctions between adjacent KV cells to disrupt lumen growth. The adherens junction component E-cadherin has been linked to cell junction stability and barrier function that maintains lumenal and tubular structures (*Tay et al., 2013*; *Tunggal et al., 2005*). E-cadherin is expressed in KV cells (*Matsui et al., 2011*; *Tay et al., 2013*), but loss of E-cadherin function in mutant embryos leads to early developmental defects during epiboly that preclude analysis of KV formation (*Kane et al., 2005*). We therefore needed tools that allow junctions to form, but with weakened integrity that allows fluid to leak out of the lumen. Transcriptome analysis of zebrafish KV cells (unpublished data) indicated that Junction plakoglobin (Jup; also called γ-catenin) is expressed in KV cells. Jup interacts in complexes at cell-cell adhesions (*Fukunaga et al., 2005*; *Lewis et al., 1997*) and is thought to link cadherins to the cytoskeleton (*Kowalczyk et al., 1998*; *Leonard et al., 2008*; *Holen et al., 2012*). Previous studies in cell cultures indicated Jup plays an essential role in maintaining cell-cell adhesions (*Fang et al., 2014*) and that perturbing Jup function results in increased epithelial permeability (*Nottebaum et al., 2008*). The zebrafish genome contains two *jup* genes, *jupa* and *jupb*. RNA in situ hybridizations confirmed *jupa* expression in KV cells and the population of precursor cells that give rise to KV called dorsal forerunner cells (DFCs) (*Figure 5—figure supplement 1*). Immunofluorescence experiments using Jup antibody (*Martin et al., 2009*) indicated Jupa protein is localized to lateral membranes of KV cells that are marked by GFP expression in *Tg(sox17: GFP-CAAX)* embryos (*Figure 5A*). Thus, we predicted that interfering with Jupa function would perturb KV cell-cell junction integrity such that the KV lumen would fail to expand properly.

To test the function of Jupa in KV morphogenesis we used an antisense morpholino (MO) to block *jupa* pre-mRNA splicing (*jupa* MO-1) or a previously reported MO that blocks translation (*jupa* MO-2) of *jupa* mRNA (*Martin et al., 2009*). Injection of either *jupa* MO efficiently reduced Jupa protein levels at KV cell junctions, as compared to embryos injected with a negative control MO (*Figure 5A*; *Figure 5—figure supplement 2A*). Reduction of Jupa expression was also confirmed using immunoblotting. Jupa antibody detected a prominent band (arrowhead) around 75 kDa—consistent with Jupa proteins (~75–80 kDa) in other vertebrates (*McKoy et al., 2000*)—that was significantly reduced in *jupa* MO injected embryos (*Figure 5B*). Interfering with Jupa expression with MOs did not alter the gross morphology of embryos at 8 ss, but did disrupt KV lumen expansion relative

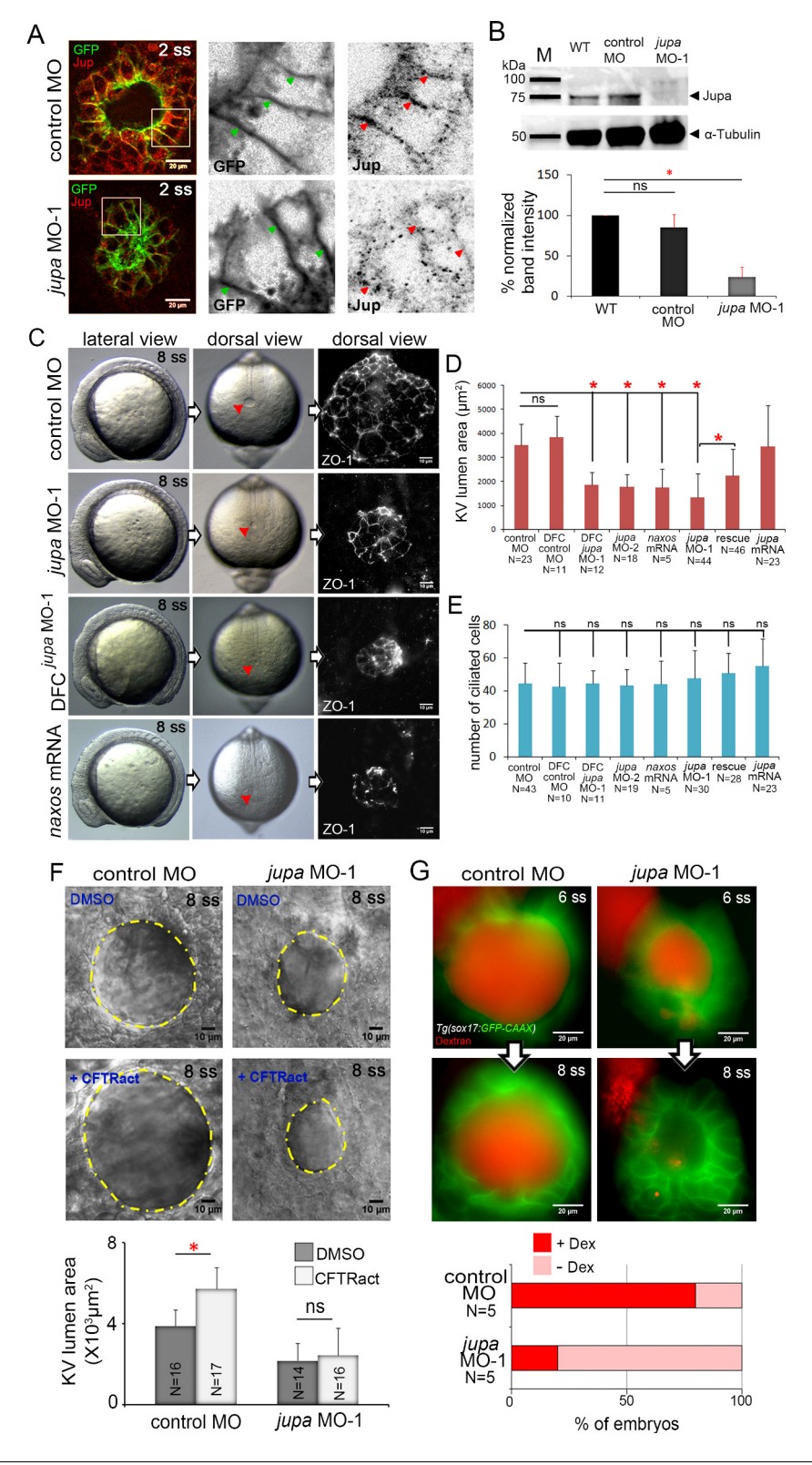

**Figure 5.** Interfering with Junction plakoglobin inhibits KV lumen expansion. (A) Immunostaining with Jup antibodies shows Jup enrichment at lateral membranes of KV cells marked by membrane-targeted GFP expression in *Tg(sox17:GFP-CAAX)* embryos. Embryos injected with *jupa* MO-1 showed reduced Jup protein levels. Boxes indicate enlarged regions shown as individual channels. Arrows point out representative lateral membranes. Scale = 20 μm. (B) Immunoblotting confirmed reduction in Jup protein level (arrowhead) in *jupa* MO-1 injected embryos relative to wild-type (WT) and

*Figure 5 continued on next page*

*Figure 5 continued*

control MO injected embryos. The graph shows normalized Jupa band intensities. Shown is the mean + SD for three independent experiments. (C) At 8 ss, control embryos showed an inflated KV lumen (red arrow) that was labeled using ZO-1 antibody staining. Embryos injected with *jupa* MO-1 to knockdown Jup expression in all cells (global knockdown) or specifically in DFC/KV cells (DFC*jupa MO-1*) appeared normal at 8 ss except that the KV lumen failed to expand. Interfering with Jup by injecting *JUP-naxos* mRNA also inhibited KV lumen expansion. Scale = 10 μm. (D) Quantification of KV lumen area in control and treated embryos at 8 ss. Co-injecting *jupa* MO-1 with *jupa* mRNA significantly rescued lumenogenesis defects. Shown are mean + SD for three independent experiments. (E) The number of ciliated KV cells was not different among the treatment groups. Shown is the mean + SD for results pooled from three independent experiments. (F) Representative images of at 8 ss in control and *jupa* MO injected embryos treated with vehicle (DMSO) or CFTRact-09. The graph shows KV lumen area (outlined by yellow line) in control and treated embryos. Scale = 10 μm. Shown is the mean + SD for two independent experiments.(G) Representative images of KV lumens of contro MO and *jupa* MO embryos injected with rhodamine-dextran. Scale = 20 μm. The graph shows percentage of embryos retaining and losing the fluorescent dye between 6 ss and 8 ss from two independent trials. N = number of embryos analyzed. *p<0.01 and ns = not significant (p>5% with Welch's T-Test).
DOI: https://doi.org/10.7554/eLife.30963.023

The following figure supplements are available for figure 5:

**Figure supplement 1.** *jupa* is maternally supplied and enriched in precursor dorsal forerunner cells (DFCs) and KV.
DOI: https://doi.org/10.7554/eLife.30963.024

**Figure supplement 2.** Interfering with Junction plakoglobin inhibits KV lumenogenesis but not ciliated cell number.
DOI: https://doi.org/10.7554/eLife.30963.025

**Figure supplement 3.** Jupa knockdown results in reduced E-cadherin levels at lateral membranes of KV cells.
DOI: https://doi.org/10.7554/eLife.30963.026

to controls (*Figure 5C,D*) as predicted. ZO-1 immunostaining of apical tight junctions was used to assess the severity of lumen expansion defects *jupa* MO treated embryos (*Figure 5C*). Delivering *jupa* MO specifically to the DFCs (*Amack and Yost, 2004*) that give rise to KV also disrupted lumen expansion (*Figure 5C,D*), indicating Jup functions cell-autonomously during KV morphogenesis. Importantly, the effect on KV lumen expansion caused by MO injection was significantly rescued by co-injection of full-length *jupa* mRNA (*Figure 5C,D*; *Figure 5—figure supplement 2B*), indicating specificity for this phenotype. As a second approach to compromise Jupa function, we injected a human *JUP* mRNA with a mutation that causes naxos disease (*McKoy et al., 2000*). This mutant 'JUP-naxos' mRNA has previously been shown to encode a dominant-negative protein that interferes with Jup function in zebrafish (*Asimaki et al., 2014*). Similar to *jupa* MO treatments, expression of the *JUP*-naxos mRNA reduced KV lumen expansion without inducing other overt defects (*Figure 5C, D*). Importantly, the number of ciliated cells in KV in *jupa* MO and *JUP*-naxos mRNA treated embryos was similar to controls (*Figure 5E*; *Figure 5—figure supplement 2C*), which demonstrates that small KV lumen area was due to reduced lumen expansion rather than a reduced number of KV cells.

We next tested our prediction that loss of Jupa weakens cell-cell adhesions that are necessary for KV lumen expansion. First, we found that reducing Jupa expression moderately reduced E-cadherin enrichment (~22% decrease) along KV cell lateral domains relative to controls (*Figure 5—figure supplement 3*), which is consistent with previous results in cell culture studies (*Fang et al., 2014*). This finding suggested that although E-cadherin is maintained at levels sufficient for epiboly movements and KV formation, the cell-cell adhesions in KV might be weaker in Jupa depleted embryos than in wild-type. To test this functionally, we treated embryos with a small molecule activator of the Cftr channel (CFTRact-09) that increases Cl⁻ ion flux and can over-inflate the KV lumen (*Gokey et al., 2016*). Treatment of control MO embryos with CFTRact-09 significantly increased KV lumen area (~50%) as compared to DMSO, but a similar increase was not observed in Jupa depleted embryos (*Figure 5F*). This suggested that fluids entering the lumen were leaking out through compromised cell-cell junctions. To test this directly, we injected a solution containing fluorescent dextran into the KV lumen. In 4 out of 5 control MO embryos the dextran remained in the lumen over a 1 hr time period. Conversely, in most Jupa depleted embryos the dextran gradually leaked out and only 1 out of 5 embryo retained significant amounts of dye (*Figure 5G*). Together, these results indicate Jupa functions to maintain KV cell-cell adhesion integrity that is critical for KV lumen expansion, and suggest Jupa depletion could provide a useful approach to block lumen expansion without affecting ion flux mediated cell volume changes in KV.

## Asymmetric KV cell shape changes occur independent of lumen expansion

We next used Jupa depleted embryos as a tool to test our hypothesis that KV cell volume changes impact KV cell shape changes in the absence of forces exerted by the process of KV lumen expansion. Morphometric analyses of individual KV cells in mosaic-labeled $Tg(sox17:Cre^{ERT2})$; $Tg(ubi: Zebrabow)$ embryos treated with $jupa$ MO-1 revealed that asymmetric KV cell volume changes occurred between 2 ss and 8 ss in Jupa depleted embryos (KV-ant cells increased volume from $2015 \pm 534\ \mu m^3$ to $2283 \pm 414\ \mu m^3$ and KV-post cells decreased volume from $1839 \pm 612\ \mu m^3$ to $1491 \pm 310\ \mu m^3$ that were similar to controls (KV-ant cells increased volume from $2108 \pm 719\ \mu m^3$ to $2709 \pm 774\ \mu m^3$ and KV-post cells decreased volume from $2273 \pm 864\ \mu m^3$ to $1434 \pm 692\ \mu m^3$) (*Figure 6A,B*). 3D analysis of cell shapes—assessed using LWRs—indicated that even though the lumen failed to expand (*Figure 6D*; *Figure 6—figure supplement 1A*), KV-ant and KV-post cells underwent normal asymmetric cell shape changes in Jupa depleted embryos just as observed in control embryos between 2 ss to 8 ss (*Figure 6A,B*). These results, which are consistent with predictions of the vertex models, provide in vivo evidence that cell shape changes can occur normally during KV remodeling in the absence of KV lumen expansion.

To corroborate results obtained using Jupa depleted embryos, we took a second approach to inhibit lumen expansion by interfering with cell-cell adhesion. We chose to use a previously characterized MO that inhibits expression of the zebrafish Lgl2 (Lethal giant larvae 2) protein (*Tay et al., 2013*). Similar to Jupa depletion, loss of Lgl2 moderately reduces the accumulation of E-cadherin at lateral KV membranes and blocks KV lumen expansion (*Tay et al., 2013*). Analyses of mosaic-labeled KV cells in Lgl2 depleted embryos yielded results that were very similar to Jupa depleted embryos. Lgl2 depletion inhibited KV lumen expansion in mosaic labeled embryos (*Figure 6D*, *Figure 6—figure supplement 1A*), but KV cells completed normal asymmetric volume changes between 2 ss to 8 ss (KV-ant cells increased volume from $2057 \pm 303\ \mu m^3$ to $2329 \pm 847\ \mu m^3$ and KV-post cells decreased volume from $2127 \pm 287\ \mu m^3$ to $1617 \pm 336\ \mu m^3$) and normal asymmetric cell shape changes during KV remodeling (*Figure 6C*). Taken together (for statistical power analysis, see *Figure 6—source data 1*), these results are consistent with Jupa knockdown results and indicate that asymmetric epithelial cell shape changes that sculpt the KV organ are separable from the process of lumen expansion.

## Discussion

The collective behavior of epithelial cells plays a key role in determining the architecture of tissues and organs. Studies of developmental processes in animal models have provided important insights into the biochemical signals and mechanical forces that regulate epithelial morphogenesis (*Quintin et al., 2008*; *Schock and Perrimon, 2002*). The zebrafish Kupffer's vesicle (KV) is a simple organ that provides a useful model system to investigate mechanisms that regulate epithelial cell shape changes in vivo. Using a mosaic labeling approach and 3D morphometric analyses of single KV cells, we identified dynamic epithelial cell volume changes during morphogenesis that are asymmetric along the anteroposterior body axis: KV-ant cells become larger during development, whereas KV-post cells become smaller. Results from experimental perturbations (summarized in *Figure 7*) indicated that interfering with ion flux prevents KV lumen expansion, asymmetric changes in KV cell volume, and asymmetric changes in KV cell shape during KV remodeling. This indicated that KV cell shape changes depend on (1) lumen expansion, (2) KV cell volume changes or (3) both. Results from mathematical simulations (summarized in *Figure 4G–H*) indicate that mechanical properties of external tissues surrounding the KV can impact cell shape changes in the KV, and that when external tissues are solid-like, asymmetric cell volume changes in KV cells contribute to cell shape changes even in the absence of lumen expansion. Experimentally, we found that when we leave ion flux and asymmetric KV cell volume changes intact and only inhibit lumen expansion with leaky KV cell-cell junctions (summarized in *Figure 7*), AP asymmetric KV cell shape changes occur normally. Together, these studies identify asymmetric cell volume regulation as an intrinsic mechanism that guides cell shape changes during epithelial morphogenesis in KV. We propose this is a genetically programmed process that depends on the properties of surrounding cells, but can be separated from the biophysical forces of lumenogenesis.

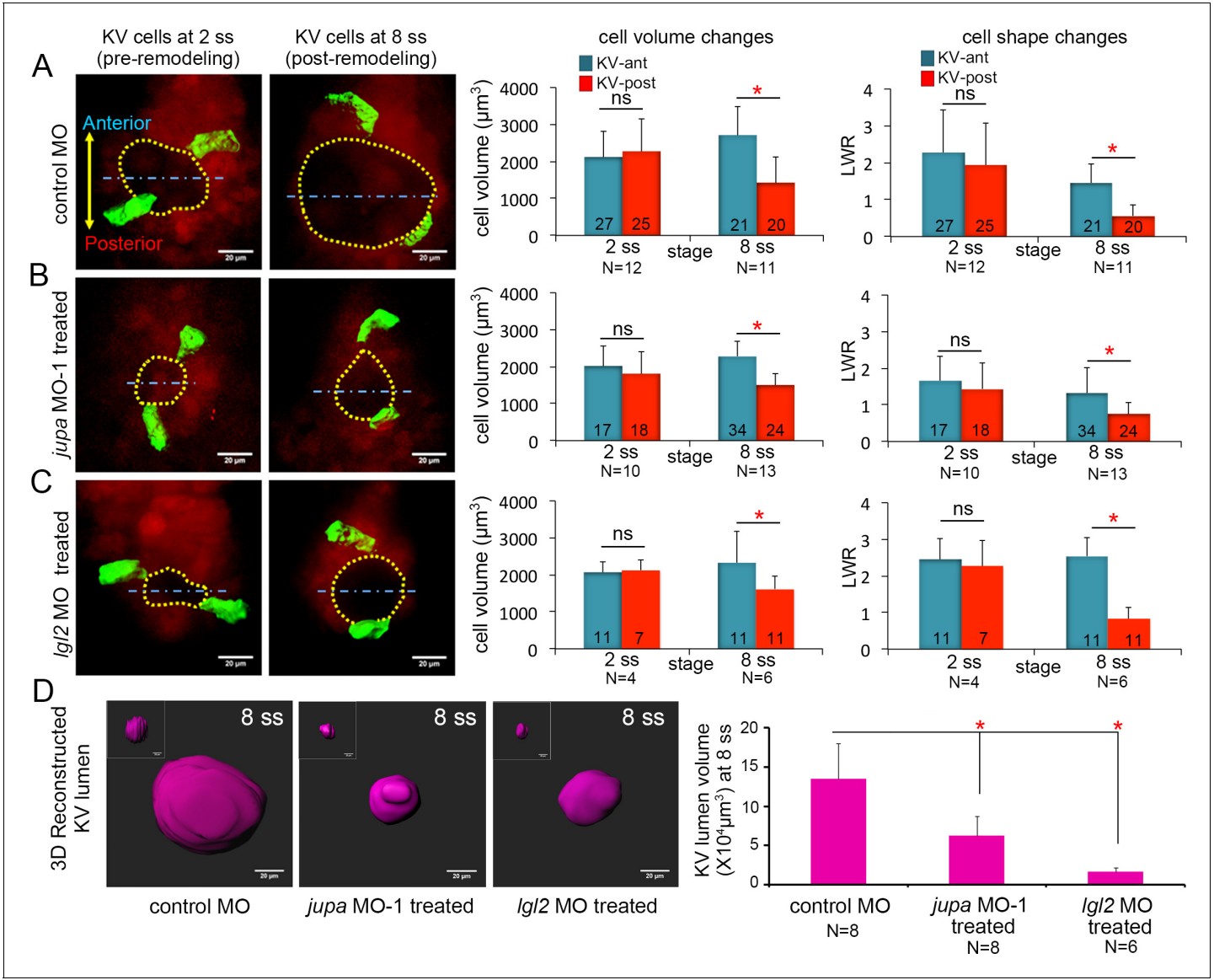

**Figure 6.** Asymmetric cell shape changes in KV are separable from lumen expansion. (A) Mosaic labeled KV cells in control MO injected embryos showed asymmetric changes in cell volumes and cell shapes between at 2 ss and 8 ss. (B–C) Perturbing cell-cell junction integrity in KV by interfering with *jupa* (B) or *lgl2* (C) expression inhibited KV lumen expansion, but asymmetric cell volume changes occurred that were similar to controls. In addition, asymmetric KV cell shape changes occurred normally in *jupa* and *lgl2* MO embryos. (D) Quantification of 3D reconstructed KV lumen volumes (insets depict KV lumen in YZ axis) in control and treated live embryos at 8 ss. For quantitative analyses, the mean + SD is shown. The number of KV-ant and KV-post cells analyzed is indicated in the graphs in A-C. N = number of embryos analyzed. Data for control MO and *jupa* MO experiments are pooled from three independent experiments and *lgl2* MO data are pooled from two experiments. Scale = 20 μm. *p<0.01 and ns = not significant (p>5% with Welch's T-Test).

DOI: https://doi.org/10.7554/eLife.30963.027

The following source data and figure supplement are available for figure 6:

**Source data 1.** Statistical power analysis for the non-significant AP cell volume and LWR differences in *Figure 6A–C*.
DOI: https://doi.org/10.7554/eLife.30963.029

**Figure supplement 1.** KV cell heights in embryos with KV cell-cell adhesion perturbations.
DOI: https://doi.org/10.7554/eLife.30963.028

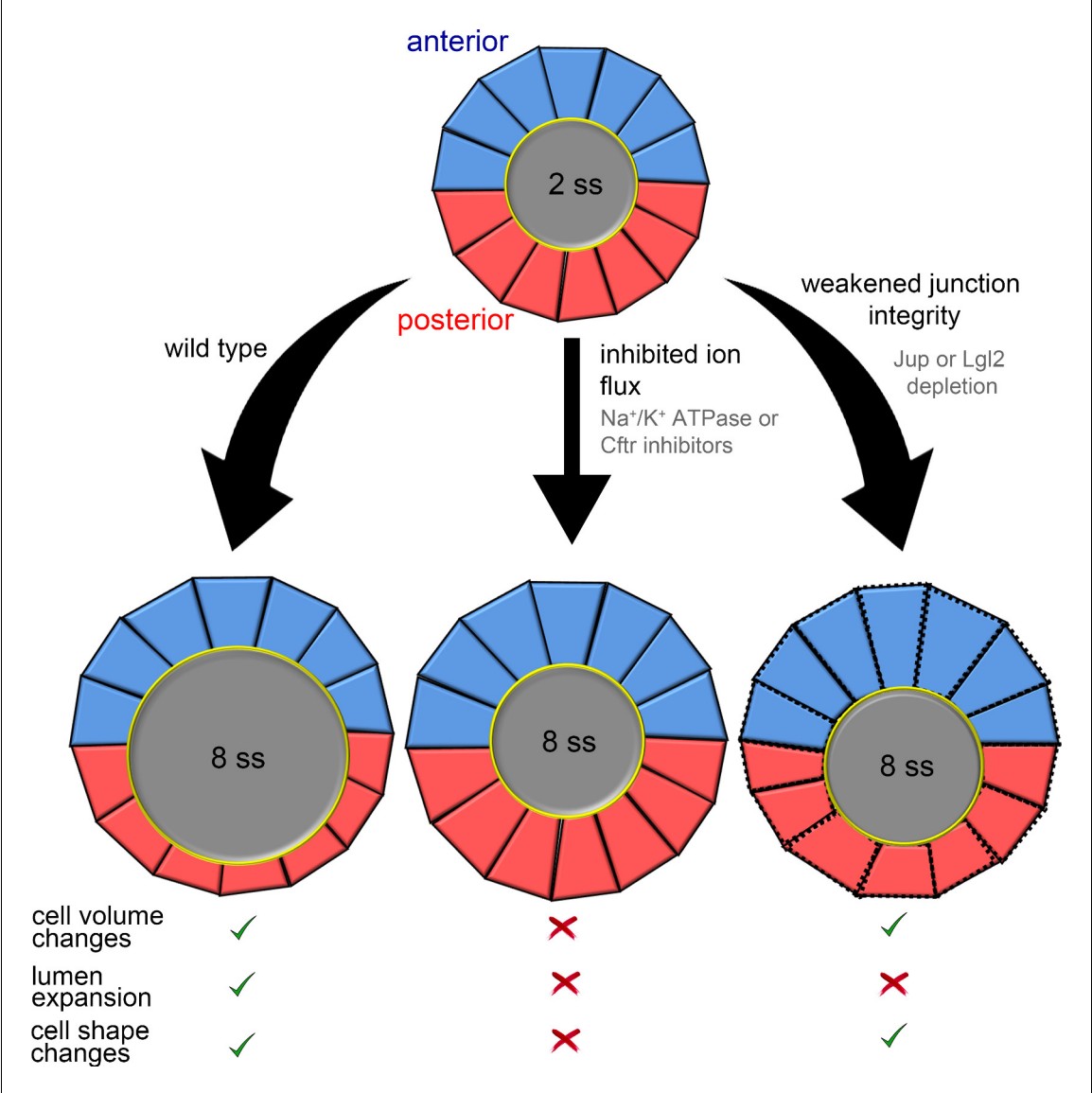

**Figure 7.** Summary and working model for epithelial cell shape changes during KV morphogenesis. Results from experiments and modeling suggest AP asymmetric cell volume changes contribute to asymmetric cell shape changes in the KV epithelium. Inhibiting ion flux blocks asymmetric cell volume changes, lumen expansion and shape changes in KV-ant (blue) and KV-post (red) cells. Vertex simulations predict that asymmetric volume (cross-sectional area) changes in KV cells can introduce AP asymmetry in KV cell shapes without lumen expansion. Consistent with this prediction, asymmetric changes in KV volume and shape occurred in the absence of lumen expansion in embryos with weakened KV cell junction integrity. These results suggest a model in which asymmetric cell volume changes contribute to cell shape changes in KV and that this process is separable from lumen growth.

DOI: https://doi.org/10.7554/eLife.30963.030

## Asymmetric changes in cell size during KV epithelial morphogenesis

The finding that KV cells change volume during development is insightful for thinking about mechanisms of epithelial morphogenesis in KV since previous analyses (*Compagnon et al., 2014*; *Wang et al., 2012*) that were limited to 2D did not predict differences in KV cell size. Our previous analysis of 2D cell cross-sectional area (*Wang et al., 2012*) suggested cells slightly reduce their size during morphogenesis, but did not detect differences between KV-ant and KV-post cells. It is therefore striking that 3D analysis shows that KV cells do indeed change volume, and do so asymmetrically along the AP axis. We recently reported that the size of the KV organ is not under tight control during development, but rather must only exceed a size threshold to function normally during left-right

patterning (*Gokey et al., 2016*). Thus, KV size can vary among wild-type embryos. Consistent with these findings, we observed variable KV cell sizes. However, it is clear that wild-type KV cells always change size in an asymmetric way along the AP axis. Anterior KV cells always increase their size, whereas posterior cells always decrease their size. Interfering with the asymmetry of these size changes by blocking ion flux prevents asymmetric cell shape changes that we know from previous studies (*Compagnon et al., 2014*; *Wang et al., 2012*) are critical for KV function. These results indicate the AP asymmetry of volume changes is important for KV morphogenesis and function.

The decrease in KV-post cell volume is mediated by ion channel activity that regulates fluid movement. Decrease in cell volume has also been observed during morphogenesis of zebrafish otic vesicle (*Hoijman et al., 2015*), where it was suggested that movement of fluids from epithelial cells into the lumen contributes to lumen expansion. It is generally thought that ion flux in epithelial cells sets up a transepithelial flow of fluids from outside the tissue into the lumen (*Gin et al., 2007*; *Frizzell and Hanrahan, 2012*). When ion flux was blocked in KV via Na$^+$/K$^+$-ATPase or Cftr inhibitors between the bud stage and 8 ss, KV-post cells did not shrink (but swelled) and the lumen failed to expand. This finding is consistent with a model in which intraepithelial fluid movement directly from KV-post cells into the lumen promotes lumen expansion. Since the amount of volume lost by KV-post cells does not fully account for the increase in lumen size, we propose ion flux in KV establishes both transepithelial flows and intraepithelial flows from KV-post cells to fill the lumen. Interestingly, brief treatments with the Na$^+$/K$^+$-ATPase inhibitor ouabain between 4–6 ss did not block lumen expansion. This may be because ouabain needs more time to penetrate deep inside the embryo to effectively block ion channel function in KV. Alternatively, these results may suggest ion channel function early in KV development (between bud stage and 4ss) is sufficient for lumen expansion or there are additional mechanisms independent of ion flux that contribute to lumen expansion and cell volume changes.

What makes KV-ant cells behave different from KV-post cells? This asymmetry likely results from a combination of intrinsic and extrinsic factors that differentially regulate KV-ant and KV-post cells. The increase in KV-ant cell size could involve cell growth. Previous studies have uncovered a role for TOR signaling in cell growth (hypertrophy) in non-dividing cells (*Guertin and Sabatini, 2006*). Interestingly, TOR signaling has been implicated in the morphogenesis of KV (*Casar Tena et al., 2015*; *DiBella et al., 2009*; *Yuan et al., 2012*). It will be interesting in future work to test for asymmetric expression/function of TOR pathway components in KV cells. Another possible intrinsic mechanism is that different KV cells develop different mechanical properties. Our previous mathematical models suggest that differential cell-cell interfacial tensions along the AP axis can generate AP asymmetric cell shape changes in KV (*Wang et al., 2012*). Interestingly, a recent study in *Drosophila* showed that contractile force induced cell shape changes are instituted via cell volume reduction (*Saias et al., 2015*), which indicates a link between cell volume regulation and mechanical force generation. In the KV system, it will be interesting to test in future work whether AP asymmetric volume changes result in differential cytoskeletal contractility between KV-ant and KV-post cells. Another possible contributing factor to asymmetric KV cell size is differential activation of ion channels in KV-ant and KV-post cells. For example, it is known that the Cftr localizes to the apical surface of all KV cells at all stages of KV development (*Navis and Bagnat, 2015*). A recent study has uncovered mechanosensitive activation of Cftr in response to membrane stretch (*Zhang et al., 2010*). Stretching the plasma membrane increased ion conductance and also the probability of open Cftr channels at cell membranes. During KV remodeling, apical membrane stretch in KV-post cells may lead to increased Cftr activity and higher ion-efflux with a loss of volume in these cells. An alternative possibility is that different KV cell fates (e.g. KV-ant and KV-post cells) may be determined early in development. By tracking the DFCs that give rise to KV, we have found that these cells maintain their relative spatial positions throughout KV development (*Dasgupta and Amack, 2016*). This suggests subpopulations of KV cells may differentiate early in development and become biochemically distinct during KV morphogenesis. Additional studies are warranted to test the hypothesis of distinct KV-ant and KV-post subpopulations of cells that have differential gene expression and/or ion channel activity.

## The impact of mechanical forces on KV epithelial morphogenesis

Our vertex model simulations suggest that asymmetric volume changes are not alone sufficient to fully induce KV cell shape changes. In addition to cell-intrinsic mechanisms, biophysical forces likely guide the formation of the KV epithelium. These mechanical forces can also arise from extrinsic sources that stem from the mechanical properties of surrounding tissues or extracellular matrix (ECM)

(*Campàs et al., 2014*; *Chanet and Martin, 2014*; *Serwane et al., 2017*; *Etournay et al., 2015*). Localized deposition of the ECM molecules laminin and fibronectin around anterior region of KV has been found to be important for asymmetric cell shape changes during KV morphogenesis (*Compagnon et al., 2014*). Interestingly, our simulations indicated that asymmetric cell shape changes are more pronounced when anterior external cells had solid-like properties and posterior external cells were fluid-like. In this case, the solid-like cells on the anterior side are able to 'pin' the interface between the KV-ant cells and the KV-post cells. An AP gradient of ECM, such as fibronectin (see *Video 5*), may help prevent neighbor exchanges and give rise to solid-like behavior only in anterior external cells. It is also possible that the solid-like ECM directly physically pins the KV-ant cells, so that the mechanism we have identified may operate even if the anterior cells beyond the ECM are fluid-like. Another possibility is that the notochord—which physically interfaces with KV-ant cells—may be a solid-like structure (*Zhou et al., 2009*) and this may provide the pinning mechanism. To further investigate these possibilities, important and technically challenging future work should focus on developing vertex models that interface with models for ECM fiber networks. In addition, fully three-dimensional models for confluent tissues, which have very recently been mechanically characterized (*Merkel and Manning, 2017*) should also be adapted for organogenesis.

Other mechanisms, which remain unexplored, are mechanical forces generated by cells surrounding the KV as it advances towards the tailbud via convergent extension movements. Tissue fluidity in the tailbud plays an important role in controlling body elongation in zebrafish (*Lawton et al., 2013*) and may have an impact on KV. Our mathematical modeling suggests that solid-like behavior of surrounding tissue may play an important role in KV remodeling. In a previous study, we used DFC/KV specific knockdown of the Rho kinase Rock2b to test whether actomyosin contractility in KV cells vs. surrounding cells is involved in KV cell shape changes (*Wang et al., 2012*). KV cell shape changes failed to occur in embryos with Rock2b knocked down in KV cells, even though surrounding cells were normal, indicating that cell-autonomous actomyosin activity is important for KV cell shape changes. However, future studies are needed to explore how the mechanical properties of neighboring cells impact the establishment of asymmetric KV-ant and KV-post cell behaviors.

Lumen expansion occurs synchronously with changes in epithelial cell shapes during KV morphogenesis (*Compagnon et al., 2014*; *Wang et al., 2012*), which raises the possibility that contractile forces and/or intraluminal pressure contributes to KV cell shape changes. This idea is supported by the observation that blocking lumen expansion with ion channel inhibitors prevents KV shape changes (*Compagnon et al., 2014*). However, blocking ion channel activity also disrupts the previously unrecognized KV cell volume changes, making it unclear whether the lack of cell shape changes are due to reduced luminal forces or absence of cell volume changes. Our mathematical models suggest that KV cells can undergo asymmetric cell shape changes even in the absence of forces associated with lumenogenesis. This prediction was experimentally tested by perturbing KV junctions, which allowed us to block lumen expansion without altering ion channel activity or cell volume changes. 3D morphometric analyses revealed KV lumen expansion involves extension of apical surfaces of both KV-ant and KV-post cells in dorsoventral axis (represented here as cell height). But, lateral extension (represented here as cell width) happens only in KV-post cells, not in KV-ant cells due to tight packing and other mechanical influences. Interestingly, lateral extension of KV-post cells can be uncoupled from the dorsoventral extensions, which play a critical role in lumen expansion. Inhibiting lumen expansion by altering junctional integrity hinders dorsoventral expansion of apical surfaces of all KV cells, but KV-post cells still lose their volume via ion channel mediated fluid efflux and undergo lateral extension to facilitate asymmetric KV cell shape changes (*Figure 6*). Thus, ion channels mediate asymmetric cell shape changes via lateral extension of KV-post cells even when

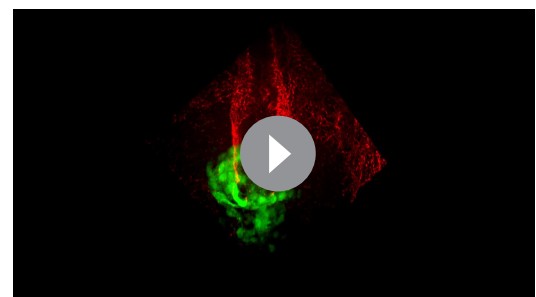

**Video 5.** 3D projection of KV in a fixed *Tg(sox17:GFP)* embryo stained with anti-fibronectin antibody at 8 ss. Fibronectin (red) shows enrichment around the notochord and anterior region of the KV (green).. Anterior = top, Posterior = bottom. KV is rotating along its anteroposterior (AP) axis. Scale = 15 μm
DOI: https://doi.org/10.7554/eLife.30963.031

overall lumen expansion is inhibited. These results provide new mechanistic insight into KV epithelial morphogenesis and suggest a working model in which asymmetric KV cell shape changes depend on intrinsic ion flux-mediated fluid movements and do not depend on extrinsic forces generated by lumen expansion.

We propose that luminal forces have a nominal impact on KV cell shape changes. Previous experimental results support this idea. First, KV cells can fail to change shape even when lumen expansion is normal. Inhibiting Rock2b function or non-muscle myosin II activity had no effect on KV lumen expansion, but prevented cell shape changes during KV remodeling (*Wang et al., 2011*; *Wang et al., 2012*). This indicates that the mechanical forces generated during lumenogenesis are not sufficient to drive KV cell shape changes without active cytoskeletal contractility. Second, the degree of KV lumen expansion is highly variable in a population of wild-type embryos. Correlations between KV lumen size and KV function show that the lumen only needs to exceed a relatively low size threshold for the KV to be functional (*Gokey et al., 2016*). Together, these findings suggest forces exerted by expansion of the lumen play a minor role in cell shape changes during KV epithelial morphogenesis.

### Cell volume changes in epithelial morphogenesis

Cell size is regulated by ion flux, but can also depend on progression through the cell cycle. Thus it is important to note that KV cells are post-mitotic epithelial cells that assemble a cilium (*Amack et al., 2007*). As discussed above, we consider the robust AP asymmetric changes in KV cell size that occurs with precise developmental timing (between 4 ss and 6 ss) as regulated cell volume changes that control KV organ architecture. In the zebrafish KV epithelium, we propose that cell volume changes work in concert with other mechanisms to drive KV remodeling. Other recent studies have also identified links between cell volume changes and epithelial morphogenesis (*Kolahi et al., 2009*; *Saias et al., 2015*; *Hoijman et al., 2015*). However, little is known about the influence of cell volume changes on cell shape regulation. As mentioned previously, during zebrafish otic vesicle development epithelial cells become thinner, suggesting intraepithelial fluid movement contributes to both lumen growth and cell/tissue shape change (*Hoijman et al., 2015*). This finding is consistent with studies in cell culture systems (*Braunstein et al., 2004*; *Vázquez et al., 2001*) that have shown that epithelial cells indeed undergo cellular fluid loss to regulate cell volume and cell shape. In the mouse embryo, a group of nonproliferative epithelial cells in the tooth primordium also decrease their volume and become thinner during tooth budding morphogenesis (*Ahtiainen et al., 2016*). Another recent study in *Drosophila* uncovered that during dorsal closure cells within the amnioserosa lose their volume by ~30% and change their shape (*Saias et al., 2015*). Additionally, in the egg chamber of *Drosophila* the follicle cell epithelium undergoes volume changes during oocyte development to attain distinct cell shapes (*Kolahi et al., 2009*). Taken together with our experimental results and mathematical models in KV, these examples suggest that cell volume change might be a common mechanism that impacts cell shape during epithelial morphogenesis in several tissues and organs.

## Materials and methods

### Key resources table

| Reagent type (species) or resource | Designation | Source or reference | Identifiers | Additional information |
|---|---|---|---|---|
| Gene (zebrafish) | *junction plakoglobin a (jupa), cystic fibrosis transmembrane conductance regulator (cftr), lethal giant larvae homolog 2 (lgl2)* | PMID: 19101534, 23482490, 23487313 | | |
| Strain, strain background (zebrafish) | Zebrafish (Danio rerio). Transgenic lines: this study-Tg(sox17:CreERT2), this study-Tg(sox17:GFP-CAAX), Tg(ubi:Zebrabow), Tg(sox17:GFP), this study-Tg(actb2:myl12.1-MKATE2). strain background (TAB) | PMID: 23757414, 17008449 | | See Materials and methods |

*Continued on next page*

Continued

| Reagent type (species) or resource | Designation | Source or reference | Identifiers | Additional information |
|---|---|---|---|---|
| Genetic reagent (zebrafish) | p5E-sox17, pENTR/D_creERT2, pME-GFP-CAAX, pME-myl12.1-MKATE2, p3E-SV40-polyA, pDest-Tol2CG2, JUP-naxos. | PMID: 22945937, 21138979, 24920660 | | Tol2 Kit V2, See Materials and methods |
| Antibody | anti-junction plakoglobin, anti-ZO-1, anti-E-cadherin, anti-acetylated tubulin, anti-GFP, anti-fibronectin | PMID: 19101534, 23482490, 28315297 | | |
| Commercial assay or kit | Tol2 Kit V2 | http://tol2kit.genetics.utah.edu/index.php/List_of_entry_and_destination_vectors | | |
| Chemical compound, drug | 4-hydroxy tamoxifen (4-OHT), ouabain, CFTRinh-172 (CFTRinh), CFTRact-09 (CFTRact) | PMID: 25535919, 26442502, 26432887 | | |
| Software, algorithm | 2D vertex model | Own code: Ph.D. thesis, Matthias Merkel, Technical University Dresden, 2014 | | |

## Zebrafish husbandry and strains

Zebrafish strains were maintained using standard procedures. Wild-type TAB zebrafish were obtained from the Zebrafish International Resource Center. In addition, the following transgenic zebrafish lines were used: *Tg(ubi:Zebrabow)* (*Pan et al., 2013*), *Tg(sox17:GFP)* (*Sakaguchi et al., 2006*), *Tg(sox17:GFP-CAAX)[sny101]* (this study), *Tg(actb2:myl12.1-MKATE2)[sny102]* (this study), *Tg(sox17:Cre[ERT2])[sny120]* (this study). Embryos were staged as described (*Kimmel et al., 1995*).

## Generation of transgenic lines

Transgene constructs were generated using the Gateway-based Tol2 kit (*Kwan et al., 2007*). To generate *Tg(sox17:Cre[ERT2])* and *Tg(sox17:GFP-CAAX)* transgenics, gateway cloning was performed by combining p5E-*sox17* (a generous gift from Stephanie Woo) (*Woo et al., 2012*), pENTR/D_creERT2 (*Mosimann et al., 2011*) or pME-GFP-CAAX (Tol2 Kit v2), p3E-SV40-polyA (Tol2 Kit v2), and pDest-Tol2CG2 (Tol2 Kit v2) plasmids and LR Clonase II Plus (Invitrogen). Verified constructs (25 ng/μl plasmid DNA) were injected separately with Tol2 Transposase mRNA (~25 ng/μl) into one cell stage TAB zebrafish embryos to generate *Tg(sox17:Cre[ERT2])[sny120]* or *Tg(sox17:GFP-CAAX)[sny101]* $F_0$ founders. Adult $F_0$ animals were then crossed with wild-type fish to generate $F_1$ heterozygotes. *Tg(sox17:Cre[ERT2])[sny120]* fish were then crossed with homozygous *Tg(ubi:Zebrabow)* (*Pan et al., 2013*) animals to generate a double *Tg(sox17:Cre[ERT2]); Tg(ubi:Zebrabow)* transgenic strain. To generate a *Tg(actb2:myl12.1-MKATE2)* transgenic fish, p5E-*actb2* (Tol2 Kit v2), pME-*myl12.1*-MKATE2 (see below), and p3E-SV40-polyA (Tol2 Kit v2) plasmids were recombined into pDestTol2CG4 destination vector as described above. Wild-type TAB embryos were injected with verified constructs and Tol2 Transposase mRNA to generate *Tg(actb2:myl12.1-MKATE2)[sny102]* $F_0$ fish. Adult *Tg(actb2:myl12.1-MKATE2)[sny102]* $F_0$ fish were crossed with wild-type TAB to to generate $F_1$ heterozygotes. *Tg(actb2:myl12.1-MKATE2)* fish were crossed with homozygous *Tg(sox17:GFP-CAAX)* animals to generate a double *Tg(actb2:myl12.1-MKATE2); Tg(sox17:GFP-CAAX)* transgenic strain.

## Generation of pME-myl12.1-MKATE2 construct

The myl12.1 ORF was PCR amplified from cDNA pool generated from 8 ss zebrafish embryos using following primers- myl12.1F: 5′-ATTAATGGATCCATGTCGAGCAAACGCGCCAA-3′ myl12.1R: 5′-ATTAATGAATTCTGCATCGTCTTTGTCTTTGGCTC-3′. The PCR amplified myl12.1 ORF was sub-cloned into pCS2+MKATE2 vector using BamH1 and EcoR1 restriction enzymes to construct pCS2+ myl12.1-MKATE2 plasmid. The myl12.1-MKATE2 construct was PCR amplified from pCS2+ myl12.1-MKATE2 plasmid using following primers- attB1: 5′-GGGGACAAGTTTGTACAAAAAAGCAGGCTATGTCGAGCAAACGCGCCAA-3′ and attB2: 5′-GGGGACCACTTTGTACAAGAAAGCTGGGTTCATCTGTGCCCCAGTTT-3′. The PCR amplified myl12.1-MKATE2 construct was then cloned into

pDONR221 vector using BP recombination to generate the middle entry pME-myl12.1-MKATE2 vector.

## Whole-mount in situ RNA hybridization

A plasmid encoding full-length *jupa* was kindly provided by Maura Grealy's lab (NUI, Galway) (*Martin et al., 2009*). It was subcloned into a pCS2+ vector and PCR amplified using following primers: jupaL- 5′-GGCTGGCCCTGTGTCCAGCC-3′ and jupaR- 5′-GTAGCCATCAAGCTCTTCAT-3′. The amplicon was TA cloned into pCRII TOPO vector and used to generate sense and antisense mRNA probes (DIG RNA labeling kit, Sigma) to detect *jupa* expression by in situ hybridization. RNA in situ hybridizations were performed as described (*Wang et al., 2011*).

## Embryo injections

Morpholino oligonucleotides (MOs) were obtained from Gene Tools, LLC (Philomath, OR). We designed *jupa* MO-1 (5′-TTATGATTGTGTCTTCTCACCTGCA-3′) to interfere with *jupa* pre-mRNA splicing of exons 2 and 3. *jupa* MO-2 (5′-GAGCCTCTCCCATGTGCATTTCCAT-3′) designed to block *jupa* mRNA translation was previously described (*Martin et al., 2009*). Other previously characterized MOs used in this study were *cftr* MO (5′-CACAGGTGATCTCTGCATCCTAAA-3′) (*Gokey et al., 2016*), *lgl2* MO-1 (5′-GCCCATGACGCCTGAACCTCTTCAT-3′) (*Tay et al., 2013*) and a standard negative control MO (5′-CCTCTTACCTCAGTTACAATTTATA-3′) (Gene Tools). MOs were injected into wild-type TAB embryos between the 1- and 2 cell stages. Dose curves were performed to determine optimal MO treatments: 2.5 ng of *jupa* MO-1, 2.5 ng of *jupa* MO-2, 1 ng of *cftr* MO (*Gokey et al., 2016*), 4.4 ng of *lgl2* MO-1 (*Tay et al., 2013*) and 2.5 ng of control MO. All MOs were co-injected with 4 ng p53 MO (5′-GCGCCATTGCTTTGCAAGAATTG-3′) to diminish off-target effects as described (*Tay et al., 2013*). To conduct rescue experiments, pCS2+ vector containing full-length *jupa* was digested with NotI restriction enzyme and the linearized plasmid was used as a template to synthesize capped *jupa* mRNA using SP6 mMessage mMachine kit (ThermoFisher Scientific, Waltham, MA. For Jup rescue experiments, *jupa* MO-1 was co-injected with 75 pg *jupa* mRNA. A construct that encodes a dominant negative JUP-naxos protein (*Asimaki et al., 2014*) was a kind gift from the Saffitz Lab. To over-express the JUP-naxos protein, 120 pg of *JUP-naxos* mRNA was injected into 1 cell stage wild-type TAB embryos.

## Fluorescent dextran injections into KV

Control MO or *jupa* MO-1 treated *Tg(sox17:GFP-CAAX)* embryos were dechorionated and mounted in 1% low melting agarose at 6 ss. KV lumens were microinjected with ~1 nL of 10 kDa dextran, Alexa Fluor-568 (Molecular Probes; Eugene, OR, Lot: 1120095) at 6 ss and imaged using a Zeiss Imager M1 microscope immediately (at 6 ss). Successfully injected embryos were then incubated at 28.5 degrees for one hour and then imaged again at 8 ss.

## Immunofluorescence and microscopy

For whole mount immunofluorescent staining experiments, embryos were fixed in 4% paraformaldehyde in 1X PBS with 0.5% Triton X-100 at 4°C overnight and then dechorionated in 1X PBS. Embryos were permeabilized in blocking solution containing 1X PBS, 0.1% Triton X-100, 0.1%DMSO, and 5% goat serum for 4 hr. Primary antibodies were diluted in fresh blocking solution and incubated with embryos at 4°C overnight. Primary antibodies used: mouse anti-junction plakoglobin (1:200, BD Transduction Laboratories, San Jose, CA), mouse anti-ZO-1 (1:200, Invitrogen, Carlsbad, CA), mouse anti-E-cadherin (1:200, BD Transduction Laboratories), mouse anti-acetylated tubulin (1:200, Sigma, St. Loius, MO), mouse anti-GFP (1:200, Molecular Probes), rabbit anti-GFP (1:200, Molecular Probes) and rabbit anti-fibronectin (1:200 Sigma, F3648). Embryos were then washed in 1X PBS with 0.1% Triton X-100, 0.1% DMSO, and 1% BSA at room temperature. AlexaFluor 488- and 568- conjugated anti-rabbit and anti-mouse secondary antibodies (Invitrogen, Molecular Probes) were used at 1:200 dilutions in blocking solution overnight. Stained embryos were then washed in 1X PBS with 0.1% Triton X-100, 0.1% DMSO, 1% BSA at room temperature. Embryos were imaged using either Zeiss Imager M1 microscope or a Perkin-Elmer UltraVIEW Vox spinning disk confocal microscope. Quantification based on fluorescent immunostaining was performed using ImageJ software. KV lumen areas were measured using maximum projections of ZO-1 staining. E-cadherin levels at KV

cell junctions were measured by determining the mean gray level (per pixel) along KV cell lateral membranes as described (*Tay et al., 2013*). This mean gray level (fluorescence intensity) was normalized to GFP intensity along lateral membranes of KV cells.

## Immunoblotting

Protein extracts from approximately 30 zebrafish embryos at 8 ss were prepared as described (*Martin et al., 2009*). 30 µL of 2X SDS sample buffer was added and samples were boiled for 5 min. Extract from approximately 10 embryos was loaded into each lane of commercially prepared 12% gels (Bio-Rad laboratories, Hercules, CA, 456–1044) and ran at 100 V for 2 hr. Semi-dry transfers were performed at 15 V for 45 min. onto a nitrocellulose membrane (Millipore, Billerica, MA HATF00010). Membranes were blocked in blocking solution (3% BSA, 100 mM NaCl, 20 mM Tris with pH 7.6, 0.2% Tween-20 in distilled water) over night at 4°C. Membranes were cut and anti-Jup (BD Transduction Laboratories) and anti-alpha tubulin antibodies (Sigma T-6199) were used at 1:1000 dilutions in primary antibody block (0.3% BSA and tris-buffer saline with Tween-20 or TBST) and incubated at 4°C over night. Membranes were washed 4 × 15 min in TBST. Anti-mouse (Bio-Rad laboratories 166–2408) secondary antibodies were used at a 1:10,000 dilution in TBST for 2 hr at room temperature. After 4 washes for 15 min in TBST (10 mM Tris with pH 8, 150 mM NaCl, 0.05% Tween-20 in distilled water) membranes were incubated in ECL (Bio-Rad laboratories 170–5060) for 1 min and imaged on a ChemiDoc MP (Bio-Rad laboratories) imager. Band intensities were quantified using ImageJ software.

## Pharmacological treatments

To induce low levels of Cre recombinase activity in *Tg(sox17:Cre^{ERT2}; Tg(ubi:Zebrabow)* double transgenic embryos, these embryos were treated with a working concentration of 5 µM 4-hydroxy tamoxifen (Sigma) in 0.1% DMSO from the dome stage to the shield stage. To inhibit ion transport, embryos were either treated with a working concentration of 1 mM ouabain (Sigma) dissolved in water or 30 µM CFTRinh-172 (Tocris, Catalog No. 3430) in 0.1% DMSO from the bud stage to 2 ss or 8 ss. To activate Cftr channels, control MO and *jupa* MO injected embryos were treated with a working concentration of 10 µM CFTRact-09 (Chem Bridge, San Diego, CA) from the bud stage to 8 ss. 0.1% DMSO was used as a vehicle control for all experiments. After pharmacological treatments, embryos were thoroughly washed with embryo medium, mounted in 1% low melting agarose and imaged using either a Perkin-Elmer UltraVIEW Vox spinning disk confocal microscope or a Zeiss Imager M1 microscope.

## Live imaging and morphometric analysis of KV cells

To image live KV cells, embryos were dechorionated and mounted in 1% low-melting point agarose on a glass-bottom MetTek dish at specific stages. Time-lapse imaging of KV was performed using 2 µm step-scan captured at 5 min. intervals for 105 min. using a Perkin-Elmer UltraVIEW Vox spinning disk confocal microscope. The acquired 3D datasets were processed and volume rendered using surface evolver tool in Imaris (Bitplane, Belfast, UK). Imaris was used to measure the length, width and height of reconstructed KV cells. The surface of the lumen was used to establish the axes of KV cells such that lateral axis (cell width) is parallel to the tangent of the curved lumen surface. To measure KV-lumen, KV-ant and KV-post cell cross-sectional areas, captured 3D images were oriented and maximum cross-sectional area from the middle plane perpendicular to the DV axis of individual cells were measured using clipping plane function in Imaris (Bitplane).

## KV cell volume measurements

Single mosaic labeled cells (YFP⁺) in the KV was 3D reconstructed using 'Create Surface' tool in Imaris (Bitplane) software. From 3D reconstructed cells the 'cell volume' was measured. To measure total KV cellular volume, double transgenic *Tg(actb2:myl12.1mKATE2); Tg(sox17:GFP-CAAX)* embryos were used to 3D reconstruct the KV lumen and the total KV cellular component. The 3D lumen was split into equal anterior and posterior halves and the cellular component associated with the two halves of the lumen were defined as the 'total KV-ant cellular volume' and 'total KV-post cellular volume.'

## Analysis of cells external to the KV

55 pg of mRNA encoding a membrane-targeted mCherry (*mCherry-CAAX*) was injected into *Tg (sox17:GFP-CAAX)* embryos at the 1 cell stage. Confocal images were captured from live embryos at 2 ss and 8 ss. The cell shape index, $q = $ [(cell cross-sectional perimeter)/$\sqrt{}$(cell cross-sectional area)] (*Bi et al., 2016*) was used to define morphology of cells surrounding KV at the middle plane of the KV organ in control and ouabain treated embryos at 2 ss and 8 ss. On average, 5 cells were measured from the anterior and posterior regions per embryo.

## Vertex model simulation of KV

We simulate KV morphogenesis using the Vertex Model with periodic boundary conditions (*Bi et al., 2015*; *Farhadifar et al., 2007*; *Fletcher et al., 2014*; *Hufnagel et al., 2007*). Because fully 3D models introduce a larger number of variables and parameters and were not until very recently well-characterized (*Merkel and Manning, 2017*) we choose a two-dimensional description where each cell $i$ is represented as a polygon with area $A_i$ and perimeter $P_i$. We focus our description on the plane through the center of the KV perpendicular to the dorso-ventral axis, and represent lumen, KV-ant cells, KV-post cells, and cells external to the KV as different cell types, which may differ in their mechanical properties. We choose to have an equal number $N$ of KV-ant and KV-post cells, respectively, and 100 external cells. Force-balanced states are defined by minima of the following effective energy functional

$$E = \frac{1}{2}\sum_i \left[ K_A(A_i - A_0)^2 + K_P P_i^2 \right] + \sum_{\langle ij \rangle,\, i<j} \Lambda_{ij} l_{ij} \,. \tag{1}$$

Here, the first sum is over all cells. The first term in it describes a cell area elasticity, where $K_A$ is the associated spring constant and $A_0$ is the preferred area. The second term in the first sum describes cell perimeter elasticity, where $K_P$ is the associated spring constant. The second sum in *Equation (1)* is over all interfaces $\langle ij \rangle$ between adjacent cells $i$ and $j$. It accounts for the interfacial tensions between cells, where $\Lambda_{ij}$ denotes the interfacial tension between cells $i$ and $j$ and $l_{ij}$ denotes the interface length. Note that in order to facilitate comparison with experimental data, we choose micrometers as length units for our vertex model simulations.

For our simulations, we choose the values of $A_0$ displayed in *Figure 4—source data 2*. The listed values for lumen and KV cells are experimentally measured average cross-sectional areas (see Materials and methods). We assumed the external cells to be about as big as the KV cells, so we set the value of the external cells at 2 ss to the average of KV-ant and KV-post cells. The total preferred area is computed as the total sum of the preferred areas $A_0$ of all cells at 2 ss. The preferred area of the external cells at 8 ss is chosen such that the total preferred area stays constant between 2 ss and 8 ss, which corresponds to only a small preferred area change of these cells (*Figure 4—source data 2*). We set $K_A = 1000$ and $K_P = 1$ for all cell types. We have chosen a very high ratio $(K_A L^2)/K_P$ with $L$ being a typical cell diameter in order to ensure that the measured cross-sectional area values in *Figure 4—source data 2* are largely fulfilled by the cells.

The values for the line tensions depend on both involved cell types. We have set the line tension between any two KV cells $i$ and $j$ to the same value $\Lambda_{ij} = \Lambda^{\mathrm{KV-KV}}$, independent of whether the KV cells are anterior or posterior cells. Similarly, the line tension between two external cells $i$ and $j$ is set to $\Lambda_{ij} = \Lambda^{\mathrm{ext-ext}}$. Since we have no measured values for these interfacial tensions, we vary both interfacial tension parameters, $\Lambda^{\mathrm{KV-KV}}$ and $\Lambda^{\mathrm{ext-ext}}$, in *Figure 4*; *Figure 4—figure supplements 2* and *3*. The interfacial tension between a KV cell $i$ and an external cell $j$ is set to the average of both homotypic interfacial tensions with an additional offset: $\Lambda_{ij} = \Lambda^{\mathrm{KV-ext}} = \left(\Lambda^{\mathrm{KV-KV}} + \Lambda^{\mathrm{ext-ext}}\right)/2 + 200$. The tension offset serves to prevent KV cells from being extruded from the KV epithelium and to allow for a smoother basal interface between KV and external cells. Between KV cells and lumen, the interfacial tension is set to a positive value of $\Lambda_{ij}^{\mathrm{lumen-KV}} = 100$ to ensure that the lumen surface is roughly spherical.

The system is initialized using the Voronoi tessellation of a pattern of cell positions. The $2N$ KV cell positions are arranged equidistantly on a circle around the central lumen 'cell' position. The radius of this circle is computed as the estimated lumen radius plus half of the estimated KV cell height. The positions for the 100 external cells are drawn randomly from a uniform distribution with

the condition of having at least a distance of lumen radius plus estimated KV cell height from the lumen cell position. Then, preferred cell areas are set to their 2 ss values and the system is relaxed by minimizing the energy functional. Afterwards, the preferred areas are set to their respective 8 ss values and the system is relaxed again. Note that the Voronoi tessellation is only used to facilitate the initialization. The subsequent energy minimizations are carried out using varying vertex positions. Also note that the dimensions of the periodic box were also allowed to vary during the minimizations. We use the conjugated gradient algorithm from the GSL library (https://www.gnu.org/software/gsl/) for the energy minimization (*Press, 2007*).

## Computation of the KV cell length-width ratio (LWR) in the simulations

We compute the LWR of a given KV cell as the quotient of its length L divided by its width W (*Figure 4—figure supplement 1*). We define the width W as the distance between the midpoints of the respective interfaces with the two adjacent KV cells. The length L is defined as the distance from the midpoint of the interface with the lumen to the midpoint between points P and Q, which are the respective endpoints of the interfaces with the adjacent KV cells. In *Figure 4E–F* we plot the respective average LWR. During the energy minimizations, KV cells occasionally lose contact with the lumen. For the averaging, we thus only take into account the KV cells that are in still contact with the lumen.

## Definition of separation of solid from fluid regimes

Earlier work on the vertex model suggested that a shape index computed from cell perimeter and area can be used to differentiate between solid and fluid regime (*Bi et al., 2015*). However, these simulations only studied vertex model tissues with a single cell type randomly arranged, while in our simulations, there are several cell types and a very distinct geometrical arrangement. Thus, to differentiate solid cell from fluid ones, we choose a different measurement, which is based on the actual cell perimeter $P_i$ and the interfacial tensions $\Lambda$. Based on the interfacial tensions, one can define another parameter, which characterizes a preferred perimeter $P_0 = -\Lambda/(2K_P)$. It has been observed that fluidity also correlates with the difference between actual and preferred perimeter $P_i - P_0$ (*Bi et al., 2015*). Solid vertex model tissues have $P_i - P_0 > 0$ while fluid vertex model tissue has $P_i - P_0 = 0$. Correspondingly, we use this criterion to differentiate between solid and fluid cells to define the positions of the dashed black lines in *Figure 4E,F* and in *Figure 4—figure supplement 2E,F* and, *Figure 4—figure supplement 3E,F*. Note that as a consequence, the positions of these lines slightly vary for different conditions.

## Case of solid anterior and fluid posterior KV cells

In our vertex model we have discovered a second mechanism that can lead to a positive APA for small $N$, which is different from the mechanism illustrated in *Figure 4G*. This mechanism is at work for instance in the hatched region in *Figure 4—figure supplement 2E*, where anterior KV cells are solid-like and posterior KV cells are fluid-like. This difference arises even though both anterior and posterior KV cells have the same interfacial tension $\Lambda^{\mathrm{KV-KV}}$, because anterior KV cells have a much higher preferred area at 8ss than posterior KV cells (*Figure 4—source data 2*). Because the condition of fluidity in the vertex model depends on both preferred area and interfacial tension (*Bi et al., 2015*), there is an intermediate regime when increasing $\Lambda^{\mathrm{KV-KV}}$ where anterior KV cells are already solid, but posterior KV cells are still fluid. When the lumen area increases in this regime, both anterior and posterior cells together have to accommodate a larger total apical interface with the lumen. However, because the anterior cells are solid while the posterior cells are fluid, the latter are more easily stretched laterally. This induces an asymmetry in cell shape that corresponds to a positive APA. Note that the effect of this mechanism appears to extend further into the region where also the posterior KV cells are solid, likely because close to the hatched region they are still more easily deformable than the anterior KV cells.

## Simulations with asymmetric properties of the external cells

To simulate asymmetric properties of external cells (see *Figure 4—figure supplement 4*), we proceeded as before with the following changes. We divide all external cells into an anterior and a posterior subset based on the randomly drawn initial Voronoi cell positions. If the initial Voronoi

position of a cell is anterior (posterior) of the initial lumen Voronoi position, we regard it as one of the anterior (posterior) external cells. The interfacial tensions between two anterior (posterior) external cells are defined by the parameter $\Lambda^{\text{ext,A}-\text{ext,A}}$ ($\Lambda^{\text{ext,P}-\text{ext,P}}$). To set the anterior cells solid (fluid) and the posterior cells fluid (solid), we choose the parameter values $\Lambda^{\text{ext,A}-\text{ext,A}} = 150$ and $\Lambda^{\text{ext,P}-\text{ext,P}} = -120$ ($\Lambda^{\text{ext,A}-\text{ext,A}} = -120$ and $\Lambda^{\text{ext,P}-\text{ext,P}} = 150$). The interfacial tension between an anterior and a posterior external cell was set to the average: $\Lambda^{\text{ext,A}-\text{ext,P}} = \left(\Lambda^{\text{ext,A}-\text{ext,A}} + \Lambda^{\text{ext,P}-\text{ext,P}}\right)/2$.

### Percentage cell volume, cell cross-sectional area and cell height change quantifications

Percentage changes were measured using the following method: If we consider, cellular properties at 8 ss = y with standard deviation δy and cellular properties at 2 ss = x with standard deviation δx, then '% change (z)' = {(y − x)/x}*100. The standard deviations can be used as the uncertainty in the measured values. Thus, the uncertainty δz in z can be represented as:

$$\delta z = \frac{100}{x}\left[\delta y^2 + \delta x^2\left(\frac{y}{x}\right)^2\right]^{1/2}$$

### Statistical power analysis

For the ion channel inhibition in *Figure 3A–D*, we found no significant AP differences in cell volume and LWR except for the DMSO control at 8 ss as discussed in the main text, where we declared an AP difference non-significant if p>5%. To verify whether the number of measured cells was large enough to conclude that the true AP differences were smaller than for the control at 8 ss, we performed a statistical power analysis. In particular, we tested against the alternative hypothesis ($H_1$) that the true AP average cell volume difference (or LWR difference) in a given case was the same as for the DMSO control at 8 ss. Given our measured averages and standard deviations of the control case at 8 ss, and using Welch's t-test, we computed for all other cases the so-called type II error rate $\beta$ (or false negative rate), i.e. the probability of wrongly identifying an AP difference as not significant. Results are shown in *Figure 3—source data 1* and the respective statistical power correspond to $1 - \beta$. The type II error probability is always below the conventionally chosen value of 20%, and for the LWR it is always at most 3%.

For the perturbation of epithelial junctions in *Figure 6A–C*, we found no significant AP differences only for 2 ss. Analogous to above, we compute the type II error rate $\beta$ for all 2 ss cases based on measured averages and standard deviations for the MO control at 8 ss (*Figure 6A*). Results are shown in *Figure 6—source data 1*, and the type II error rate is always at most 8%.

### Statistics

The significance of pairwise differences between groups of biological data was computed by Welch's two-tailed t-test.

## Acknowledgement

We thank Jeffrey Saffitz, Stephanie Woo and Maura Grealy for sharing reagents. We also thank members of the Amack and Manning groups for helpful discussions and Gonca Erdemci-Tandogan for providing critical feedback on the manuscript. A special thank you to Fiona Foley and Sharleen Buel for outstanding technical support. This work was supported by NIH grants R01HL095690 (JDA) and R01GM117598 (MLM). Additional support was provided by a grant from the Simons Foundation (#446222 MLM and MM), the Research Corporation for Scientific Advancement through the Cottrell Scholars Program (MLM and MM), and through the Gordon and Betty Moore Foundation (MLM and MM). Computing infrastructure support was provided through NSF ACI-1541396.

## Additional information

### Funding

| Funder | Grant reference number | Author |
|---|---|---|
| Simons Foundation | #446222 | Matthias Merkel<br>M Lisa Manning |
| Research Corporation for Scientific Advancement | Cottrell Scholar program | Matthias Merkel<br>M Lisa Manning |
| Gordon and Betty Moore Foundation | | Matthias Merkel<br>M Lisa Manning |
| National Institutes of Health | R01GM117598 | M Lisa Manning |
| National Institutes of Health | R01HL095690 | Jeffrey D Amack |

The funders had no role in study design, data collection and interpretation, or the decision to submit the work for publication.

### Author contributions

Agnik Dasgupta, Conceptualization, Data curation, Formal analysis, Validation, Investigation, Visualization, Methodology, Writing—original draft, Writing—review and editing; Matthias Merkel, Conceptualization, Software, Investigation, Writing—original draft, Writing—review and editing; Madeline J Clark, Investigation, Methodology, Writing—review and editing; Andrew E Jacob, Validation, Investigation, Methodology; Jonathan Edward Dawson, Conceptualization, Software, Investigation; M Lisa Manning, Jeffrey D Amack, Conceptualization, Resources, Supervision, Funding acquisition, Writing—original draft, Writing—review and editing

### Author ORCIDs

Agnik Dasgupta (iD) http://orcid.org/0000-0003-0860-1006
Matthias Merkel (iD) http://orcid.org/0000-0001-9118-1270
Jonathan Edward Dawson (iD) http://orcid.org/0000-0001-9770-8475
Jeffrey D Amack (iD) http://orcid.org/0000-0002-5465-9754

### Decision letter and Author response

Decision letter https://doi.org/10.7554/eLife.30963.036
Author response https://doi.org/10.7554/eLife.30963.037

## Additional files

### Supplementary files

• Transparent reporting form
DOI: https://doi.org/10.7554/eLife.30963.032

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
