## [Decision Letter]

[Editors’ note: a previous version of this study was rejected after peer review, but the authors submitted for reconsideration. The first decision letter after peer review is shown below.]

Thank you for submitting your work entitled "Asymmetric cell volume changes regulate epithelial remodeling of the left-right organizer" for consideration by *eLife*. Your article has been reviewed by three peer reviewers, and the evaluation has been overseen by a Senior/Reviewing Editor.

Our decision has been reached after consultation between the reviewers. Based on these discussions and the individual reviews below, we regret to inform you that your work will not be considered further for publication in *eLife*.

Specifically, all three reviewers were enthusiastic about many aspects of your manuscripts but also asked for rather detailed changes that are likely to take considerable time. This includes improving the illustrations, improving the quality of the results, increasing statistical significance, and improving the writing. It is the policy of *eLife* to reject papers if it would take longer than two months to make the necessary revisions. Therefore, we have no choice but to decline the present version of your manuscript. However, if you feel you can address the comments of the reviewers, detailed below, we would be happy to consider a new version of the paper as a new submission and every effort would be made to return the paper to the same reviewers. I hope you find the attached reviews useful in revising the manuscript.

*Reviewer #1:*

This paper describes experiments and a model exploring the relationships among geometric changes and the forces of morphogenesis, an understudied topic that deserves more exposure. I believe this paper could be a model of an important way to study morphogenesis. My recommendation is for this elegant series of hypothesis-driven experiments, which combine mechanics with molecular tools and mathematical hypothesis generation, to be published with some revision.

Strengths: Adding the third dimension to studies of morphology changes raises the bar in an important way. The study itself examines interactions among the players, including an epithelium, a lumen, and the surrounding cells, rather than characterizing a single component of the system, such as one ion channel. The trajectory of the paper, from observations of 3D morphology, to a model used to generate hypotheses about the underlying mechanisms including force generation and material properties, to explicit tests of those hypotheses using modern molecular techniques, is a good model of how these types of experiments can be conducted.

Weaknesses: the illustrations are not as clear as they need to be, and labels are inconsistent making it difficult to follow, particularly the first half, (up through the description of the model). The first half of the document is difficult to read in a way that I associate with an author who knows a subject so well that she or he forgets to provide the reader with the "obvious" background needed.

1) There is no discussion of total system volume. I wanted to know what is getting pushed where as the lumen increases in volume. I quickly counted the number of external cells in Figure 3, and there were fewer at 8ss. Where did they go? It doesn't seem like the decrease in volume of posterior KV cells would be sufficient for the KV as whole to have a constant volume. A clearer explanation of the boundary conditions is needed in the text.

2) I think it is imperative that they add a description of the effect of the drugs and the MOs on the rest of the embryo, especially the external cells. That, plus the differences in the apparent extent of the KV in Figure 2, also made me want to know about total volumes and relative versus absolute changes in shape and size.

3) It was not clear to me why knocking out E-cadherin directly is lethal, but knocking out its accumulation is not. Is it just a dosage effect, or is something else going on?

4) The manuscript should be revised one more time, preferably by someone not intimately familiar with the project, who can identify where and why clarity is lacking. I've made some specific suggestions below, but it needs to be revised for consistency in terms (x,y,z or l,w,h or apical-basal etc.) and illustrations (line graphs vs. bar graphs, consistency of colors – in different places red means lumen, untransformed KV cell, anterior cells only, etc.). The section describing the model, in particular, needs to include more information up front – even with long experience interpreting models, I had to go back and forth to the supplements many times before finding the background I needed to understand the model and its predictions.

Highly recommended: Since the authors have the 3D data, I was disappointed that the model was nevertheless simplified to 2D. I have to assume the authors are heading in that direction, so I was torn about whether to list this under required. I chose to recommend it highly on the basis that the model as is clearly provides useful testable predictions, which are cleverly explored experimentally in the latter half of the paper. But, it would be great if the model could be that much more representative of the actual geometry.

*Reviewer #2:*

The study by Dasgupta and colleagues show for the first time asymmetric volume changes in the KV and argue that this asymmetry contributes to the asymmetric cell shape changes contribute to KV cell shape changes. They also provide experiments to show that inhibiting lumen expansion does not affect cell shape changes associated with KV morphogenesis. Although asymmetric volume change represents an interesting potential mechanism, the paper is severely hampered by the quality of the results, and specifically, the huge standard deviations for some experiments. I also have some doubts about the impact of the study considering that the described process lead to change in 20% of the size of the KV and that the Amack group showed that left-right patterning occurs normally even though KV size can change vary more than 30% between embryos (see Gokey et al. 2015).

First main point of paper: There are asymmetric cell volume changes that correlate with asymmetric cell shape changes.

The presented data are not completely convincing. The reason is the approach used by the authors is not validated. For example the authors never made sure that the results are not simply due to photobleaching, changes in depth, etc. The authors should perform alternative cell volume measurements in KVs where cells are labeled in a mosaic with a membrane marker either by late mRNA injection, DNA injection or even cell transplantation and assess if the signal and therefore the cell volume measurements affected by the location of cells within the KV or by the stage of the embryos (the organ is deeper?) or by photo-bleaching (prolonged imaging). These points need to be addressed since the volume change reported here is only of about 20% between 2 and 8 ss. Overall the best would be to provide a comprehensive analysis of cell surface or volume and present convincing results (and significant data for statistical analysis) of cell volume changes throughout the KV. At present the authors are dependent on mosaic experiments where the cells location is random and we are left with data presenting huge variability with scales on the plots that are not consistent and comparable between conditions (see Figures: 1, 2 and 5).

Given that their simulations are in 2D, it might help to also perform a simpler experiment where they label cells exactly as they did in Wang 2012, and see if they see the expected differences in cross-sectional area between anterior and posterior cells. They could also see if their cross-sectional area measurements match up with their volume data (i.e. assume that the cells are columnar and calculate volume).

In addition, the authors need to clarify the following points:

- Sample size and number of independent experiments;

- Experimental design; e.g. cell volume changes between 2 and 8 ss: did they follow the same cells overtime and therefore show single cell growth or shrinkage or simply measured the volume of different cells in embryos at different stages and compared average cell volumes?

- Explain better the statistical tests and how the analysis was done; e.g. How was the cell volume change quantified?

They write in the Discussion:

"The finding that KV epithelial cells change volume during development is surprising and insightful since previous 2D analyses had not identified any statistically significant changes in KV cell size."

Their previous study showed that cells reduced their cross-sectional area during morphogenesis (Wang 2012), suggesting that cells reduced their volume, but this study shows for first time that these volume changes are asymmetric. This point should be clarified.

Second main point of paper: Inhibiting ion channels inhibits cell shape changes.

Large error bars and inconsistent scales are a problem.

Third main point of paper: In simulations, asymmetric cell volume changes could account partially for cell shape changes, provided that the mechanical properties of the cells satisfy certain conditions.

The issue about mechanical properties of cells is important. It is unclear if the anterior and posterior mechanical properties of the KV environment are different and how this contributes to KV morphogenesis. If the mechanical properties of surrounding cells are important, could they simulate what happens when there are asymmetric differences in the environment (e.g. If the anterior surrounding cells are solid like, but basal surrounding cells are fluid like, etc.), or when there are asymmetric differences in the KV cells (when anterior KV cells are solid like, but basal KV cells are fluid like, etc.).

Given that the effect of cell volume changes relies on the condition that surrounding cells are solid like, it would be important to perform an experiment to test that. It brings a different light to the Wang et al., 2012 paper, which used blebbistatin and MOs to disrupt mechanical properties of KV cells. Could some of the effects they saw be due to the changes in mechanical properties of surrounding cells rather than the KV cells themselves?

Given that their simulations show that cell volume changes alone could not account for cell shape changes, it is weird that in Figure 2 cell shape changes are completely gone.

Is cell volume change somehow a prerequisite for cell shape changes via actomyosin contractility? They suggest there could be a link in the Discussion, referencing Saias et al., 2015.

Considering the importance of actomyosin in the process of KV morphogenesis, it is important to show what happens to KV cell volume (anterior and posterior) when myosinII/rock2b is inhibited. It might be also good to check phosphorylated myosin via antibody staining (as they did in Wang 2012) to see if inhibiting ion channels somehow affects actomyosin contractility.

Repeat experiments in Figure.5 with DFC specific knockdown and possibly in lgl mutant, which is available.

Fourth main point: Lumen expansion does not affect KV cell shape change.

This is Figure 5. Once again, the huge error bars and differences between controls are problematic.

Fifth and concluding point: Inhibiting cell volume changes disrupts cell shape changes.

Here key experiments are missing. The authors never demonstrate that asymmetric (anterior-posterior) cell volume changes are necessary or contribute KV morphogenesis. The authors should directly test the impact of cell volume change on KV morphogenesis and in particular the impact of asymmetric volume change between anterior and posterior. Photomorpholinos could provide temporal and spatial specificity. If they can get it to work, blocking cftr in anterior vs. posterior cells, and blocking cftr in left vs. right cells to see if an additional axis of asymmetry arise (they could also input this into their simulations) would be necessary.

Finally, the authors need to rethink their final drawing because it is really misleading.

Reviewer #3:

The work of Dasgupta et al. uses the Kupffer's vesicle of zebrafish as a model to study the epithelial morphogenesis, in particular the possible role of programmed cell volume changes and the relationship between lumenogenesis and epithelial remodeling.

Positive aspects of the manuscript include: Methodology: the use of an elegant mosaic labelling approach to perform 3D volumetric and shape analysis of single epithelial cells in vivo. Results: (a) The description of AP asymmetry in cell volume in the KV, with A cells increasing while P decreasing volume between 2 and 8 somites (this data add to the previously described AP asymmetry in cell shape, seen in 2D, which the authors called "tissue remodeling"). (b) The finding that ion flux is required not only for KV lumen expansion (as previously described) but also for the establishment of AP polarity in cell volume. (c) The experimental separation of AP changes in cell shape and volume from lumen expansion, which indicate that the former changes might be autonomous and not a response to external forces generated during lumen expansion. Results from a-c led show that changes in KV cell volume might contribute to epithelial remodeling independent from the forces generated by lumen expansion.

My main concerns of the paper are:

Methodology:

a) The authors mix results and discussion of experimental data of cell shape (measured in 2D) with volume changes (in 3D). This is not correct and could lead to wrong interpretations. For making a clearer connection between the two, parameters of cell shape should incorporate the 3 principal axes. Also, the authors equal the term "epithelial remodeling" with the changes in cell shape in 2D. The authors should restrict the use of the term epithelial remodeling when addressing the epithelial changes in a broader manner, for example, in the introduction and discussion, and use "changes in cell shape" in the other cases.

b) The authors developed a mathematical approach using a 2D vertex model to test the relationship between changes in cell shape and volume and lumen expansion. The model works well for testing the relationship between cell shape changes and lumen expansion in 2D. However, the authors extend the interpretation of the results to the changes in cell volume, assuming that changes in cell cross sectional areas (2D) are equivalent to the changes in cell volume (3D). However, this is not necessarily the case, and thus all the conclusions and predictions that involve changes in cell volume are not valid. The authors should be very careful and make more explicit the limitations of the model. Although the use of 2D models for 3D data is in many cases useful, it is not in this particular case, where the changes of cell volume are central for the mechanisms.

Results:

It remains unclear what the main conclusions of the paper are. The authors propose that changes in cell volume work upstream of the changes in tissue remodeling but there is no direct evidence of this, nor they explore the mechanisms that could mediate this process.

Also, it remains unexplored what makes the changes in cell volume to be AP asymmetric, and whether these changes are really "programmed autonomously" or if the mechanisms that control cell volume are symmetric but modulated in an asymmetric manner by non-autonomous forces (e.g. asymmetric ECM deposition).

Finally, the lack of requirement of lumen expansion for KV tissue remodeling and changes in cell volume is an interesting finding, but again this is not explored further.

In summary, the paper provides interesting methodology and data, but in its current state has a lack of focus, which makes unclear its main contribution to our understanding of epithelial morphogenesis.

[Editors’ note: what now follows is the decision letter after the authors submitted for further consideration.]

Thank you for submitting your article "Asymmetric cell volume changes regulate epithelial morphogenesis in zebrafish Kupffer's vesicle" for consideration by *eLife*. Your article has been reviewed by three peer reviewers, and the evaluation has been overseen by Marianne Bronner as the Senior and Reviewing Editor. The following individual involved in review of your submission has agreed to reveal her identity: Dany S Adams (Reviewer #1).

The reviewers have discussed the reviews with one another and the Reviewing Editor has drafted this decision to help you prepare a revised submission.

While the reviewers felt that the manuscript was much improved, there remain important issues that need to be addressed, as outlined below in the specific comments of reviewers 2 and 3. We hope you find their comments useful and look forward to receiving your revised manuscript.

*Reviewer #1:*

The clarity of the manuscript is greatly improved, and extra data that I requested has been collected, analyzed thoughtfully, and integrated. I believe it is a strong paper that makes a contribution to our understanding of KV development and, more generally, the interplay of different forces during morphogenesis. I also believe it makes a strong argument for the importance of addressing questions about morphogenetic forces and shape generation, and I think the structure of the project could serve as a useful model for how to conduct such studies. Specifically, the sequence they present, careful description of 3D morphology at multiple time points, followed by hypothesis generation (in their case using modeling), followed by experimental testing of those hypotheses, is a useful outline for these kinds of studies. I look forward to following this project, especially if these authors are able to expand their models into 3D the way they have expanded their descriptions. I recommend publication.

*Reviewer #2:*

The manuscript by Dasgupta et al. describes how asymmetric cell shape changes between the anterior and posterior cells of the zebrafish Kupffer's vesicle (KV). Using 3D analysis of single cells at defined single time points, the authors show that asymmetric cell shape changes in length, width and volume (but not in height, xz) occur between 4 and 6 somite-stages (ss) and that lumen expansion does not primarily contribute to asymmetric cell shape changes.

Although the conclusion from the latter part is convincing, what drives asymmetric cell shape changes is largely unclear from the 3D analysis other than involvement of ion channels. The authors emphasize that live embryos are imaged at one stage to avoid potential photobleaching. This approach is good to initially identify the crucial parameters and duration. However, to identify the mechanisms underlying asymmetric cell shape changes requires the analysis of single cell behavior continuously during the critical time points (from 4ss to 6ss) based on time-lapse movies.

There are several concerns to be clarified.

1) It is confusing about the term "asymmetric cell volume changes", which reflects asymmetric cell shape changes in length and width but not in height (Figure 2). What is the biological significance of asymmetric cell volume changes from a mechanistic point of view? Would it be all explained by changes in length and width (2D)? Also, it is ambiguous to include this term in the title.

2) Despite the fact that the authors found out the crucial time point in this process between 4 ss and 6 ss (Figure 2), the majority of their analyses have been done at 2 ss and 8 ss stages. The problem is that the experiments using ion channel inhibitors have been treated from tailbud to 2ss or tailbud to 8ss. Because of the effects of those ion channel inhibitors on height of both anterior and posterior cells at 2 ss, the authors should treat with those inhibitors from 4ss to 6ss to see if there is the effect only on the morphometric properties of posterior cells. How could ion channels mediate asymmetric cell shape changes if lumen expansion (apical expansion) were not important in this process?

3) I am uncertain about how informative the modeling part is (Figure 4). The vertex model simulations are basically done in 2D without measurements for the tensions in anterior and posterior cells and the external forces during the transition between 4ss and 6ss. The limitation is there is no data showing shape changes of a single cell in the anterior or posterior KV during this crucial transition (4ss to 6ss) based on time-lapse movies. However, I cannot assess on the mathematical modeling using solid-like or fluid-like cells that eventually leads to the conclusion that cell shape changes do not depend on lumen expansion.

4) There is inconsistency of the data upon ouabain treatment at 2ss (Figure 3 vs. Figure 3—figure supplement 2). The former shows a reduction of the lumen size, whereas the latter shows that the lumen size looks unaffected.

5) What is the biological significance of cell height changes at 2 ss in embryos treated with ion channel inhibitors or lgl2-morphant embryos? It is confusing, as cell height does not significantly change between anterior and posterior cells from 2ss to 8ss.

*Reviewer #3:*

The study by Dasgupta and colleagues show for the first time asymmetric volume changes in the KV and argue that this asymmetry contributes to the asymmetric cell shape changes contribute to KV cell shape changes. They also provide experiments to show that inhibiting lumen expansion does not affect cell shape changes associated with KV morphogenesis. The authors use a nice combination of experiments and modeling to assess this interesting issue. This study proposes a new process that could at work in the process of KV morphogenesis that promises to start exciting new avenues of investigation. Overall the article is a great piece of work and has been significantly improved since the last revision. We have a few comments.

1) Inhibiting ion channels inhibits cell shape changes/asymmetric cell changes in KV are separable from lumen expansion.

The statistical analysis. Ouabain treatment and cftr knock down lead to similar conclusion which is great. However, both experiments are based on the fact that the results are not significant in the treated embryos at 2SS and 8SS. For this type of analysis, the authors should provide a power analysis to demonstrate that the sample size is enough to conclude about not significant results. Similar comment for jupa and igl2 treatments.

2) Mechanical properties of the surrounding cells. In simulations, asymmetric cell volume changes could account partially for cell shape changes, provided that the mechanical properties of the cells satisfy certain conditions.

The issue about mechanical properties of cells is important. The mechanical properties of surrounding cells are important and need to be solid like at least anteriorly. Please provide reasonable explanation as to why anterior and posterior could be solid like and more generally, what are the arguments implying that mesodermal cells are solid like (this can be part of the Discussion). The 'pinning' hypothesis is interesting in that aspect and could be discussed. Lance Davidson’s work could be discussed as it seems that the notochord is a pretty rigid structure from his work. In addition, could the fact that fibronectin is enriched in the anterior side of the KV could increase the possibility that cells are solid like anteriorly?

3) Cell volume changes and actomyosin. Given that their simulations show that cell volume changes alone could not account for cell shape changes, it is weird that in Figure 3 cell shape changes are completely gone. A way to interpret these data is that ouabain does not specifically target cell volume but also affects elements modulating cell shape, such actomyosin. This raises the question of the effect of ouabain on the actomyosin network. The demonstration is that the actomyosin network is not altered by the treatment. The authors should check it with phosphorylated myosin via antibody staining (as they did in Wang 2012) to see if inhibiting ion channels somehow affects actomyosin contractility. Or is it a problem with the model?

4) The conclusion 'Asymmetric cell volume changes regulate epithelial morphogenesis'. It should be toned down providing that we do not know if the surrounding cells are solid like and that we do not know how the actomyosin network is compromised by the treatments used. The term 'regulate' seems too strong at this point.

---

## [Author Response]

[Editors’ note: the author responses to the first round of peer review follow.]

Reviewer #1:[…] Weaknesses: the illustrations are not as clear as they need to be, and labels are inconsistent making it difficult to follow, particularly the first half, (up through the description of the model). The first half of the document is difficult to read in a way that I associate with an author who knows a subject so well that she or he forgets to provide the reader with the "obvious" background needed.

Thank you for making this point. We have made significant revisions to the text and figures to improve clarity and consistency. Additional background has been added to the text and new Figure 1.

1) There is no discussion of total system volume. I wanted to know what is getting pushed where as the lumen increases in volume. I quickly counted the number of external cells in Figure 3, and there were fewer at 8ss. Where did they go? It doesn't seem like the decrease in volume of posterior KV cells would be sufficient for the KV as whole to have a constant volume. A clearer explanation of the boundary conditions is needed in the text.

Our apologies that the manuscript did not make this clear. We do not change the

number of cells in our simulations. Therefore, in the simulations, the increase in lumen area is accommodated by a (small) decrease in the preferred area for each of the external cells. The force-balanced states depicted in former Figure 3(now Figure 4) and in the related figure supplements do not represent the whole system. We cropped these images to focus on the KV itself rather than all the external cells. Therefore, the number change you observed is due to cropping. We now added the explanation of the cropping into the main text and the caption of Figure 4. In addition, we now also include Figure 4—figure supplement 1, where the full system is depicted as an example for the case of Figure 4. Also, Figure 4—source data 2now includes the explanation that the preferred areas of the external cells at 8 ss is chosen such that the total preferred area of the whole system stays constant. Also, we agree that it is an interesting question how the KV volume change is accommodated by the external cells in the experiments. From preliminary experiments in which we have tracked cell nuclei (expressing mCherry) in the tailbud region, it is clear that there are at least 10-20 cell diameters between the KV and the edge of the embryo and the nuclei density is not significantly disturbed, so it seems plausible that the volume change is distributed throughout the external cells and does not change much about the external cell geometry. We plan to focus on the fluidity, geometry, and mechanical properties of external cells in future work, but we think this large endeavor is beyond the scope of this manuscript.

2) I think it is imperative that they add a description of the effect of the drugs and the MOs on the rest of the embryo, especially the external cells. That, plus the differences in the apparent extent of the KV in Figure 2, also made me want to know about total volumes and relative versus absolute changes in shape and size.

Thank you for making this suggestion. We have now examined the effect of the

pharmacological inhibitor ouabain on the rest of the embryo and specifically on external cells surrounding the KV. We focused on ouabain treatments since it is already known that Cftr is expressed specifically in KV cells (and not external cells around the KV) and that loss of Cftr function does not alter tissues other than KV between 2 ss and 8 ss (Navis et al., 2013). This analysis is included in new Figure 3—figure supplement 2and has been described in the Results:

“Since these treatments were global, we wanted to test whether blocking ion channels altered other tissues in the embryo, including cells surrounding KV that could have an impact on KV cell shapes. […] These results indicate that ouabain does not alter the geometry of cells surrounding KV and suggest that defects in KV cell shape changes result from altered ion flux in KV”.

We did not repeat this analysis for Jup MO and Lgl2 MO treatments (Figure 6) since KV cells (shape changes and volume changes) are normal in these treatments. The differences in the apparent extent of the KV arise from natural variations in the size of KV from embryo to embryo. We recently investigated KV variation and found that KV size can vary significantly among wild-type embryos (Gokey et al., 2016). The reason for this is that KV size is not tightly regulated, but rather only needs to exceed a certain (relatively small) threshold to function normally to establish left-right asymmetry. This source of variability has been clarified in the revised text. To corroborate size changes at the single-cell level, we measure the total volume of the KV cellular component and KV lumen at 2 ss and 8 ss stages and found asymmetric volume changes that were similar to our single cell analysis (Figure 2—figure supplement 2). We report the raw data (absolute size/volumes of single cells) at 2 ss and 8 ss stages (Figure 3 and Figure 6) from all our control and treatment groups. However, given the variability in KV size (discussed above), we have focused the manuscript on relative changes rather than absolute changes. In other words, it is not the amount of volume change, but rather the AP asymmetry of the volume changes—anterior KV cells always increase in size and posterior cells always decrease in size—that is important for KV morphogenesis.

3) It was not clear to me why knocking out E-cadherin directly is lethal, but knocking out its accumulation is not. Is it just a dosage effect, or is something else going on?

Yes, this is a dosage effect. Loss-of-function mutations in E-cadherin result in the arrest of epiboly movements of deep cells at the 70-80% epiboly stage development (8 hpf) prior to the appearance of KV (11 hpf) (Kane et al., 2005). With the MO dose we used in Jup knockdown embryos, E-cadherin enrichment at lateral KV membranes is only moderately reduced (~22% decrease) (Figure 5—figure supplement 3). These knockdown embryos complete gastrulation and appear similar to controls except for KV (Figure 5), indicating Ecadherin is maintained at levels sufficient for epiboly movements and KV formation. However, our functional results (see Figure 5) suggest loss of Jup that leads to a moderate reduction in E-cadherin expression weakens the cell-cell adhesions between KV cells relative to wild-type. This has been clarified in the text.

4) The manuscript should be revised one more time, preferably by someone not intimately familiar with the project, who can identify where and why clarity is lacking. I've made some specific suggestions below, but it needs to be revised for consistency in terms (x,y,z or l,w,h or apical-basal etc.) and illustrations (line graphs vs. bar graphs, consistency of colors – in different places red means lumen, untransformed KV cell, anterior cells only, etc.).

Thank you for these suggestions. We recruited two additional readers that have now critically evaluated our manuscript and figures and made helpful comments. We have made significant revisions to the text and figures to improve clarity and consistency. Additionally, in the schematics and in the graphs we have now consistently pseudo-colored KV-ant and KV-post cells associated with the middle plane of the organ as blue and red, respectively.

The section describing the model, in particular, needs to include more information up front – even with long experience interpreting models, I had to go back and forth to the supplements many times before finding the background I needed to understand the model and its predictions.

We have now extensively revised the modeling section and the corresponding

supplement. See below for details. We hope that clarity has significantly improved.

Highly recommended: Since the authors have the 3D data, I was disappointed that the model was nevertheless simplified to 2D. I have to assume the authors are heading in that direction, so I was torn about whether to list this under required. I chose to recommend it highly on the basis that the model as is clearly provides useful testable predictions, which are cleverly explored experimentally in the latter half of the paper. But, it would be great if the model could be that much more representative of the actual geometry.

This is an excellent point, which we carefully considered ourselves in writing this

manuscript. We are in fact headed in the direction of 3D models and we agree that it is important, but in order to use such a model for the KV, we first have to understand the simple, bulk mechanical properties of 3D models for tissues, which has not really been studied before.

Some of us have a preprint that just appeared online (Merkel and Manning, 2017), showing that there is indeed a fluid-solid transition in the 3D model. We next need to investigate whether adding additional interfacial tensions (as we’ve done here in 2D) are numerically stable, as there are some subtleties about numerical stability in 3D that we are actively working through now. Therefore, we didn’t want to use a model we do not yet fully understand here, and, as discussed below, we do not think it is necessary to explain the science.

For us, the question one must always ask with modeling is whether the mechanism that one is proposing as biologically relevant can survive realistic perturbations to the model. In this case, the mechanism we propose is that solid-like behavior of the surrounding tissue can “pin” the interface between the anterior and posterior KV cells so that it remains at the mid-plane despite changes to cell volume and cross-sectional area, and this helps to generate the observed shape changes. We have added a new figure and text to highlight this mechanism.

Now, we must ask whether this mechanism would survive a (major) perturbation to the model, which is going from 2D to 3D. We provide data that demonstrates that the 3rd dimension (e.g. the length of the cell along the dorsal-ventral axis, which we call the “height”) contributes much less differential changes to cell volume than changes in the cross-sectional areas. We have added a table Figure 4—source data 1, which shows that the AP-asymmetry in cell volumes at 8ss significantly alters the AP-asymmetry of cross sectional areas, but only marginally alters the height asymmetry. This suggests that KV remodeling in the middle plane perpendicular to the DV axis can be understood independently of the precise processes that control cellular DV extension. Specifically, one can imagine that the mechanism we have identified in 2D works the same way in 3D: when pinning the A-P interface at the middle plane, the average apical areas of the A and of the P cells are also pinned respectively. Thus, cell volume changes would directly affect the apico-basal dimension of these cells, leading to a change in aspect ratio. Of course, this relies on the fact there is solid-like pinning of the interface in 3D, and we have added a note and a reference in the modeling section to note the work from our preprint showing that solid-fluid transitions do exist in a 3D version of the model.

Reviewer #2:The study by Dasgupta and colleagues show for the first time asymmetric volume changes in the KV and argue that this asymmetry contributes to the asymmetric cell shape changes contribute to KV cell shape changes. They also provide experiments to show that inhibiting lumen expansion does not affect cell shape changes associated with KV morphogenesis. Although asymmetric volume change represents an interesting potential mechanism, the paper is severely hampered by the quality of the results, and specifically, the huge standard deviations for some experiments. I also have some doubts about the impact of the study considering that the described process lead to change in 20% of the size of the KV and that the Amack group showed that left-right patterning occurs normally even though KV size can change vary more than 30% between embryos (see Gokey et al. 2015).

The sizeable standard deviations that can come with measuring KV cells are due to the wide variability in the size of KV from embryo to embryo. These standard deviations are important, as they accurately reflect the degree of variability in the system. We now report all raw cell volume data as an average + one standard deviation in the revised text (Results section). As noted, we recently reported a comprehensive study of the natural variation of KV size and number of ciliated cells in several wild-type and transgenic strains (Gokey et al., 2016).

Direct comparisons of KV size with left-right patterning (a readout for KV function) in individual embryos revealed that KV size only needed to exceed a certain (relatively small) threshold to function normally. Thus, we agree with the reviewer that the amount that the KV cells change in size/volume, which varies from embryo to embryo, may not be that important. However, it is clear from our results that wild-type KV cells always change size in an asymmetric way along the anterior-posterior (AP) axis. Anterior KV cells always increase in size and posterior cells always decrease in size. These asymmetric cell size changes are linked to asymmetric cell shape changes along the AP axis that we know from previous studies are critical for KV function.

We have modified the Results section and added a new paragraph to the Discussion section to clarify the point that it is the AP asymmetry rather than the absolute value of volume changes that are important for KV morphogenesis and function.

First main point of paper: There are asymmetric cell volume changes that correlate with asymmetric cell shape changes.The presented data are not completely convincing. The reason is the approach used by the authors is not validated. For example the authors never made sure that the results are not simply due to photobleaching, changes in depth, etc. The authors should perform alternative cell volume measurements in KVs where cells are labeled in a mosaic with a membrane marker either by late mRNA injection, DNA injection or even cell transplantation and assess if the signal and therefore the cell volume measurements affected by the location of cells within the KV or by the stage of the embryos (the organ is deeper?) or by photo-bleaching (prolonged imaging). These points need to be addressed since the volume change reported here is only of about 20% between 2 and 8 ss.

We apologize that there was confusion about our methodology. We have now substantially revised the text and added new figures and movies (Figure 1 and Video 2) to clarify our approach to 3D measurements of KV cells.

We did not use an alternative labeling approach (mRNA injection, DNA injection, cell transplantation), since each approach would be subject to the same potential imaging artifacts (e.g. photobleaching or tissue depth). Instead, we took several precautions to avoid these artifacts. In our initial experiments, time-lapse imaging (Video 3) was used to identify cell dynamics during KV development. However, to avoid potential photobleaching problems, live embryos in subsequent experiments were imaged only once at one stage of development – not continuously or at multiple stages. Thus, using Figure 2as an example, different live embryos were imaged and analyzed at 2 ss, 4 ss, 6ss and 8 ss stages and representative cell morphologies from each stage are depicted.

To avoid differences in fluorescence signal due to differences in imaging depth, cells were not selected at random positions in the KV, but rather had to reside at the middle plane along the AP axis (defined as where the lumen diameter is largest) of the KV in all embryos analyzed. This was not made clear in our original manuscript and is now emphasized in the text. All images were also visualized in an YZ orientation (Figure 2) to determine that signals were not affected by depth. This methodology has now been clarified in the text, figures and videos (Figure 2 and Video 2).

Overall the best would be to provide a comprehensive analysis of cell surface or volume and present convincing results (and significant data for statistical analysis) of cell volume changes throughout the KV. At present the authors are dependent on mosaic experiments where the cells location is random and we are left with data presenting huge variability with scales on the plots that are not consistent and comparable between conditions (see Figures: 1, 2 and 5).

Mosaic labeling indeed randomly labeled KV cells. By selecting enough labeled cells from several embryos, we sampled KV cells from all positions along the middle plane of KV.

This is now shown in new Figure 2—figure supplement 1. To corroborate size changes observed in randomly labeled single cells, we also measure the total volume of the KV cellular component and KV lumen at 2 ss and 8 ss stages and found asymmetric volume changes that were similar to our single cell analysis Figure 2—figure supplement 2.

We also adjusted the scales on our graphs so that they are consistent and all data are comparable from figure to figure. We have emphasized that AP asymmetry, rather than the specific amount of volume change, is important for KV morphogenesis. This has been clarified in the text and added to the Discussion.

Given that their simulations are in 2D, it might help to also perform a simpler experiment where they label cells exactly as they did in Wang 2012, and see if they see the expected differences in cross-sectional area between anterior and posterior cells. They could also see if their cross-sectional area measurements match up with their volume data (i.e. assume that the cells are columnar and calculate volume).

We did this analysis (see Figure 4—source code 1) and KV cell cross-sectional areas do correlate with KV cell volume changes. In particular, Figure 4—source code 1clearly shows that the AP volume difference at 8ss is fully accounted for by the AP difference in cross-sectional areas, but not by an AP cell height difference.

We also performed an analysis to see if the cross-sectional area matches with volume data and cell height data. As the reviewer suggests, we first thought to assume that the cells were columnar, but we quickly realized that simple assumptions like cuboidal cell shapes were not enough to account for cell volumes based on cell cross-sectional areas and cell heights, but that an effective geometrical factor was needed, which accounted for effects like cell-cell interfacial curvature. The need for such a factor was consistent with our observation of curved cell interfaces and its value was always approximately two, but the precise value varied depending on experimental stage and cell type (A/P). Thus, interpreting these results in a systematic way was difficult and so we left this computation out of the manuscript.

In addition, the authors need to clarify the following points:- Sample size and number of independent experiments;

Thank you for pointing this out. This information has now been clarified in the figures and figure legends:

Figure 2: Results are pooled from three independent experiments. The number of cells and embryos analyzed is indicated in the graph in panel C and explained in figure legend.

Figure 3: All the experiments including DMSO control, pharmacological treatments and morpholino injections were repeated two separate times at each stage (2 ss and 8 ss). The number of cells and embryos analyzed is indicated in the graphs and figure legend.

Figure 5: Immunofluorescence experiments were performed three times and representative images are shown here. For immunoblotting experiment, quantitative analyses of band intensities were performed on three independent control and knockdown experiments. Average normalized band intensity data from those experiments are represented in the graph.

Figure 5: Control MO and *jupa* MO knockdown experiments were repeated four independent times and rescue experiments were performed three independent times. DFC specific knockdown of Jup was performed two independent times. Dominant negative *Jup-naxos* mRNA was also injected two separate times to validate MO knockdown phenotype. This is clarified in the figure legend. The number of embryos (N) analyzed for each experiment is indicated under the graphs.

Figure 5: Each experiment was repeated two times and the total number of embryos analyzed (N) is denoted in the graph.

Figure 5: Each experiment was repeated two times and the total number of embryos analyzed (N) is denoted in the graph.

Figure 6: Control MO and *jupa* MO knockdown experiments were repeated three independent times and *lgl2* MO experiment was repeated two separate times at each stages (2 ss and 8 ss). The number of embryos (N) and cells analyzed are mentioned in the graphs and in the figure legend.

Figure 2—figure supplement 2: Total KV anterior and posterior cellular component volumes were measured from two independent experiments at each stage (2 ss and 8 ss) and average values are plotted. Number of embryos (N) analyzed at each stage is denoted in the graph.

Figure 3—figure supplement 2: Experiments were repeated two independent times at each stage (2 ss and 8 ss) and number of embryos (N) and cells (n) analyzed are mentioned in the graphs.

Figure 5—figure supplement 3: Control MO and *jupa* MO knockdown experiments were repeated two independent times and number of embryos analyzed (N) is denoted in the graph.

- Experimental design; e.g. cell volume changes between 2 and 8 ss: did they follow the same cells overtime and therefore show single cell growth or shrinkage or simply measured the volume of different cells in embryos at different stages and compared average cell volumes?

We used both approaches, but all data presented in the paper are averages that

come from pooling results obtained from imaging different cells in different embryos at different stages. This method was selected to guard against potential imaging artifacts as discussed above. Our experimental design has now been clarified in the revised text:

“To investigate the dynamics of KV cells in 3D, we first performed time-lapse imaging of mosaic labeled KVs in live *Tg(sox17:CreERT2;ubi:Zebrabow)* embryos from the 2-somite stage (2 ss) when the lumen first forms to the 8 somite stage (8 ss) when the lumen is fully expanded (Amack et al., 2007, Wang et al., 2012, Gokey et al., 2016). […] By analyzing enough embryos, we sampled KV cells from all positions along the middle plane of KV at different stages of development (Figure 2—figure supplement 1).”

- Explain better the statistical tests and how the analysis was done; e.g. How was the cell volume change quantified?

We have modified our images and now show the raw data (previously represented in the supplementary images) of KV cell volumes at 2 ss and 8 ss instead of percentage volume changes (Figure 3, Figure 6). In wild-type embryos at 2 ss the KV-ant and KV-post cells have very similar volumes and cell shapes. This changes at 8 ss where KV-ant cells increase their volume and KV-post cells decrease their volume and become significantly asymmetric. These asymmetric cell volume changes result in asymmetric in cell shapes. Inhibiting ion flux perturbs asymmetric cell volume changes and asymmetric cell shape changes. This has been clarified in the main text.

They write in the Discussion:"The finding that KV epithelial cells change volume during development is surprising and insightful since previous 2D analyses had not identified any statistically significant changes in KV cell size."Their previous study showed that cells reduced their cross-sectional area during morphogenesis (Wang 2012), suggesting that cells reduced their volume, but this study shows for first time that these volume changes are asymmetric. This point should be clarified.

We have revised the Discussion to clarify this point:

“The finding that KV epithelial cells change volume during development is insightful for thinking about mechanisms of KV morphogenesis since previous analyses (Compagnon et al., 2014, Wang et al., 2012) that were limited to 2D had not predicted any differences in KV cell size. […] It is therefore a striking that 3D analysis shows that KV cells do indeed change volume, and do so asymmetrically along the AP axis.”

Second main point of paper: Inhibiting ion channels inhibits cell shape changes.Large error bars and inconsistent scales are a problem.

As discussed above, the error bars represent the natural variation in KV observed in a population of embryos. Even with variability, sampling enough embryos allowed us to identify statistically significant differences. We have changed the scales in our graphs such that they are now consistent throughout the paper.

Third main point of paper: In simulations, asymmetric cell volume changes could account partially for cell shape changes, provided that the mechanical properties of the cells satisfy certain conditions.The issue about mechanical properties of cells is important. It is unclear if the anterior and posterior mechanical properties of the KV environment are different and how this contributes to KV morphogenesis. If the mechanical properties of surrounding cells are important, could they simulate what happens when there are asymmetric differences in the environment (e.g. If the anterior surrounding cells are solid like, but basal surrounding cells are fluid like, etc.).

Thank you for this suggestion. We now additionally studied the influence of

asymmetries in the external cells. As may have been expected from our proposed mechanism (that solid-like cells “pin” the interface between the anterior and posterior cells at the middle plane during volume changes), we find that it does make a difference whether only anterior or only posterior external cells were solid or all external cells were solid.

In particular, if only posterior external cells were solid (Figure 4—figure supplement 4), the AP asymmetry in KV cell shape was much weaker than if only anterior external cells were solid (Figure 4—figure supplement 4), because they do a poor job of “pinning” the interface between anterior and posterior cells. In contrast, when only anterior external cells are solid-like, the pinning works well and the APA is very slightly larger than in the case where all external cells were solid. We now added a discussion of these simulations to the main text.

In future work, we are planning to extensively study the fluidity, geometry, and mechanical properties of the external cells in more detail, in order to better constrain their effects on KV mechanics, but we believe this is beyond the scope of the current manuscript.

Or when there are asymmetric differences in the KV cells (when anterior KV cells are solid like, but basal KV cells are fluid like, etc.).

The case of solid anterior KV cells and fluid posterior KV cells is already included in our simulations. For small numbers of cells, the APA is also positive. We have added a brief discussion of this case to the main text and a more extensive discussion to a dedicated section in the supplement.

Given that the effect of cell volume changes relies on the condition that surrounding cells are solid like, it would be important to perform an experiment to test that. It brings a different light to the Wang et al., 2012 paper, which used blebbistatin and MOs to disrupt mechanical properties of KV cells. Could some of the effects they saw be due to the changes in mechanical properties of surrounding cells rather than the KV cells themselves?

This is an interesting point. Our previously reported blebbistatin treatments and

global MO knockdowns of the Rho kinase Rock2b (Wang et al., 2012) likely impact the mechanical properties of both KV cells and surrounding cells. We previously did an experiment to test whether actomyosin contractility in KV cells vs. surrounding cells is involved in KV cell shape changes (Wang et al., 2011). To test this, MOs against Rock2b were injected into midblastula stage embryos for DFC/KV specific knockdown. KV cell shape changes failed to occur in embryos with Rock2b knocked down in KV cells, even though surrounding cells were normal. This indicates that cell-autonomous function of Rock2b and actomyosin activity is important for KV cell shape changes. Of course, this doesn’t rule out a role for surrounding cells, and future studies are needed to test this possibility. This point is now addressed in the Discussion.

Given that their simulations show that cell volume changes alone could not account for cell shape changes, it is weird that in Figure 2 cell shape changes are completely gone.Is cell volume change somehow a prerequisite for cell shape changes via actomyosin contractility? They suggest there could be a link in the Discussion, referencing Saias et al., 2015.Considering the importance of actomyosin in the process of KV morphogenesis, it is important to show what happens to KV cell volume (anterior and posterior) when myosinII/rock2b is inhibited. It might be also good to check phosphorylated myosin via antibody staining (as they did in Wang 2012) to see if inhibiting ion channels somehow affects actomyosin contractility.

We agree that is interesting and important to tease apart the relationships among ion flux, cell volume changes and cytoskeletal contractility. The suggested experiments are excellent and we plan to pursue this line of investigation in future work. As suggested, there are several possible outcomes. First, as mentioned in our Discussion section, contractile forces that generate cell shape changes may be directly linked to cell volume changes. In the KV system, it will be interesting to test whether AP asymmetric volume changes result in differential cytoskeletal contractility between KV-ant and KV-post cells. To do this, we are generating fluorescent reporters of actomyosin dynamics in KV cells to analyze this in real time.

A second possibility is that actomyosin contractility is necessary for cells to change volume. Here, we can use treatments (e.g. blebbistating, Rock2b MO, and *mypt1* mRNA) to perturb contractility and then assay volume changes. In addition, we are building genetic tools to address this possibility. Since we have established the sox17:CreERT2 transgenic line, we are using CRISPR to introduce loxP sites into cytoskeletal regulatory genes such as Rock2b. This system would allow us to delete Rock2b specifically in KV and then follow contractility, volume changes and shape changes.

A third possibility, as pointed out, is that ion flux might influence contractility independent of volume changes. This may be difficult to separate out, but we will use antibody staining and fluorescent reporters to assess contractility in KV at early stages prior to volume changes.

Finally, it is possible that the relationship between cell volume changes, ion flux and cytoskeletal contractility is not linear, but rather more complex and interdependent on one another. Additional experiments and potentially new mathematical models will be needed if this is the case.

Of note, in response to this comment, we did few blebbistatin treatments in mosaic labeled embryos to get an idea of what might be happening. Preliminary results were inconsistent, suggesting either technical problems (variable drug efficacy or wrong dose) or that the relationship between contractility and cell volume is complex (see possible outcomes discussed above).

We feel that the experiments proposed to test the interplay between cell volume changes and cytoskeletal contractility represent a new line of investigation that goes beyond the scope of this paper and any new results would not change the conclusions of the present manuscript.

Repeat experiments in Figure 5 with DFC specific knockdown and possibly in lgl mutant, which is available.

As suggested, we used DFC/KV-specific knockdown to determine whether Jup

functions cell-autonomously in KV cells to regulate KV lumen expansion. Similar to global Jup knockdowns, the DFC/KV-specific knockdowns resulted in a reduced KV lumen size without affecting the number of ciliated KV cells. These new data, presented in Figure 5, indicate Jup functions in KV cells cell-autonomously to regulate KV lumen expansion. This has been added to the Results section.

We did not repeat experiments in old Figure 5(which is now Figure 6) with DFC/KV-specific knockdowns for two reasons: 1) global and DFC/KV-specific knockdowns have the same effect/phenotype (Figure 5) and 2) KV cell behaviors (shape changes and volume changes) in knockdown embryos are normal, similar to control embryos (Figure 6). If we had observed a defect in KV cell shape or volume changes in global knockdowns, we would want to test whether this was a cell-autonomous effect using DFC/KV-specific knockdowns. But this is not the case here.

Unfortunately, available lgl2 mutants are not useful for KV studies. As previously described (Sonawane et al., 2005) lgl2 mutants do not have phenotypes during early development (prior to 4 days post-fertilization) due to maternal contribution. Only MOs that target both maternal and zygotic expression develop KV phenotypes (Tay et al., 2013).

Fourth main point: Lumen expansion does not affect KV cell shape change.This is Figure 5. Once again, the huge error bars and differences between controls are problematic.

A discussion of error bars and differences among control embryos appears above

and is now included in the revised text.

Fifth and concluding point: Inhibiting cell volume changes disrupts cell shape changes.Here key experiments are missing. The authors never demonstrate that asymmetric (anterior-posterior) cell volume changes are necessary or contribute KV morphogenesis.

Our work with mosaic labeled cells has uncovered asymmetric volume changes

along the AP axis of KV that occur at the same developmental timing as AP asymmetric cell shape changes that were previously described. A previous study that used ouabain treatments in inhibit ion flux and lumen expansion (Compagnon et al., 2014) predicted that the forces associated with rapid KV lumen growth impacts asymmetric KV cell shape changes. Here, we show that in addition to blocking lumen expansion, inhibiting ion flux also blocked asymmetric changes in KV cell volume. Thus, the failure of cells to change shape when ion flux was altered could be due to blocked lumen expansion or blocked volume changes or both.

To test this, we blocked lumen expansion by creating a leaky KV. In these embryos (Figure 6), asymmetric volume changes occurred normal in KV even though the lumen didn’t expand. KV cells also completed normal cell shape changes in the absence of normal lumen expansion.

From these experiments, we conclude that asymmetric cell shape changes that occur in KV depend on ion flux-mediated cell volume changes and do not depend on extrinsic mechanical forces associated with lumen expansion. We also used vertex models to test whether changes in cell volume (cross-sectional area) contribute to KV cell shape changes. This has been clarified in the Results section:

“First, to investigate whether lumen expansion is necessary to create an asymmetry in KV cell elongation, we repeated the numerical simulations shown in Figure 4 – which included both asymmetric cell cross-sectional area changes and increase in lumen cross-sectional area between 2 ss and 8 ss – except in this simulation we kept the lumen cross-sectional area fixed (Figure 4). […] These results suggest that in an environment in which cells have solid-like mechanical properties, asymmetric volume changes in KV cells can partially drive asymmetric KV cell shape changes even in the absence of lumen expansion (Figure 4).”

Taken together, these results indicate asymmetric cell volume changes contribute to asymmetric cell shape changes in KV. These results have been emphasized for clarity in the revised manuscript.

The authors should directly test the impact of cell volume change on KV morphogenesis and in particular the impact of asymmetric volume change between anterior and posterior. Photomorpholinos could provide temporal and spatial specificity. If they can get it to work, blocking cftr in anterior vs. posterior cells, and blocking cftr in left vs. right cells to see if an additional axis of asymmetry arise (they could also input this into their simulations) would be necessary.

Whether volume changes in anterior KV cells or posterior KV cells make equal

contributions to overall KV morphogenesis is very interesting question that we plan to address in future work. In addition to the suggested photo-MO experiments, we are currently generating genetic tools to test the contributions of asymmetric volume changes. We have identified a handful of genes that are asymmetrically expressed in KV– either in anterior cells or posterior cells – that we are using to generate transgenic lines that can overexpress or interfere with ion channels. As suggested, we plan to integrate results from these experiments with our models to fully test the contribution of asymmetric volume changes to KV form and function. We agree with the reviewer and predict that asymmetric ion channel activity is driving asymmetric volume changes. However, we feel that rigorously testing this hypothesis is beyond the scope of this paper.

Finally, the authors need to rethink their final drawing because it is really misleading.

We have significantly simplified Figure 7and have clarified that this is a working

model. In this model, cell volume changes (represented in 2D by changes in cell cross-sectional area) introduce cell shape changes along the AP axis of the KV in wild-type embryos. Perturbations of ion flux inhibit lumen expansion, prevent cell volume (or cell cross-sectional area) changes and prevent KV cell shape changes. Weakening junction integrity inhibited normal lumen expansion, but asymmetric cell volume and cell shape changes occurred normally, albeit to a lesser extent compared to wild type.

Reviewer #3:[…] My main concerns of the paper are:Methodology:a) The authors mix results and discussion of experimental data of cell shape (measured in 2D) with volume changes (in 3D). This is not correct and could lead to wrong interpretations.

We have worked to ensure that we more clearly separate now between our

experimental findings about the 3D KV and our 2D modeling of a cross section through the center of the KV along a plane that is perpendicular to the dorso-ventral (DV) axis. We also explain more clearly now that volume changes are correlated with cross-sectional area changes, and that our simulations show that asymmetric changes in the KV cell areas can induce asymmetric KV cell shape changes.

For making a clearer connection between the two, parameters of cell shape should incorporate the 3 principal axes.

Cell parameters for all 3 axes (L, W and H) were measured using 3D rendered cells and are reported in the paper. We chose to use a length to width ratio (LWR) to describe cell shapes for two reasons: 1) to make connections with our previous work (done in 2D) that used LWRs to describe cell shape changes (Wang et al., 2011, Wang et al., 2012) and 2) to make connections with simulations using vertex models. In our simulations, we quantify cell shape anisotropy based on a LWR, where the length corresponds to the apico-basal extension of the cells, and the width corresponds to the lateral extension perpendicular to the DV axis. Thus, the LWR quantified in the simulations corresponds to the LWR quantified in the experiments.

Since the 2D simulations focus on a plane perpendicular to the DV axis, the cell extension along the DV axis (called “height”) is not included in the modeling. Please see our response to reviewer 1 Point #6 for a detailed discussion of our justification for this choice. We have also edited the paper significantly to address this point.

Also, the authors equal the term "epithelial remodeling" with the changes in cell shape in 2D. The authors should restrict the use of the term epithelial remodeling when addressing the epithelial changes in a broader manner, for example, in the introduction and discussion, and use "changes in cell shape" in the other cases.

We agree and have reformulated. We now are explicit about ‘changes in cell shape’ and ‘changes in cell volume.’

b) The authors developed a mathematical approach using a 2D vertex model to test the relationship between changes in cell shape and volume and lumen expansion. The model works well for testing the relationship between cell shape changes and lumen expansion in 2D. However, the authors extend the interpretation of the results to the changes in cell volume, assuming that changes in cell cross sectional areas (2D) are equivalent to the changes in cell volume (3D). However, this is not necessarily the case, and thus all the conclusions and predictions that involve changes in cell volume are not valid. The authors should be very careful and make more explicit the limitations of the model. Although the use of 2D models for 3D data is in many cases useful, it is not in this particular case, where the changes of cell volume are central for the mechanisms.

We agree that one has to be careful when studying the effect of cell volume changes using a 2D model. However, we have explicitly measured the changes in cell volume and cell cross-sectional area in the plane perpendicular to the dorsal-ventral axis (e.g. where our 2D model resides) as well as the “height” of cells – their length along the DV axis. These are shown in the main text in Figure 2and Figure 4—source data 1 and Figure 4—source data 2. These data demonstrate that differential cell volume changes between anterior and posterior cells correlate with cell cross-sectional area changes while heights of anterior and posterior cells remain approximately same. This gives us confidence to base our 2D simulations on the change of the cell cross-sectional areas. We have clarified this in the main text. Please also see our detailed response to reviewer 1 Point#6 for a discussion of why we expect that the mechanism identified by our 2D model is valid in 3D.

Results:It remains unclear what the main conclusions of the paper are. The authors propose that changes in cell volume work upstream of the changes in tissue remodeling but there is no direct evidence of this, nor they explore the mechanisms that could mediate this process.

In this work, we found experimental conditions that separate the process of lumen expansion (and its associated biophysical forces) from the process that drives cell shape changes in KV. We conclude that ion flux and its associated asymmetric volume changes contribute to cell shape changes during epithelial morphogenesis in KV. We have explored possible mechanisms of how asymmetric KV cell volume changes create asymmetric KV cell shapes. However, we might not have explained this clearly enough.

To improve our manuscript, we have modified the main text to more clearly explain our conclusions and the mechanism that is responsible for asymmetric KV cell shapes in the simulations when the external cells are solid. In addition, we created illustrations of these mechanisms for both presence and absence of lumen expansion (Figure 4). Moreover, we added to the main text a brief explanation of another mechanism that can create asymmetric KV cell shapes in the case where anterior KV cells are solid, posterior KV cells are fluid, and there is only a small number of KV cells. A more detailed discussion of this second mechanism was added to the supplement.

Also, it remains unexplored what makes the changes in cell volume to be AP asymmetric, and whether these changes are really "programmed autonomously" or if the mechanisms that control cell volume are symmetric but modulated in an asymmetric manner by non-autonomous forces (e.g. asymmetric ECM deposition).Finally, the lack of requirement of lumen expansion for KV tissue remodeling and changes in cell volume is an interesting finding, but again this is not explored further.

As discussed above in response to reviewer #2, we are very interested in

determining how KV cell volumes change in an asymmetric way and how this asymmetry contributes to KV form and function. We are generating several genetic tools to perturb volume changes and test whether they are cell-autonomous. We will also perform experiments and simulations to test the role of external forces that include ECM and the mechanical properties of surrounding cells. Multiple approaches will be needed to tease apart contributions from cell autonomous and non-cell autonomous effects, which we feel is beyond the scope of this paper.

In summary, the paper provides interesting methodology and data, but in its current state has a lack of focus, which makes unclear its main contribution to our understanding of epithelial morphogenesis.

Thank you for this feedback. We have revised the manuscript text to more sharply focus on the goals and the main contributions of this study. We set out to develop new mathematical models and experimental methods that allow morphometric analysis of epithelial cells in the simple KV organ in order to investigate the interplay between intrinsic and extrinsic mechanisms that contribute to cell shape changes during epithelial morphogenesis. Using this combination of modeling and experimental approaches, we uncovered new mechanistic insights that contribute to our understanding of epithelial morphogenesis:

1) Epithelial cells in KV undergo volume changes that are asymmetric along the AP axis during morphogenesis. KV-ant cells increase in volume and KV-post cells decrease in volume. These AP asymmetric volume changes coincide with AP asymmetric cell shape changes.

2) Ion flux is an intrinsic mechanism that regulates asymmetric cell volume changes and cell shape changes in KV.

3) Mathematical models indicate that mechanical properties of external cells surrounding the KV can impact cell shape changes in the KV. Models predicted that when external cells are solid like, asymmetric cell volume changes in KV cells contribute to cell shape changes even in the absence of lumen expansion. In particular, we have uncovered a possible mechanism for KV cell shape change: if the interface between KV-ant and KV-post cells was pinned, the volume changes directly affects the apico-basal dimension of the cells while the apical area would remain constant. Hence, asymmetric cell volume changes can induce asymmetric cell shape changes.

4) Experiments determined that asymmetric cell volume and shape changes are separable from extrinsic biophysical forces associated with lumen expansion.

Taken together, these results demonstrate ion flux serves as an intrinsic mechanism to regulate asymmetric epithelial morphogenesis in the KV organ and that changes cell morphology can be uncoupled from mechanical forces exerted during lumen expansion.

These take-home points have been clarified throughout the revised text.

[Editors' note: the author responses to the re-review follow.]

Reviewer #2:[…] Although the conclusion from the latter part is convincing, what drives asymmetric cell shape changes is largely unclear from the 3D analysis other than involvement of ion channels. The authors emphasize that live embryos are imaged at one stage to avoid potential photo-bleaching. This approach is good to initially identify the crucial parameters and duration. However, to identify the mechanisms underlying asymmetric cell shape changes requires the analysis of single cell behavior continuously during the critical time points (from 4ss to 6ss) based on time-lapse movies.

Thank you for your comments and this suggestion. We have now performed time-lapse imaging of mosaic labeled KVs between 4 ss and 6 ss and quantified volume dynamics of single cells at 5 minute intervals during this critical time point. This has allowed us to track and quantify changes in KV-ant and KV-post cell morphology in real time. Consistent with our results from analyzing ‘snap shots’ at specific developmental stages (e.g. 2 ss, 4 ss, 6 ss or 8 ss), we found that individual KV-anterior cells increase their volume between 4-6 ss and KV-posterior cells decrease their volume. This live imaging approach sheds light on KV cell behavior and how these cells change their morphology during morphogenesis. Our new results are presented in Figure 2—figure supplement 2 and Video 4.

There are several concerns to be clarified.1) It is confusing about the term "asymmetric cell volume changes", which reflects asymmetric cell shape changes in length and width but not in height (Figure 2). What is the biological significance of asymmetric cell volume changes from a mechanistic point of view? Would it be all explained by changes in length and width (2D)? Also, it is ambiguous to include this term in the title.

We use the term ‘asymmetric cell volume changes’ to describe regional changes in KV cell size along the anterior-posterior (AP) body axis: KV-anterior cells increase in volume over developmental time, whereas KV-posterior cells decrease in size. We apologize for the confusion, and have now clarified this in the Results section of the manuscript (subsection “3D analysis of single cells reveals asymmetric cell volume changes during KV morphogenesis”, last paragraph).

Changes in cell volume are not necessarily tied to changes in cell shape: cells can change volume but remain the same shape or change shape without changing volume. Results presented in this paper indicate KV cell volume changes contribute to KV cell shape changes. Inhibitors of ion flux inhibit lumen expansion and AP asymmetric KV cell volume changes. This indicates ion flux, which drives fluid movements, is necessary for both lumen expansion (previously published) and KV cell volume changes (not surprising and consistent with a role for ion flux in cell volume changes in other systems). When we looked at KV cell shape changes by length and width (2D) in embryos treated with ion flux inhibitors, they did not occur. This indicates that KV cell shape changes depend on 1) lumen expansion, 2) KV cell volume changes or 3) both. Mathematical modeling suggested that KV cell shapes could change if KV cell volume changes occurred without lumen expansion. We next tested this experimentally. When we leave ion flux and asymmetric KV cell volume changes intact and only inhibit lumen expansion, KV cell shape changes (measured by length and width in 2D) occur normally. Taken together, this suggests that KV cell shape changes do not depend on full lumen expansion, but do depend on asymmetric KV cell volume changes. Thus, mechanistically, KV cell volume changes are required (drive) for cell shape changes. This has been clarified in the first paragraph of the Discussion section.

As suggested, we have changed the title to “Cell volume changes contribute to epithelial morphogenesis in zebrafish Kupffer’s vesicle.”

2) Despite the fact that the authors found out the crucial time point in this process between 4 ss and 6 ss (Figure 2), the majority of their analyses have been done at 2 ss and 8 ss stages. The problem is that the experiments using ion channel inhibitors have been treated from tailbud to 2ss or tailbud to 8ss. Because of the effects of those ion channel inhibitors on height of both anterior and posterior cells at 2 ss, the authors should treat with those inhibitors from 4ss to 6ss to see if there is the effect only on the morphometric properties of posterior cells.

Thank you for suggesting this experiment to test the effect of inhibiting ion channels during the critical stages of 4 to 6 ss. Our rationale for starting pharmacological treatments at the tailbud stage was to allow the compounds to penetrate inside the embryo and block ion channel function in KV throughout the lumen expansion process. We have now repeated the mosaic-labeling experiments with ouabain treatments that started at 4 ss and then imaged KVs at 6 ss. Over the course of several trials, we found, unfortunately, that treatments between 4-6 ss did not reduce KV lumen expansion (see Author response image 1 for a typical result). This indicates longer treatments are indeed necessary to effectively block ion channels, and therefore we are unable to assess the effect of acutely blocking ion flux between 4-6 ss.

**Author response image 1. respfig1:** Ouabain treatments between 4-6 ss do not block KV lumen expansion. (**A-B**) Mosaic-labeled KV in a control embryo (**A**) and an embryo treated with ouabain from 4ss to 6ss (**B**). (**C**) Measurements of maximum KV lumen area indicated that ouabain treatments between 4-6 ss were not effective at blocking KV lumen expansion. N=number of embryos analyzed. Error bars=one standard deviation. ns=not significant, (Welch’s T-Test).

How could ion channels mediate asymmetric cell shape changes if lumen expansion (apical expansion) were not important in this process?

We apologize for not making this clear. In wild-type embryos, KV lumen expansion involves extension of apical surfaces of both KV-ant and KV-post cells in dorsoventral axis (represented here as cell height, Figure 2). But, lateral extension (represented here as cell width,) happens only in KV-post cells, not in KV-ant cells due to tight packing and other mechanical influences. Interestingly, lateral extension of KV-post cells can be uncoupled from the dorsoventral extensions, which play a critical role in lumen expansion. Inhibiting lumen expansion by altering junctional integrity hinders dorsoventral expansion of apical surfaces (Figure 6—figure supplement 1)of all KV cells but KV-post cells still lose their volume via ion channel mediated fluid efflux and undergo lateral extension to facilitate asymmetric KV cell shape changes (Figure 6). Thus, ion channels mediate asymmetric cell shape changes via lateral extension of KV-post cells even when overall lumen expansion is inhibited (Figure 6). This has been clarified in the Discussion section.

3) I am uncertain about how informative the modeling part is (Figure 4). The vertex model simulations are basically done in 2D without measurements for the tensions in anterior and posterior cells and the external forces during the transition between 4ss and 6ss. The limitation is there is no data showing shape changes of a single cell in the anterior or posterior KV during this crucial transition (4ss to 6ss) based on time-lapse movies. However, I cannot assess on the mathematical modeling using solid-like or fluid-like cells that eventually leads to the conclusion that cell shape changes do not depend on lumen expansion.

The purpose of our vertex model simulations was not a literal simulation of the precise time-dependent cell shape changes in vivo. As the referee correctly points out, we would be missing a lot of information for that (cell-cell interfacial tensions, external forces, tracking of cell shapes). Rather, the purpose of our simulations was to identify potential mechanisms that could induce an asymmetric LWR in the KV cells. Since we do not know the values of the cell-cell interfacial tensions and the properties of the surrounding cells, we scanned the parameter space for many possible combinations of both, and we found there was a large regime where asymmetric cell volume change drives asymmetric KV cell shape change. Based on these simulations, we could identify a potential mechanism that translates volume asymmetry into cell shape asymmetry (Figure 4). Then we wanted to check whether this mechanism depended on the lumen expansion, and found that even without an expanding lumen, asymmetric cell volume change can drive asymmetric cell shape change (Figure 4).

4) There is inconsistency of the data upon ouabain treatment at 2ss (Figure 3 vs. Figure 3—figure supplement 2). The former shows a reduction of the lumen size, whereas the latter shows that the lumen size looks unaffected.

Thank you for pointing this out. We have now quantified the effect of ouabain on KV lumen expansion and present these results in Figure 3—figure supplement 2. The results were consistent with previous experiments, as reported in Figure 3. We have therefore included a more representative image in Figure 3—figure supplement 2.

5) What is the biological significance of cell height changes at 2 ss in embryos treated with ion channel inhibitors or lgl2-morphant embryos? It is confusing, as cell height does not significantly change between anterior and posterior cells from 2ss to 8ss.

Height changes reflect reduced lumen expansion as compared to controls (see also the response to point #2 above). This difference is already significantly different at 2ss in ouabain, CFTR inh, CFTR MO treated embryos and becomes significant by 8ss in Lgl2 MO treated embryos. In wild-type and all treated embryos, height responds similarly in both anterior and posterior cells to lumen expansion (or lack thereof).

Reviewer #3:[…] 1) Inhibiting ion channels inhibits cell shape changes/asymmetric cell changes in KV are separable from lumen expansion.The statistical analysis. Ouabain treatment and cftr knock down lead to similar conclusion which is great. However, both experiments are based on the fact that the results are not significant in the treated embryos at 2SS and 8SS. For this type of analysis, the authors should provide a power analysis to demonstrate that the sample size is enough to conclude about not significant results. Similar comment for jupa and igl2 treatments.

We have repeated the treatments with ouabain, *cftr* MO and *lgl2* MO to increase the sample size of both KV-ant and KV-post cells. We have updated Figure 3 and Figure 6 accordingly. Moreover, we dedicate a new Materials and methods section and two new tables (Figure 3—source data 1 and Figure 6—source data 1) to a statistical power analysis to substantiate our results of non-significance in Figure 3 (ion channel inhibition) and 6A-C (jupa and lgl2). In particular, for the ion channel inhibition, we test the non-significance against the alternative hypothesis that the AP volume and LWR differences are as high as in the unperturbed case (DMSO control) at 8ss. We find false error rates for the volume always below the conventional chosen value of 20%, and for the LWR even below 5%. This corresponds to a statistical power of more than 80% and 95%, respectively. We performed a similar analysis for the jupa and lgl2 treatments and always found a statistical power of at least 90%.

2) Mechanical properties of the surrounding cells. In simulations, asymmetric cell volume changes could account partially for cell shape changes, provided that the mechanical properties of the cells satisfy certain conditions.The issue about mechanical properties of cells is important. The mechanical properties of surrounding cells are important and need to be solid like at least anteriorly. Please provide reasonable explanation as to why anterior and posterior could be solid like and more generally, what are the arguments implying that mesodermal cells are solid like (this can be part of the Discussion). The 'pinning' hypothesis is interesting in that aspect and could be discussed. Lance Davidson’s work could be discussed as it seems that the notochord is a pretty rigid structure from his work. In addition, could the fact that fibronectin is enriched in the anterior side of the KV could increase the possibility that cells are solid like anteriorly?

Our modeling results indicate that asymmetric cell shape changes in KV are more pronounced if only the anterior external cells have solid-like mechanical properties than if only the posterior external cells are solid-like, as the solid-like cells on the anterior side are able to "pin" the interface between the KV-ant cells and the KV-post cells. In order to determine general mechanisms, we have indeed kept the model very simple – for example, it does not take into account effects of additional tissues/structures (e.g. notochord) or extra-cellular matrix (ECM). It is possible that the notochord – which physically interfaces with KV-ant cells – may be a solid-like structure, and this may provide the pinning mechanism. Additionally, an AP gradient of ECM may help prevent neighbor exchanges and give rise to solid-like behavior only in anterior external cells. It is also possible that the solid-like ECM directly physically pins the KV-ant cells, so that the mechanism we have identified may operate even if the anterior cells beyond the ECM are fluid-like. These possibilities have been expanded upon in the discussion. In addition, we have provided new Video 5 that depicts enrichment of the ECM molecule fibronectin associated with the notochord and anterior external cells, which is consistent with previous findings (Campagnon, et al. 2014. Dev Cell. PMID: 25535919). Our future work will investigate the contributions of notochord and ECM to the mechanical properties of external cells and asymmetric cell shape changes in KV.

3) Cell volume changes and actomyosin. Given that their simulations show that cell volume changes alone could not account for cell shape changes, it is weird that in Figure 3 cell shape changes are completely gone. A way to interpret these data is that ouabain does not specifically target cell volume but also affects elements modulating cell shape, such actomyosin. This raises the question of the effect of ouabain on the actomyosin network. The demonstration is that the actomyosin network is not altered by the treatment. The authors should check it with phosphorylated myosin via antibody staining (as they did in Wang 2012) to see if inhibiting ion channels somehow affects actomyosin contractility. Or is it a problem with the model?

Thank you for bringing up this important point. We propose that cell volume changes contribute to epithelial morphogenesis in KV and are likely linked to additional important factors (e.g. mechanical properties of cells, tissue-tissue interactions between notochord and KV, and actomyosin cytoskeletal contractility) that are not currently in our model. We are quite intrigued about the relationship between cell volume changes and actomyosin contractility in KV epithelial morphogenesis. As suggested here, we used antibodies that detect phosphorylated myosin light chain (pMLC) to test whether treating embryos with ouabain alters non-muscle myosin II activity in KV cells. Our group (Wang, et al. 2012. Developmental Biology. PMCID: PMC3586254) and the Heisenberg group (Compagnon, et al. 2014. Developmental Cell. PMID: 25535919) have previously used this pMLC antibody to qualitatively assess myosin II activity in KV. Over the course of several trials, we detected cortical pMLC staining in control and ouabain treated KV cells that localized to the cortex of KV cells marked by the tight junction protein ZO1 (see Author response image 2). However, it was not clear to us whether pMLC levels were different in treated KVs as compared to controls. Despite significant efforts to make this assay quantitative, we have not overcome variabilities in the antibody staining that appear to be due to variable penetration of the antibody into deep tissue layers where KV is located.

As an alternative approach to measure myosin II activity in KV, we took advantage of our stable transgenic *Tg(actb2:myl12.1-MKATE2)* embryos that express the fluorescent mKate2 protein fused to myosin light chain (Myl12.1). We and others (Compagnon, et al. 2014. Developmental Cell. PMID: 25535919) have found that the accumulation and intensity of Myl12.1-fusion proteins correlates with phosphorylated myosin II antibody staining intensity, and thereby serve as a good proxy for active myosin II. Embryos from homozygous *Tg(actb2:myl12.1-MKATE2)* parents were treated with ouabain starting at the tailbud stage and then KVs in live treated and control embryos were imaged at the 6 somite stage (see Author response image 2). Imaging live KVs eliminated the need to fix embryos and perform the antibody staining procedure. Blind measurements of Myl12.1-mKate2 intensity at randomly selected cell-cell interfaces did not detect a difference between control and ouabain treated KVs (see Author response image 2). Similar results were obtained in multiple experiments, and measurements of KV area (see Author response image 2) were used to test efficacy of ouabain treatments.

Although these results suggest actomyosin is not dramatically changed in KVs treated with ouabain, we cannot rule out more subtle effects. Thus, we do not feel it is appropriate to draw conclusions from this one experimental approach and include these data in the present manuscript. In future work that is beyond the scope of this paper, we will investigate more rigorously the relationships among ion flux, cell volume changes and cytoskeletal contractility.

**Author response image 2. respfig2:** Effect of ouabain treatments on the actomyosin network in KV. (**A**) Antibodies that recognize phosphorylated myosin light chain (pMLC) were used to detect active myosin II at the apical cortex of KV cells marked by ZO1 staining. pMLC staining was present in both control and ouabain treated KVs, but this signal proved difficult to quantify. (**B**) Expression of fluorescent myosin light chain (myl12.1)-mKate2 fusion proteins at the apical cortex of KV cells in live embryos. Dashed yellow lines outline KV lumen. Scale bars=20 mm. (**C**) Expression intensity of myl12.1-mKate2 measured at cell-cell interfaces was found to be similar between control (n=15 interfaces) and ouabain (n=25 interfaces) treated embryos. (**D**) Measurements of maximum KV lumen area indicated that ouabain treatments were effective at blocking ion flux and lumen expansion in these embryos. N=number of embryos analyzed. Error bars=one standard deviation. *p=0.001. ns=not significant, (Welch’s T-Test).

4) The conclusion 'Asymmetric cell volume changes regulate epithelial morphogenesis'. It should be toned down providing that we do not know if the surrounding cells are solid like and that we do not know how the actomyosin network is compromised by the treatments used. The term 'regulate' seems too strong at this point.

We have toned down the conclusions and changed the title to “Cell volume changes contribute to epithelial morphogenesis in zebrafish Kupffer’s vesicle.”